# Why Ask One When You Can Ask $k$? Learning-to-Defer to the Top-$k$ Experts

**Yannis Montreuil**
School of Computing
National University of Singapore
Singapore, 117418
yannis.montreuil@u.nus.edu

**Axel Carlier**
Fédération ENAC ISAE-SUPAERO ONERA
Université de Toulouse
Toulouse, 31555, France

**Lai Xing Ng**
Agency for Science, Technology and Research
Institute for Infocomm Research
Singapore, 138634

**Wei Tsang Ooi**
School of Computing
National University of Singapore
Singapore, 117418

## Abstract

Existing *Learning-to-Defer* (L2D) frameworks are limited to *single-expert deferral*, forcing each query to rely on only one expert and preventing the use of collective expertise. We introduce the first framework for *Top-$k$ Learning-to-Defer*, which allocates queries to the $k$ most cost-effective entities. Our formulation unifies and strictly generalizes prior approaches, including the *one-stage* and *two-stage* regimes, *selective prediction*, and classical cascades. In particular, it recovers the usual Top-1 deferral rule as a special case while enabling principled collaboration with multiple experts when $k > 1$. We further propose *Top-$k(x)$ Learning-to-Defer*, an adaptive variant that learns the optimal number of experts per query based on input difficulty, expert quality, and consultation cost. To enable practical learning, we develop a novel surrogate loss that is Bayes-consistent, $\mathcal{H}_h$-consistent in the one-stage setting, and $(\mathcal{H}_r, \mathcal{H}_g)$-consistent in the two-stage setting. Crucially, this surrogate is independent of $k$, allowing a single policy to be learned once and deployed flexibly across $k$. Experiments across both regimes show that Top-$k$ and Top-$k(x)$ deliver superior accuracy–cost trade-offs, opening a new direction for multi-expert deferral in L2D.

## 1 Introduction

Learning-to-Defer (L2D) enables models to defer uncertain queries to external experts, explicitly trading off predictive accuracy and consultation cost (Madras et al., 2018; Mozannar & Sontag, 2020; Verma et al., 2022). Classical L2D, however, routes each query to a *single* expert. This design is ill-suited for complex decisions that demand collective judgment. For instance, in oncology, patient cases are routinely reviewed by multidisciplinary tumor boards comprising radiologists, pathologists, oncologists, and surgeons. Each specialist contributes a different perspective—imaging, histopathology, treatment protocols, and surgical considerations—and only through aggregation can an accurate and safe recommendation be made (Jiang et al., 1999; Fatima et al., 2017). Similar multi-expert deliberation underpins fraud detection, cybersecurity, and judicial review (Dieterich, 2000). We believe this reliance on a single expert constitutes a *fundamental limitation* of existing L2D frameworks: in many high-stakes domains, deferring to only one expert is not desirable.

Motivated by these challenges, we introduce *Top-$k$ Learning-to-Defer*, a unified framework that allocates each query to the $k$ most cost-effective experts. Our formulation supports both major regimes of L2D. In the *two-stage* setting, all experts are trained offline, and a routing function is then trained to allocate queries either to one of the experts or to a fixed main predictor (Narasimhan et al., 2022; Mao et al., 2023a; 2024c; Montreuil et al., 2025b;a). In contrast, the *one-stage* setting jointly learns the main prediction task and the allocation policy within a single model, allowing

both components to adapt during training (Madras et al., 2018; Mozannar & Sontag, 2020). Our framework admits instantiations in both regimes, ensuring broad applicability.

We further propose *Top-$k(x)$*, an adaptive extension that learns the number of experts to consult per query based on input complexity, expert competence, and consultation cost. To enable both fixed-$k$ and adaptive deferral, we design a novel surrogate loss that is Bayes-consistent, $(\mathcal{H}_r, \mathcal{H}_g)$-consistent in the two-stage setting, $\mathcal{H}_h$-consistent in the one-stage setting, and independent of $k$, allowing efficient reuse across cardinalities without retraining. Finally, we show that our framework strictly generalizes prior paradigms: *selective prediction* (Chow, 1970; Cortes et al., 2016) and classical model cascades (Viola & Jones, 2001; Saberian & Vasconcelos, 2014; Laskaridis et al., 2021), with the usual Top-1 Bayes policy arising as a special case. This situates Top-$k$/Top-$k(x)$ as a unifying and strictly more general framework for Learning-to-Defer.

Our main contributions are: (**i**) We introduce **Top-$k$ L2D**, the first framework for deferral to the top-$k$ experts, unifying both *one-stage* and *two-stage* regimes. (**ii**) We develop a $k$-**independent surrogate loss** with Bayes-, $\mathcal{H}_h$-, and $(\mathcal{H}_r, \mathcal{H}_g)$-consistency guarantees, allowing a single policy to be reused across all values of $k$ without retraining. (**iii**) We show that **classical model cascades** are strictly subsumed as a special case, and that the *usual Top-1 Bayes policy rule* is recovered from both one-stage and two-stage formulations, as well as from **selective prediction** within our framework. (**iv**) We propose **Top-$k(x)$**, an adaptive variant that learns the optimal number of experts per query under accuracy–cost trade-offs. (**v**) We provide **extensive empirical results** demonstrating that Top-$k$ and Top-$k(x)$ consistently achieve superior accuracy–cost trade-offs compared to prior L2D methods.

## 2 RELATED WORK

**Learning-to-Defer.** Learning-to-Defer (L2D) extends selective prediction (Chow, 1970; Bartlett & Wegkamp, 2008; Cortes et al., 2016; Geifman & El-Yaniv, 2017; Cao et al., 2022; Cortes et al., 2024) by allowing models not only to abstain on uncertain inputs but also to defer them to external experts (Madras et al., 2018; Mozannar & Sontag, 2020; Verma et al., 2022). Two main approaches have emerged. In *two-stage* frameworks, the base predictor and experts are trained offline, and a separate allocation function is learned to decide whether to predict or defer (Narasimhan et al., 2022; Mao et al., 2023a), with extensions to regression (Mao et al., 2024c), multi-task learning (Montreuil et al., 2025b), adversarial robustness (Montreuil et al., 2025a; 2026b), online (Montreuil et al.; 2026c), and applied systems (Strong et al., 2024; Palomba et al., 2025; Montreuil et al., 2026a). In contrast, *one-stage* frameworks train prediction and deferral jointly. The score-based formulation of Mozannar & Sontag (2020) established the first Bayes-consistent surrogate and has since become the standard, with follow-up work improving calibration (Verma et al., 2022; Cao et al., 2024), surrogate design (Charusaie et al., 2022; Mao et al., 2024a; Wei et al., 2024), and guarantees such as $\mathcal{H}$-consistency and realizability (Mozannar et al., 2023; Mao et al., 2024b; 2025). Applications span diverse classification tasks (Verma et al., 2022; Cao et al., 2024; Keswani et al., 2021; Kerrigan et al., 2021; Hemmer et al., 2022; Benz & Rodriguez, 2022; Tailor et al., 2024; Liu et al., 2024).

**Top-$k$ Classification.** Top-$k$ classification generalizes standard classification by predicting a set of top-ranked labels rather than a single class. Early hinge-based approaches (Lapin et al., 2015; 2016) were later shown to lack Bayes consistency (Yang & Koyejo, 2020), and non-convex formulations raised optimization challenges (Yang & Koyejo, 2020; Thilagar et al., 2022). More recent advances have established Bayes- and $\mathcal{H}$-consistency for a broader family of surrogates, including cross-entropy (Mao et al., 2023b) and constrained losses (Cortes & Vapnik, 1995), with cardinality-aware refinements providing stronger theoretical guarantees (Cortes et al., 2024).

**Gap.** Existing L2D frameworks are restricted to *single-expert deferral*, a critical limitation: in high-stakes domains, robust decisions demand aggregating complementary expertise, while reliance on a single expert amplifies bias and error. Crucially, no prior work enables top-$k$ or adaptive top-$k(x)$ deferral in either one-stage or two-stage regimes, nor provides surrogate losses with provable consistency guarantees. We address this gap by introducing the first unified framework for Top-$k$ and Top-$k(x)$ L2D, supported by a $k$-independent surrogate loss that ensures statistically sound and cost-efficient multi-expert allocation.

## 3 PRELIMINARIES

Let $\mathcal{X}$ be the input space and $\mathcal{Z}$ the output space, with training examples $(x, z)$ drawn i.i.d. from an unknown distribution $\mathcal{D}$.

**One-Stage L2D.** In the one-stage regime (Madras et al., 2018; Mozannar & Sontag, 2020), prediction and deferral are optimized jointly through a single model in a multiclass setting with label space $\mathcal{Z} = \mathcal{Y} = \{1, \ldots, n\}$, corresponding to $n$ distinct categories. The system has access to $J$ offline experts. In the deterministic case, each expert is a mapping $\hat{m}_j : \mathcal{X} \to \mathcal{Z}$. An expert may be modeled as a stochastic predictor. In this case, its output $M_j$ is defined as a random variable jointly distributed with $(X, Y)$, and training samples include realizations $\hat{m}_j$ drawn from the conditional distribution $\mathbb{P}(M_j \mid X = x, Y = y)$. All our results remain valid when experts are stochastic; the analysis extends verbatim by treating each expert's output as a random variable jointly distributed with $(X, Y)$.

We treat both class labels and experts uniformly as *entities*. The corresponding entity set is

$$\mathcal{A}^{1s} = \{1, \ldots, n\} \cup \{n + 1, \ldots, n + J\},$$

where indices $j \leq n$ correspond to predicting class $j$, and indices $j > n$ correspond to deferring to expert $m_{j-n}$. We define the hypothesis class of score-based classifier as $\mathcal{H}_h = \{h : \mathcal{X} \times \mathcal{A}^{1s} \to \mathbb{R}\}$. For any $h \in \mathcal{H}_h$, the induced decision rule selects $\hat{h}(x) = \arg\max_{j \in \mathcal{A}^{1s}} h(x, j)$, i.e., the entity in $\mathcal{A}^{1s}$ with the highest score. If $\hat{h}(x) \leq n$, the predictor outputs class $\hat{h}(x) \in \mathcal{Y}$; otherwise, it defers to expert $m_{\hat{h}(x)-n}$. The hypothesis $h$ is learned by minimizing the risk induced by the deferral loss (Mozannar & Sontag, 2020; Verma et al., 2022; Cao et al., 2024; Mao et al., 2024a).

**Definition 3.1** (One-Stage Deferral Loss). Let $x \in \mathcal{X}$, $y \in \mathcal{Y}$, and $h \in \mathcal{H}_h$ be a score-based classifier. The one-stage deferral loss is

$$\ell_{\text{def}}^{1s}(\hat{h}(x), y) = \mathbf{1}\{\hat{h}(x) \neq y\}\mathbf{1}\{\hat{h}(x) \leq n\} + \sum_{j=1}^{J} c_j(x, y)\,\mathbf{1}\{\hat{h}(x) = n + j\},$$

with surrogate $\Phi_{\text{def}}^{1s,u}(h, x, y) = \Phi_{01}^u(h, x, y) + \sum_{j=1}^{J}(1 - c_j(x, y))\,\Phi_{01}^u(h, x, n + j)$, where $\Phi_{01}^u$ belongs to the cross-entropy family (Mohri et al., 2012; Mao et al., 2023b). The cost is defined as $c_j : \mathcal{X} \times \mathcal{Y} \to [0, 1]$ with $c_j(x, y) = \alpha_j \mathbf{1}\{\hat{m}_j(x) \neq y\} + \beta_j$, where $\alpha_j \geq 0$ penalizes prediction error and $\beta_j \geq 0$ is a fixed consultation fee.

**Two-Stage L2D.** In the two-stage regime (Narasimhan et al., 2022; Mao et al., 2023a; 2024c; Montreuil et al., 2025b;a), the main predictor and experts are trained offline and remain fixed. Unlike the one-stage setting, where a single augmented classifier jointly performs prediction and deferral, the two-stage approach introduces a separate *rejector* that allocates queries among entities. Formally, we consider an output space $\mathcal{Z}$ and a main predictor $g \in \mathcal{H}_g$ with predictions $\hat{g}(x) \in \mathcal{Z}$, which is fully observable to the system. We also assume access to a collection of $J$ experts $\{\hat{m}_j : \mathcal{X} \to \mathcal{Z}\}_{j=1}^{J}$. We treat both the main predictor and experts uniformly as *entities*. The corresponding entity set is

$$\mathcal{A}^{2s} = \{1, \ldots, J + 1\},$$

where $j = 1$ denotes the base predictor and $j \geq 2$ denotes expert $\hat{m}_{j-1}$. We define the hypothesis class of rejectors as $\mathcal{H}_r = \{r : \mathcal{X} \times \mathcal{A}^{2s} \to \mathbb{R}\}$. For any $r \in \mathcal{H}_r$, scores are assigned to entities, and the induced decision rule is $\hat{r}(x) = \arg\max_{j \in \mathcal{A}^{2s}} r(x, j)$. If $\hat{r}(x) = 1$, the system outputs the base predictor's label $\hat{g}(x)$; otherwise, it defers to expert $m_{\hat{r}(x)-1}$. The deferral loss is then defined as follows.

**Definition 3.2** (Two-Stage Deferral Loss). Let $x \in \mathcal{X}$, $z \in \mathcal{Z}$, and $r \in \mathcal{R}$ be a rejector. The two-stage deferral loss and its convex surrogate are

$$\ell_{\text{def}}^{2s}(\hat{r}(x), z) = \sum_{j=1}^{J+1} c_j(x, z)\mathbf{1}\{\hat{r}(x) = j\}, \quad \Phi_{\text{def}}^{2s,u}(r, x, z) = \sum_{j=1}^{J+1} \tau_j(x, z)\,\Phi_{01}^u(r, x, j),$$

where $c_j : \mathcal{X} \times \mathcal{Z} \to \mathbb{R}_+$ is defined as $c_1(x, z) = \alpha_1 \psi(\hat{g}(x), z) + \beta_1$ with $\psi$ a task-specific penalty (e.g., RMSE, mAP, or 0-1 loss) and $c_j(x, z) = \alpha_j \psi(\hat{m}_j(x), z) + \beta_j$ for $j \geq 2$. The term $\tau_j(x, z) = \sum_{i \neq j} c_i(x, z)$ aggregates the costs of all non-selected entities.

**Consistency.** We restrict attention to the one-stage regime for clarity. The objective is to learn a hypothesis $h \in \mathcal{H}_h$ that minimizes the expected deferral risk $\mathcal{E}_{\ell_{\text{def}}^{1s}}(h) = \mathbb{E}_{X,Y}[\ell_{\text{def}}^{1s}(\hat{h}(X), Y)]$, with Bayes-optimal value $\mathcal{E}_{\ell_{\text{def}}^{1s}}^B(\mathcal{H}_h) = \inf_{h \in \mathcal{H}_h} \mathcal{E}_{\ell_{\text{def}}^{1s}}(h)$. Direct optimization is intractable due to discontinuity and non-differentiability (Zhang, 2002; Steinwart, 2007; Awasthi et al., 2022; Mozannar & Sontag, 2020; Mao et al., 2024a), motivating the use of convex surrogates. A prominent class is the *comp-sum* family (Mao et al., 2023b), which defines cross-entropy surrogates as

$$\Phi_{01}^u(h, x, j) = \Psi^u \left( \sum_{j' \in \mathcal{A}} e^{h(x,j') - h(x,j)} - 1 \right),$$

where the outer function $\Psi^u$ is parameterized by $u > 0$. Specific choices recover canonical losses: $\Psi^1(v) = \log(1 + v)$ (logistic), $\Psi^u(v) = \frac{1}{1-u}[(1 + v)^{1-u} - 1]$ for $u \neq 1$, covering sum-exponential (Weston & Watkins, 1998), logistic regression (Ohn Aldrich, 1997), generalized cross-entropy (Zhang & Sabuncu, 2018), and MAE (Ghosh et al., 2017).

A fundamental criterion for surrogate adequacy is *consistency*, which requires that minimizing surrogate excess risk also reduces true excess risk (Zhang, 2002; Bartlett et al., 2006; Steinwart, 2007; Tewari & Bartlett, 2007). To formalize this, Awasthi et al. (2022) introduced the notion of $\mathcal{H}_h$-consistency bounds, which quantify consistency with respect to a restricted hypothesis class rather than all measurable functions. The following bound has been established in the one-stage L2D setting (Mao et al., 2024a).

**Theorem 3.3** ($\mathcal{H}_h$-consistency bounds). *Suppose the surrogate $\Phi_{01}^u$ is $\mathcal{H}_h$-calibrated for any distribution $\mathcal{D}$. Then there exists a non-decreasing function $\Gamma_u^{-1} : \mathbb{R}_+ \to \mathbb{R}_+$, depending on $u$, such that for all $h \in \mathcal{H}_h$,*

$$\mathcal{E}_{\ell_{\text{def}}^{1s}}(h) - \mathcal{E}_{\ell_{\text{def}}^{1s}}^B(\mathcal{H}_h) + \mathcal{U}_{\ell_{\text{def}}^{1s}}(\mathcal{H}_h) \leq \Gamma_u^{-1}\left(\mathcal{E}_{\Phi_{\text{def}}^{1s,u}}(h) - \mathcal{E}_{\Phi_{\text{def}}^{1s,u}}^*(\mathcal{H}_h) + \mathcal{U}_{\Phi_{\text{def}}^{1s,u}}(\mathcal{H}_h)\right).$$

Here $\mathcal{U}_{\ell_{\text{def}}^{1s}}(\mathcal{H}_h) = \mathcal{E}_{\ell_{\text{def}}^{1s}}^B(\mathcal{H}_h) - \mathbb{E}_X\left[\inf_{h \in \mathcal{H}_h} \mathbb{E}_{Y|X=x}[\ell_{\text{def}}^{1s}(\hat{h}(x), Y)]\right]$ is the *minimizability gap*, which measures the irreducible approximation error due to the expressive limitations of $\mathcal{H}_h$. When $\mathcal{H}_h$ is sufficiently rich (e.g., $\mathcal{H}_h = \mathcal{H}_h^{\text{all}}$), the gap vanishes, and the inequality recovers Bayes-consistency guarantees (Steinwart, 2007; Awasthi et al., 2022). Taking the limit of this bound recovers the same Bayes-consistency established in Mozannar & Sontag (2020).

# 4 GENERALIZING LEARNING-TO-DEFER TO THE TOP-$k$ EXPERTS

## 4.1 FROM SINGLE TO TOP-$k$ EXPERT SELECTION

**Notations.** Prior Learning-to-Defer methods allocate each input $x \in \mathcal{X}$ to exactly one entity, corresponding to a *top-1* decision rule (Mozannar & Sontag, 2020; Verma et al., 2022; Mao et al., 2024a). Formally, this is captured by the one-stage deferral loss in Definition 3.1 or its two-stage counterpart in Definition 3.2. To unify notation across both regimes, we define the hypothesis class of decision rules as $\mathcal{H}_\pi = \{\pi : \mathcal{X} \times \mathcal{A} \to \mathbb{R}\}$. For any $\pi \in \mathcal{H}_\pi$, the function assigns a score $\pi(x, j)$ to each entity $j \in \mathcal{A}$, and the induced selection rule is

$$\hat{\pi}(x) = \arg\max_{j \in \mathcal{A}} \pi(x, j).$$

In the one-stage regime, $\pi$ coincides with the augmented classifier $h$ and $\mathcal{A} = \mathcal{A}^{1s}$, while in the two-stage regime, $\pi$ coincides with the rejector $r$ and $\mathcal{A} = \mathcal{A}^{2s}$. For clarity, we will henceforth use $\mathcal{A}$ without superscripts, with the understanding that it denotes the appropriate entity set for the regime under consideration.

**Top-$k$ Selection.** We generalize L2D to a *top-$k$* rule, where each query may be assigned to several entities simultaneously, enabling multi-expert deferral and joint use of complementary expertise. We first formalize the top-$k$ selection set:

**Definition 4.1** (Top-$k$ Selection Set). Let $x \in \mathcal{X}$ and let $\pi : \mathcal{X} \times \mathcal{A} \to \mathbb{R}$ be a decision rule that assigns a score $\pi(x, j)$ to each entity $j \in \mathcal{A}$. For any $1 \leq k \leq |\mathcal{A}|$, the *top-$k$ selection set* is

$$\Pi_k(x) = \{[1]_\pi^\downarrow, [2]_\pi^\downarrow, \ldots, [k]_\pi^\downarrow\},$$

where $[i]_\pi^\downarrow$ denotes the index of the $i$-th highest-scoring entity under $\pi(x, \cdot)$. The ordering is non-increasing: $\pi(x, [1]_\pi^\downarrow) \geq \pi(x, [2]_\pi^\downarrow) \geq \cdots \geq \pi(x, [k]_\pi^\downarrow)$.

Choosing $k = 1$ recovers the standard top-1 rule $\Pi_1(x) = \{\arg\max_{j \in \mathcal{A}} \pi(x, j)\}$, which corresponds to $\Pi_1(x) = \{\arg\max_{j \in \mathcal{A}^{1s}} h(x, j)\}$ in the one-stage setting (Mozannar & Sontag, 2020; Cao et al., 2024; Mao et al., 2024a) and $\Pi_1(x) = \{\arg\max_{j \in \mathcal{A}^{2s}} r(x, j)\}$ in the two-stage setting (Narasimhan et al., 2022; Mao et al., 2023a; 2024c; Montreuil et al., 2025b).

*Remark* 1. We further show in Appendix A.6 that the Top-$k$ Selection Set subsumes classical cascade approaches (Viola & Jones, 2001; Saberian & Vasconcelos, 2014; Dohan et al., 2022; Jitkrittum et al., 2023) as a strict special case, thereby unifying cascaded inference and multi-expert deferral under a single framework.

**Top-$k$ True Deferral Loss.** L2D losses are tailored to top-1 selection and do not extend directly to $k > 1$. In the one-stage case (Definition 3.1), terms such as $\mathbf{1}\{h(x) \neq y\}\mathbf{1}\{h(x) \leq n\}$ enforce exclusivity, assuming exactly one entity is chosen. This assumption breaks in the top-$k$ setting: the selection set $\Pi_k(x)$ may simultaneously include the true label $y$ and multiple experts with heterogeneous accuracy and cost. A naive extension, e.g. $\mathbf{1}\{y \in \Pi_k(x)\}$, is inadequate for three reasons: (i) it collapses correctness to the mere inclusion of $y$, ignoring whether the consulted experts themselves are reliable; (ii) it fails to account for the cumulative consultation costs incurred when querying several entities; and (iii) it yields non-decomposable set-based indicators, which obstruct surrogate design since the accuracy–cost tradeoff is determined jointly at the *set* level rather than per entity. These issues motivate a reformulation of L2D losses to handle top-$k$ deferral.

Each regime specifies an entity set $\mathcal{A}$ and associated functions $\{\hat{a}_j : \mathcal{X} \to \mathcal{Z}\}_{j \in \mathcal{A}}$:

- One-stage: $\mathcal{A}^{1s} = \{1, \ldots, n + J\}$, where $\hat{a}_j(x) = j$ for $j \leq n$ (predicting label $j$), and $a_{n+j}(x) = \hat{m}_j(x)$ for $j = 1, \ldots, J$ (deferring to expert $m_j$).
- Two-stage: $\mathcal{A}^{2s} = \{1, \ldots, J + 1\}$, where $a_1(x)$ is the base predictor prediction $\hat{g}(x)$, and $a_{1+j}(x) = \hat{m}_j(x)$ for $j = 1, \ldots, J$ (deferring to expert $m_j$).

For any entity $j \in \mathcal{A}$, we define an *augmented cost* $\mu_j(x, z) = \alpha_j \psi(\hat{a}_j(x), z) + \beta_j$, where $\alpha_j, \beta_j \geq 0$, and $\psi$ is a task-specific error measure (the 0–1 loss in classification, or any non-negative loss otherwise). By construction, $\mu_j(x, z) \in \mathbb{R}_+$.

**Definition 4.2** (Top-$k$ True Deferral Loss). Let $x \in \mathcal{X}$, $z \in \mathcal{Z}$, and $\Pi_k(x) \subseteq \mathcal{A}$ be the top-$k$ selection set. Let $\mu_j(x, z)$ the cost of selecting entity $j$ for input $(x, z)$. The uniformized top-$k$ true deferral loss is

$$\ell_{\text{def},k}(\Pi_k(x), z) = \sum_{j=1}^{|\mathcal{A}|} \mu_j(x, z)\mathbf{1}\{j \in \Pi_k(x)\},$$

We give a detailed explanation in Appendix A.7. This loss quantifies the *total cost* of allocating a query to $k$ entities, thereby unifying the one-stage (Definition 3.1) and two-stage (Definition 3.2) objectives into a single formulation that explicitly supports joint decision-making across multiple entities. Unlike classical top-1 deferral, which only evaluates the outcome of a single choice, the top-$k$ loss accumulates both predictive errors and consultation costs across all selected entities.

For instance, in binary classification with $\mathcal{Y} = \{1, 2\}$ and two experts, the entity set is $\mathcal{A} = \{1, 2, 3, 4\}$, where $j \leq 2$ correspond to labels and $j > 2$ to experts. If the top-2 selection set is $\Pi_2(x) = \{3, 1\}$, the incurred loss is $\mu_3(x, y) + \mu_1(x, y)$, jointly reflecting the cost of deferring to both expert $m_1$ and predicting label 1.

*Remark* 2. For $k = 1$, the top-$k$ deferral loss reduces exactly to the classical objectives: the one-stage loss in Definition 3.1 and the two-stage loss in Definition 3.2.

### 4.2 SURROGATES FOR THE TOP-$k$ TRUE DEFERRAL LOSS

In Lemma 4.2, the top-$k$ true deferral loss is defined via a hard ranking operator over the selection set $\Pi_k(x)$. This makes it discontinuous and non-differentiable, hence unsuitable for gradient-based optimization. To enable practical learning, we follow standard practice in Learning-to-Defer (Mozannar & Sontag, 2020; Charusaie et al., 2022; Cao et al., 2024; Mao et al., 2024a; Montreuil et al.,

2025b;a) and introduce a convex surrogate family grounded in the theory of calibrated surrogate losses (Zhang, 2002; Bartlett et al., 2006).

**Lemma 4.3** (Upper Bound on the Top-$k$ Deferral Loss). *Let $x \in \mathcal{X}$, $z \in \mathcal{Z}$, and let $1 \leq k \leq |\mathcal{A}|$. Let $\Phi_{01}^u$ a convex surrogate in the cross-entropy family. Then the top-$k$ deferral loss satisfies*

$$\ell_{def,k}(\Pi_k(x), z) \leq \sum_{j \in \mathcal{A}} \left( \sum_{i \neq j} \mu_i(x, z) \right) \Phi_{01}^u(\pi, x, j) - (|\mathcal{A}| - 1 - k) \sum_{j \in \mathcal{A}} \mu_j(x, z),$$

We prove Lemma 4.3 in Appendix A.8. The key observation is that the cost term $\sum_{j \in \mathcal{A}} \mu_j(x, z)$ does not depend on the decision rule $\pi$, since each $\mu_j(x, z) = \alpha_j \psi(\hat{a}_j(x), z) + \beta_j$ is fixed for a given $(x, z)$ in both the one-stage and two-stage regimes. Furthermore, for all $k \leq |\mathcal{A}|$, we have $\mathbf{1}\{j \in \Pi_k(x)\} \leq \Phi_{01}^u(\pi, x, j)$ (Lapin et al., 2017; Cortes et al., 2024). Consequently, minimizing the upper bound reduces to minimizing only the first term, and the optimization becomes *independent of $k$*. This directly yields the following tight surrogate family:

**Corollary 4.4** (Surrogates for the Top-$k$ Deferral Loss). *Let $x \in \mathcal{X}$, $z \in \mathcal{Z}$, and let $\pi : \mathcal{X} \times \mathcal{A} \to \mathbb{R}$ be a decision rule. The corresponding surrogate family for the top-$k$ deferral loss is*

$$\Phi_{def,k}^u(\pi, x, z) = \sum_{j \in \mathcal{A}} \left( \sum_{i \neq j} \mu_i(x, z) \right) \Phi_{01}^u(\pi, x, j),$$

*which is independent of $k$.*

This independence is a key strength: a single decision rule $\pi$ can be trained once and reused for any cardinality level $k$, eliminating the need for retraining and allowing practitioners to adjust the number of consulted experts dynamically at inference time depending on budget or risk constraints. Algebraically, the surrogate in Corollary 4.4 coincides with the formulation of Mao et al. (2024c), but our derivation shows that this form arises as a convex upper bound *for all $k$*. Thus, the loss expression itself remains unchanged, while our framework extends the underlying deferral objective, the decision rule, and the guarantees from top-1 to the general top-$k$ setting.

However, convexity and boundedness alone do not suffice for statistical validity (Zhang, 2002; Bartlett et al., 2006). Crucially, the fact that our surrogate coincides algebraically with that of Mao et al. (2024c) does not imply that their guarantees transfer: their analysis establishes consistency only in the top-1 regime, leaving the multi-entity case $k > 1$ unresolved. Extending consistency from $k = 1$ to $k > 1$ is generally non-trivial, as shown in the top-$k$ classification literature (Lapin et al., 2015; 2016; 2017; Yang & Koyejo, 2020; Cortes et al., 2024), where set-valued decisions introduce fundamentally different statistical challenges. Closing this gap requires new analysis. In the next subsection, we establish that minimizing any member of the surrogate family $\Phi_{def,k}^u$ yields consistency for both one-stage and two-stage L2D, thereby guaranteeing convergence to the Bayes-optimal top-$k$ deferral policy as the sample size grows.

## 4.3 THEORETICAL GUARANTEES

While recent work by Cortes et al. (2024) has established that the cross-entropy family of surrogates $\Phi_{01}^u$ is $\mathcal{H}_\pi$-consistent for the top-$k$ *classification* loss $\ell_k(\Pi_k(x), j) = \mathbf{1}\{j \in \Pi_k(x)\}$, the consistency of top-$k$ *deferral* surrogates remains unresolved and requires dedicated theoretical analysis. Unlike standard classification, deferral introduces an additional layer of complexity: costs depend jointly on predictive accuracy and consultation with heterogeneous experts, and errors propagate differently depending on whether the system predicts directly or defers. These factors fundamentally alter the Bayes-optimal decision rule, making existing results insufficient. Prior analyses have addressed the $k = 1$ case in one-stage and two-stage settings (Mozannar & Sontag, 2020; Verma et al., 2022; Mao et al., 2024a;c), but extending consistency guarantees to $k > 1$ is non-trivial. Our theoretical analysis fills this gap by proving that the surrogate family $\Phi_{def,k}^u$ is Bayes- and class-consistent for the top-$k$ deferral objective, thereby establishing statistical validity of learning in this more general regime.

To proceed, we impose only mild regularity conditions on the hypothesis class $\mathcal{H}_\pi$: (i) *Regularity*: for any input $x$, the scores $\pi(x, \cdot)$ induce a strict total order over all entities in $\mathcal{A}$; (ii) *Symmetry*: the

scoring rule is invariant under permutations of entity indices, i.e., relabeling entities does not affect the induced scores; (iii) *Completeness*: for every fixed $x$, the range of $\pi(x, j)$ is dense in $\mathbb{R}$.

These assumptions are standard and are satisfied by common hypothesis classes, including fully connected neural networks and the space of all measurable functions $\mathcal{H}_\pi^{\text{all}}$ (Awasthi et al., 2022).

### 4.3.1 OPTIMALITY OF THE TOP-$k$ SELECTION SET

A central challenge in Learning-to-Defer is deciding which entities to consult at test time, given their heterogeneous accuracies and consultation costs. For $k = 1$, prior work has established that the Bayes-optimal policy selects the single entity with the lowest expected cost (Mozannar & Sontag, 2020; Verma et al., 2022; Narasimhan et al., 2022; Mao et al., 2023a; 2024a). The key question we address is how this principle extends to the richer regime $k > 1$. We prove the following Lemma in Appendix A.9.

**Lemma 4.5** (Bayes-Optimal Top-$k$ Selection). *Let $x \in \mathcal{X}$. For each entity $j \in \mathcal{A}$, define the expected cost $\overline{\mu}_j(x) = \mathbb{E}_{Z|X=x}[\mu_j(x, Z)]$, its Bayes-optimal expected cost as $\overline{\mu}_j^B(x) = \inf_{g \in \mathcal{H}_g} \overline{\mu}_j(x)$. Then the Bayes-optimal top-$k$ selection set is*

$$\Pi_k^B(x) = \underset{\substack{\Pi_k \subseteq \mathcal{A} \\ |\Pi_k| = k}}{\arg\min} \sum_{j \in \Pi_k} \overline{\mu}_j^B(x) = \{[1]_{\overline{\mu}^B}^\uparrow, [2]_{\overline{\mu}^B}^\uparrow, \dots, [k]_{\overline{\mu}^B}^\uparrow\},$$

*where $[i]_{\overline{\mu}^B}^\uparrow$ denotes the index of the $i$-th smallest expected cost in $\{\overline{\mu}_j^B(x) : j \in \mathcal{A}\}$. In the one-stage regime, where no base predictor class $\mathcal{H}_g$ is defined, we simply set $\overline{\mu}_j^B(x) = \overline{\mu}_j(x)$.*

Lemma 4.5 shows that Bayes-optimal top-$k$ deferral is obtained by ranking entities according to their expected cost and selecting the $k$ lowest.

**Corollary 4.6** (Special cases for $k = 1$). *The Bayes rule in Lemma 4.5 recovers prior Top-1 results:*

1. ***One-stage L2D.*** *For any entity $j$ (labels $j \leq n$ and experts $j > n$),*
   $$\overline{\mu}_j^B(x) = \alpha_j \mathbb{P}(\hat{a}_j(x) \neq Y \mid X = x) + \beta_j,$$
   *which yields the Top-1 Bayes policy of Mozannar & Sontag (2020).*

2. ***Two-stage L2D.*** *Let $j = 1$ denote the base predictor and $j \geq 2$ the experts. Then*
   $$\overline{\mu}_1^B(x) = \alpha_1 \inf_{g \in \mathcal{H}_g} \mathbb{E}_{Z|X=x}\left[\psi(\hat{g}(x), Z)\right] + \beta_1,$$
   *and for $j \geq 2$, $\overline{\mu}_j^B(x) = \alpha_j \mathbb{E}_{Z|X=x}\left[\psi(\hat{m}_{j-1}(x), Z)\right] + \beta_j$,*
   *recovering the Top-1 allocation in Narasimhan et al. (2022); Mao et al. (2023a); Montreuil et al. (2025b).*

3. ***Selective prediction (reject option).*** *We take the set of label entities and augment it with an abstain entity $\perp$, defined by $\alpha_\perp = 0$ and $\beta_\perp = \lambda > 0$, while label entities use $\alpha_j = 1, \beta_j = 0$. Then*
   $$\overline{\mu}_j^B(x) = \mathbb{P}(j \neq Y \mid X = x) \quad (j \in \{1, \dots, n\}), \qquad \overline{\mu}_\perp^B(x) = \lambda,$$
   *yielding the Chow's rule (Chow, 1970).*

We defer the proof of this corollary and give additional details in Appendix A.10. The Top-$k$ Bayes policy strictly generalizes all prior Top-1 results: it reduces to known rules when $k = 1$, but for $k > 1$ it yields a principled way to combine multiple entities under a unified cost-sensitive criterion.

### 4.3.2 CONSISTENCY OF THE TOP-$k$ DEFERRAL LOSS SURROGATES

Having established the Bayes-optimal policy in Lemma 4.5, we now turn to the surrogate family $\Phi_{\text{def},k}^u$. The central question is whether minimizing the surrogate risk guarantees convergence toward the Bayes-optimal policy for the top-$k$ true deferral loss (Lemma 4.2). This property, known as *consistency*, is crucial: without it, empirical risk minimization may converge to arbitrarily suboptimal policies. While consistency has been established for $k = 1$ in both one-stage (Mozannar & Sontag, 2020; Verma et al., 2022; Mao et al., 2024a) and two-stage (Narasimhan et al., 2022; Mao et al., 2024c; Montreuil et al., 2025b), no prior results extend to the richer regime $k > 1$.

**Theorem 4.7** (Unified Consistency for Top-$k$ Deferral). *Let $\mathcal{A}$ denote the set of entities. Assume that $\mathcal{H}_\pi$ is symmetric, complete, and regular for top-$k$ deferral, and that in the two-stage case, $\mathcal{H}_g$ is the base predictor class. Let $S := (|\mathcal{A}| - 1) \sum_{j \in \mathcal{A}} \mathbb{E}_X[\overline{\mu}_j(X)]$. Suppose $\Phi_{01}^u$ is $\mathcal{H}_\pi$-consistent for top-$k$ classification with a non-negative, non-decreasing, concave function $\Gamma_u^{-1}$.*

***One-stage.*** *Let $\mathbb{E}_X[\overline{\mu}_j(X)] = \alpha_j \mathbb{P}(\hat{a}_j(X) \neq Y) + \beta_j$. For any $h \in \mathcal{H}_h$,*

$$
\mathcal{E}_{\ell_{\text{def},k}}(h) - \mathcal{E}_{\ell_{\text{def},k}}^B(\mathcal{H}_h) + \mathcal{U}_{\ell_{\text{def},k}}(\mathcal{H}_h) \leq k\, S\, \Gamma_u^{-1}\left( \frac{\mathcal{E}_{\Phi_{\text{def},k}^u}(h) - \mathcal{E}_{\Phi_{\text{def},k}^u}^*(\mathcal{H}_h) + \mathcal{U}_{\Phi_{\text{def},k}^u}(\mathcal{H}_h)}{S} \right).
$$

***Two-stage.*** *Let $\mathbb{E}_X[\overline{\mu}_j(X)] = \alpha_j \mathbb{E}_{X,Z}\left[\psi(\hat{a}_j(X), Z)\right] + \beta_j$. For any $(r, g) \in \mathcal{H}_r \times \mathcal{H}_g$,*

$$
\mathcal{E}_{\ell_{\text{def},k}}(r, g) - \mathcal{E}_{\ell_{\text{def},k}}^B(\mathcal{H}_r, \mathcal{H}_g) + \mathcal{U}_{\ell_{\text{def},k}}(\mathcal{H}_r, \mathcal{H}_g) \leq \mathbb{E}_X\left[\overline{\mu}_1(X) - \inf_{g \in \mathcal{H}_g} \overline{\mu}_1(X)\right]
$$
$$
+ k\, S\, \Gamma_u^{-1}\left( \frac{\mathcal{E}_{\Phi_{\text{def},k}^u}(r) - \mathcal{E}_{\Phi_{\text{def},k}^u}^*(\mathcal{H}_r) + \mathcal{U}_{\Phi_{\text{def},k}^u}(\mathcal{H}_r)}{S} \right)
$$

*with $\Gamma_1(v) = \frac{1+v}{2}\log(1+v) + \frac{1-v}{2}\log(1-v)$ (logistic), $\Gamma_0(v) = 1 - \sqrt{1-v^2}$ (exponential), and $\Gamma_2(v) = v/|\mathcal{A}|$ (MAE).*

We give the proof in Appendix A.11. Theorem 4.7 provides the first consistency guarantees for top-$k$ deferral across both one-stage and two-stage regimes. The bounds reveal that the excess deferral risk depends explicitly on $k$: consulting more entities enlarges the cost term $k\,S$. At the same time, calibration of $\Phi_{01}^u$ ensures that minimizing surrogate risk drives the excess true risk to zero, establishing both $\mathcal{H}_h$, $(\mathcal{H}_r, \mathcal{H}_g)$, and Bayes-consistency: learned policies converge to the Bayes-optimal top-$k$ deferral rule from Lemma 4.5 as data grows.

In the two-stage regime, we assume the Bayes-optimal cost is attainable (or can be arbitrarily well approximated), i.e., there exists a sequence $g_t \in \mathcal{H}_g$ such that $\mathbb{E}_X[\overline{\mu}_1(X) - \overline{\mu}_1^B(X)] \to 0$. Furthermore, if there exists $r_t \in \mathcal{H}_r$ with $\mathcal{E}_{\Phi_{\text{def},k}^u}(r_t) - \mathcal{E}_{\Phi_{\text{def},k}^u}^*(\mathcal{H}_r) + \mathcal{U}_{\Phi_{\text{def},k}^u}(\mathcal{H}_r) \to 0$, then by Theorem 4.7 and the fact that $v \mapsto kS\,\Gamma_u^{-1}(v/S)$ is nonnegative and nondecreasing on $[0, \infty)$ with $\Gamma_u^{-1}(0) = 0$, we obtain $\mathcal{E}_{\ell_{\text{def},k}}(r_t, g_t) - \mathcal{E}_{\ell_{\text{def},k}}^B(\mathcal{H}_r, \mathcal{H}_g) + \mathcal{U}_{\ell_{\text{def},k}}(\mathcal{H}_r, \mathcal{H}_g) \to 0$, which shows that the surrogate indeed minimizes its target loss.

The minimizability gap vanishes under realizability and, more generally, whenever the hypothesis class is sufficiently rich, for instance when $\mathcal{H}_\pi = \mathcal{H}_\pi^{\text{all}}$ (Steinwart, 2007). Importantly, by setting $k = 1$, we recover the established $\mathcal{H}_h$-consistency bounds for one-stage L2D (Mao et al., 2024a) and $(\mathcal{H}_r, \mathcal{H}_g)$-consistency bounds for two-stage L2D (Mao et al., 2024c; 2023a; Montreuil et al., 2025b). Thus our result strictly generalizes prior work, while covering the entire cross-entropy surrogate family, including log-softmax, exponential, and MAE. This unification provides the first rigorous statistical foundation for multi-expert deferral.

# 5 TOP-$k(x)$: ADAPTING THE NUMBER OF ENTITIES PER QUERY

While our Top-$k$ deferral framework enables richer allocations than prior works, it still assumes a uniform cardinality $k$ across all queries. In practice, input complexity varies: some instances may require only one confident decision, while others may benefit from aggregating over multiple entities. To address this heterogeneity, we propose an adaptive mechanism that selects a query-specific number of entities.

Following the principle of cardinality adaptation introduced in Top-$k$ classification (Cortes et al., 2024), we define a *cardinality function* $k_\theta : \mathcal{X} \to \mathcal{A}$, parameterized by a hypothesis class $\mathcal{H}_k$. For a given input $x$, the function selects the cardinality level via $\hat{k}_\theta(x) = \arg\max_{i \in \mathcal{A}} k(x, i)$ and returns the Top-$k(x)$ subset $\Pi_{\hat{k}_\theta(x)}(x) \subseteq \Pi_{|\mathcal{A}|}(x)$ produced by the scoring function $\pi(x, \cdot)$.

**Definition 5.1** (Cardinality-Aware Deferral Loss). Let $x \in \mathcal{X}$, and let $\Pi_{\hat{k}_\theta(x)}(x)$ denote the adaptive Top-$k(x)$ subset. Let $d$ denote a metric, $\xi : \mathbb{R}^+ \to \mathbb{R}^+$ a non-decreasing function, and $\lambda \geq 0$ a

regularization parameter. Then, the adaptive cardinality loss is defined as

$$\ell_{\text{card}}(\Pi_{\hat{k}_\theta(x)}(x), \hat{k}_\theta(x), x, z) = d(\Pi_{\hat{k}_\theta(x)}(x), x, z) + \lambda \xi\Big(\sum_{i=1}^{\hat{k}_\theta(x)} \beta_{[i]_\pi^\downarrow}\Big),$$

with surrogate $\Phi_{\text{card}}(\Pi_{|\mathcal{A}|}(x), k_\theta, x, z) = \sum_{v \in \mathcal{A}} \left(1 - \tilde{\ell}_{\text{card}}(\Pi_v(x), v, x, z)\right) \Phi_{01}^u(k_\theta, x, v)$, where $\tilde{\ell}_{\text{card}}$ is a normalized version of the cardinality-aware loss and $\beta_{[i]_\pi^\downarrow}$ denotes the consultation cost of the $i$-th ranked entity. This surrogate is $\mathcal{H}_k$–consistent, and the proof follows the same structure as in Cortes et al. (2024).

The term $d(\Pi_{\hat{k}_\theta(x)}(x), x, z)$ captures the predictive error of the selected set and can be instantiated using top-$k$ accuracy, majority-voting error, or any other task-dependent aggregation metric (see Appendix A.13 for examples). The second component penalizes costly selections, encouraging the model to query additional entities only when the expected accuracy gain justifies the consultation cost.

**Bayes-Optimal Cardinality Selection.** We now characterize the behavior of the adaptive cardinality mechanism induced by the cardinality-aware deferral loss. For a fixed input $x \in \mathcal{X}$ and any cardinality level $k \in \{1, \ldots, |\mathcal{A}|\}$, define the conditional risk associated with selecting the top-$k$ entities as

$$\mathcal{C}_{\ell_{\text{card}}}(k) := \mathbb{E}[\ell_{\text{card}}(\Pi_k(x), k, x, Z) \mid X = x] = \mathbb{E}[d(\Pi_k(x), x, Z) \mid X = x] + \lambda \xi\left(\sum_{i=1}^{k} \beta_{[i]_\pi^\downarrow}\right). \tag{1}$$

The Bayes-optimal cardinality function is therefore given pointwise by

$$k^B(x) \in \arg\min_{k \in \{1, \ldots, |\mathcal{A}|\}} \mathcal{C}_{\ell_{\text{card}}}(k). \tag{2}$$

To understand when it is beneficial to query an additional entity, define the marginal difference in conditional risk

$$\delta\mathcal{C}_{\ell_{\text{card}}}(k) := \mathcal{C}_{\ell_{\text{card}}}(k) - \mathcal{C}_{\ell_{\text{card}}}(k-1), \qquad S_k := \sum_{i=1}^{k} \beta_{[i]_\pi^\downarrow}.$$

A direct decomposition yields

$$\delta\mathcal{C}_{\ell_{\text{card}}}(k) = \underbrace{\mathbb{E}[d(\Pi_k(x), x, Z) - d(\Pi_{k-1}(x), x, Z) \mid X = x]}_{:= \delta D_x(k)} + \lambda[\xi(S_k) - \xi(S_{k-1})]. \tag{3}$$

Consequently, increasing the cardinality from $k$ to $k+1$ strictly decreases the conditional risk if and only if

$$\mathcal{C}_{\ell_{\text{card}}}(k+1) \leq \mathcal{C}_{\ell_{\text{card}}}(k) \iff -\delta D_x(k+1) \geq \lambda[\xi(S_{k+1}) - \xi(S_k)]. \tag{4}$$

Equation equation 4 provides a transparent decision rule for adaptive querying: an additional entity is selected if and only if the expected reduction in predictive error obtained by aggregating it outweighs its marginal consultation cost. The regularization parameter $\lambda$ directly controls this trade-off, interpolating between conservative policies that favor small cardinalities and accuracy-driven regimes that aggregate over larger sets when beneficial. This characterization shows that adaptive Top-$k(x)$ deferral is a principled consequence of risk minimization rather than a heuristic design choice.

## 6 EXPERIMENTS

We evaluate our proposed methods—*Top-k L2D* and its adaptive extension *Top-k(x) L2D*—against state-of-the-art one-stage (Mozannar & Sontag, 2020; Mao et al., 2024a) and two-stage (Narasimhan

et al., 2022; Mao et al., 2023a; 2024c; Montreuil et al., 2025b) baselines. Across all tasks, *Top-k* and *Top-k(x)* consistently outperform single-expert deferral methods, demonstrating both improved accuracy–cost trade-offs and strict generalization beyond $k = 1$.

In the main text, we report two-stage results on the **California Housing** dataset (Kelley Pace & Barry, 1997), **while deferring additional experiments for both settings on CIFAR-100 and SVHN to Appendix B.2**. For the one-stage setting, we provide detailed evaluations on CIFAR-10 (Krizhevsky, 2009) and SVHN (Goodfellow et al., 2013) in Appendix B.3. Evaluation metrics are formally defined in Appendices A.13 and B.1.1. We track the *expected budget* $\overline{\beta}(k) = \mathbb{E}_X[\sum_{j=1}^{k} \beta_{[j]_{\pi\downarrow}}]$ and the expected number of queried entities $\overline{k} = \mathbb{E}_X[|\Pi_k(X)|]$, where $k$ is fixed in Top-$k$ L2D and input-dependent in Top-$k(x)$ L2D. Algorithms are provided in Appendix A.4, with illustrations in Appendix A.5.

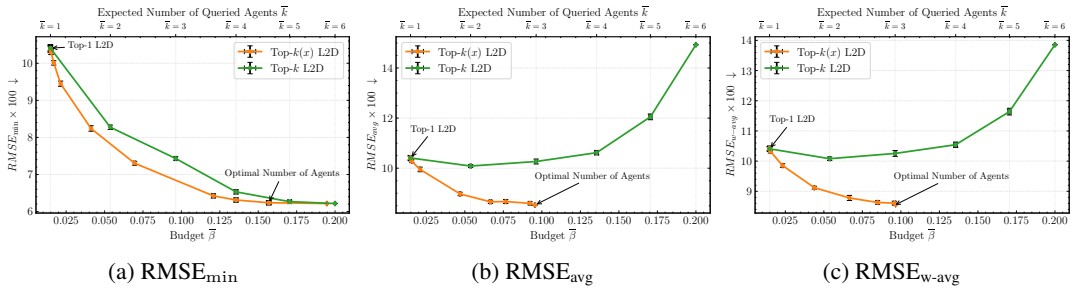

(a) RMSE$_{\text{min}}$        (b) RMSE$_{\text{avg}}$        (c) RMSE$_{\text{w-avg}}$

Figure 1: Performance of Top-$k$ and Top-$k(x)$ L2D across varying budgets $\overline{\beta}$. Each plot reports a different metric: (a) minimum RMSE, (b) uniform average RMSE, and (c) weighted average RMSE (B.1.1). Our approach outperforms the Top-1 L2D baseline (Mao et al., 2024c).

**Interpretation.** In Figure 1a, Top-$k(x)$ achieves a near-optimal RMSE of 6.23 with a budget of $\overline{\beta} = 0.156$ and an expected number of entities $\overline{k} = 4.77$, whereas Top-$k$ requires the full budget $\overline{\beta} = 0.2$ and $\overline{k} = 6$ entities to reach a comparable score (6.21). This demonstrates the ability of Top-$k(x)$ to allocate resources more efficiently by querying only the necessary entities, in contrast to Top-$k$, which tends to over-allocate costly or redundant ones. Additionally, our approach outperforms the Top-1 L2D baseline (Mao et al., 2024c), confirming the limitations of single-entity deferral.

Figures 1b and 1c evaluate Top-$k$ and Top-$k(x)$ L2D under more restrictive metrics—RMSE$_{\text{avg}}$ and RMSE$_{\text{w-avg}}$—where performance is not necessarily monotonic in the number of queried entities. In these settings, consulting too many or overly expensive entities may degrade overall performance. Top-$k(x)$ consistently outperforms Top-$k$ by carefully adjusting the number of consulted entities. In both cases, Top-$k(x)$ achieves optimal performance using a budget of just $\beta = 0.095$—a level that Top-$k$ fails to attain. For example, in Figure 1b, Top-$k(x)$ achieves an RMSE$_{\text{avg}} = 8.53$, compared to 10.08 for Top-$k$. This demonstrates that our Top-$k(x)$ L2D selectively chooses the appropriate entities—when necessary—to enhance the overall system performance.

## 7    CONCLUSION

We introduced *Top-k Learning-to-Defer*, a generalization of the two-stage L2D framework that allows deferring queries to multiple agents, and its adaptive extension, *Top-k(x) L2D*, which dynamically selects the number of consulted agents based on input complexity, consultation costs, and the agents' underlying distributions. We established rigorous theoretical guarantees, including Bayes and $(\mathcal{H}_r, \mathcal{H}_g)$-consistency, $\mathcal{H}_h$-consistency, and showed that model cascades arise as a restricted special case of our framework. Through experiments on both one-stage and two-stage regimes, we demonstrated that Top-$k$ and Top-$k(x)$ L2D consistently outperforms single-agent baselines.

## 8    REPRODUCIBILITY STATEMENT

All code, datasets, and experimental configurations are publicly released to facilitate full reproducibility. Results are reported as the mean and standard deviation over four independent runs, with

a fixed set of experts. For random baseline policies, metrics are averaged over fifty repetitions to reduce stochastic variability. All plots include error bars indicating one standard deviation. Dataset details are provided in Appendix B.1.3, while the training procedures for both the policy and the cardinality function are described in Algorithm 1 and Algorithm 2. Proofs, intermediate derivations, and explicit assumptions are included in the Appendix.

## ACKNOWLEDGMENT

This research is supported by the National Research Foundation, Singapore under its AI Singapore Programme (AISG Award No: AISG2-PhD-2023-01-041-J) and by A*STAR, and is part of the programme DesCartes which is supported by the National Research Foundation, Prime Minister's Office, Singapore under its Campus for Research Excellence and Technological Enterprise (CREATE) programme.

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

CONTENTS

# A APPENDIX

## A.1 GENERAL NOTATIONS

Table 1: Summary of main notation used throughout the paper.

| Symbol | Description |
|--------|-------------|
| $\mathcal{X}$ | Input space; $x \in \mathcal{X}$ denotes an input/query. |
| $\mathcal{Z}$ | Output space; a generic outcome variable. In multiclass classification, $\mathcal{Z} = \mathcal{Y} = \{1, \ldots, n\}$. |
| $\mathcal{Y}$ | Label space in classification, $\mathcal{Y} = \{1, \ldots, n\}$. |
| $(X, Z)$ | Random variables taking values in $\mathcal{X} \times \mathcal{Z}$. Training examples $(x, z)$ are drawn i.i.d. from $\mathcal{D}$. |
| $\mathcal{D}$ | Unknown data distribution over $\mathcal{X} \times \mathcal{Z}$. |
| $m_j, \hat{m}_j$ | Expert $j \in \{1, \ldots, J\}$, with prediction map $\hat{m}_j : \mathcal{X} \to \mathcal{Y}$ (classification) or $\mathcal{X} \to \mathcal{Z}$ (general task). |
| $g, \hat{g}$ | Base predictor in the two-stage regime, $\hat{g} : \mathcal{X} \to \mathcal{Z}$. |
| $\pi, \hat{\pi}$ | policy, $\hat{\pi} : \mathcal{X} \to \mathcal{Z}$. |
| $\mathcal{A}$ | Generic entity set; $\mathcal{A}$ denotes either $\mathcal{A}_{1s}$ or $\mathcal{A}_{2s}$ depending on the regime. |
| $\hat{a}_j$ | Entity map $\hat{a}_j : \mathcal{X} \to \mathcal{Z}$ associated with entity $j \in \mathcal{A}$ (label or expert prediction) used in the unified top-$k$ formulation. |
| $\Pi_k(x)$ | Top-$k$ selection set $\Pi_k(x) \subseteq \mathcal{A}$ of size $k$, containing the $k$ entities with the largest scores under $\pi(x, \cdot)$. |
| $[i]_{\uparrow\mu}$ | Index of the $i$-th smallest expected cost $\mu_j(x)$; used to describe the Bayes-optimal selection rule. |
| $\ell_{\text{def},k}$ | Unified top-$k$ true deferral loss $\ell_{\text{def},k}(\Pi_k(x), z) = \sum_{j \in \mathcal{A}} \mu_j(x, z) \mathbf{1}\{j \in \Pi_k(x)\}$ (uniformized top-$k$ deferral cost). |
| $\Phi_{\text{def},k}^u$ | Top-$k$ deferral surrogate based on cross-entropy family members $\Phi_{01}^u$, used for training a $k$-independent policy $\pi$. |
| $k_\theta$ | Cardinality function $k_\theta : \mathcal{X} \to \mathcal{A}_k$ used in Top-$k(x)$; parameterized by $\theta$ and learned with surrogate $\Phi_{\text{card}}$. |
| $\ell_{\text{card}}$ | Cardinality-aware deferral loss $\ell_{\text{card}}(\Pi_{\hat{k}_\theta(x)}(x), \hat{k}_\theta(x), x, z) = d(\Pi_{\hat{k}_\theta(x)}(x), x, z) + \lambda \xi\big(\sum_{i=1}^{\hat{k}_\theta(x)} \beta_{[i]\downarrow_\pi}\big)$. |
| $d$ | Metric measuring task-specific error between the selected entity set using an aggregation mechanism and the true outcome (used in $\ell_{\text{card}}$). |
| $\Phi_{\text{card}}$ | Surrogate loss for adaptive cardinality, $\Phi_{\text{card}}(\Pi_{|\mathcal{A}|}(x), k_\theta, x, z)$, built from a normalized variant $\tilde{\ell}_{\text{card}}$. |

## A.2 NOTATIONS FOR ORDERED SETS

**Definition A.1** (Orderings on a finite set). Let $\Omega = \{1, \ldots, N\}$ be a set of cardinality $N := |\Omega|$ and let
$$f : \mathcal{M} \times \Omega \longrightarrow \mathbb{R}, \qquad (m, \omega) \mapsto f(m, \omega),$$
where $\mathcal{M}$ is a measurable input space (typically $\mathcal{M} = \mathcal{X}$ or $\mathcal{M} = \mathcal{X} \times \mathcal{Y}$).

**Descending permutation.** For every fixed $m \in \mathcal{M}$, let
$$\rho_f^\downarrow(m) : \Omega \longrightarrow \Omega$$
be the (tie-broken) permutation that satisfies
$$f\big(m, \rho_f^\downarrow(m)(1)\big) \geq f\big(m, \rho_f^\downarrow(m)(2)\big) \geq \ldots \geq f\big(m, \rho_f^\downarrow(m)(N)\big).$$

The element occupying the $i$-th *largest* position is denoted by

$$[i]_f^\downarrow \;:=\; \rho_f^\downarrow(m)(i), \qquad i = 1, \ldots, N.$$

**Ascending permutation.** Analogously, define

$$\rho_f^\uparrow(m) : \Omega \;\longrightarrow\; \Omega$$

such that

$$f\big(m, \rho_f^\uparrow(m)(1)\big) \;\leq\; f\big(m, \rho_f^\uparrow(m)(2)\big) \;\leq\; \ldots \;\leq\; f\big(m, \rho_f^\uparrow(m)(N)\big),$$

and set

$$[i]_f^\uparrow \;:=\; \rho_f^\uparrow(m)(i), \qquad i = 1, \ldots, N.$$

**Top-$k$ Selection Set.** For $k \in \{1, \ldots, J+1\}$ and an order indicator $o \in \{\downarrow, \uparrow\}$, the *top-k selection set* is

$$\Pi_k(x) \;:=\; \big\{[1]_f^o, \, [1]_f^o, \ldots, [k]_f^o\big\}.$$

*Remark* 3 (Typical instantiations). In particular:

1. **Policy scores.** Take $\Omega = \mathcal{A} = \{1, \ldots, J+1\}$, $\mathcal{M} = \mathcal{X}$, and $f_\pi(x, j) := \pi(x, j)$. Descending order ($o =\downarrow$) ranks agents from most to least confident at retaining the query.

2. **Agent-specific consultation costs.** Fix $\Omega = \mathcal{A}$ but enlarge the input space to $\mathcal{M} = \mathcal{X} \times \mathcal{Z}$ and define

$$f_c\big((x, z), j\big) := c_j\big(\hat{a}_j(x), z\big).$$

   Ascending order ($o =\uparrow$) lists agents from cheapest to most expensive for the specific pair $(x, z)$.

## A.3 USEFUL DEFINITION

**Definition A.2** ($\mathcal{H}_\pi$-consistency)**.** Let $\mathcal{H}_\pi$ be a hypothesis set and let $(\pi_t)_{t \geq 1} \subset \mathcal{H}_\pi$. We say that the loss $\Phi_{def}$ is $\mathcal{H}_\pi$-consistent with respect to the loss $\ell_{def,k}$ if

$$\mathcal{E}_{\Phi_{def}}(\pi_t) - \mathcal{E}_{\Phi_{def}}^B(\mathcal{H}_\pi) + \mathcal{U}_{\Phi_{def}}(\mathcal{H}_\pi) \xrightarrow[t \to \infty]{} 0$$
$$\implies \quad \mathcal{E}_{\ell_{def,k}}(\pi_t) - \mathcal{E}_{\ell_{def,k}}^B(\mathcal{H}_\pi) + \mathcal{U}_{\ell_{def,k}}(\mathcal{H}_\pi) \xrightarrow[t \to \infty]{} 0,$$

where $\mathcal{E}_{\Phi_{def}}^B(\mathcal{H}_\pi) := \inf_{H_\pi \in H_\pi} \mathcal{E}_{\Phi_{def}}(H_\pi)$ and $\mathcal{U}_{\Phi_{def}}(\mathcal{H}_\pi)$ (resp. $\mathcal{U}_{\ell_{def,k}}(\mathcal{H}_\pi)$) denotes the minimizability gap associated with $\Phi_{def}$ (resp. $\ell_{def,k}$).

**Definition A.3** ($\mathcal{H}_\pi$-calibration)**.** Let $\mathcal{H}_\pi$ be a hypothesis set. We say that the loss $\Phi_{def}$ is $\mathcal{H}_\pi$-calibrated with respect to the loss $\ell_{def,k}$ if, for any $\epsilon > 0$, there exists $\delta > 0$ such that for all $\pi \in \mathcal{H}_\pi$ and all $x \in \mathcal{X}$,

$$\mathcal{C}_{\Phi_{def}}(\pi, x) < \mathcal{C}_{\Phi_{def}}^*(\mathcal{H}_\pi, x) + \epsilon \quad \implies \quad \mathcal{C}_{\ell_{def,k}}(\pi, x) < \mathcal{C}_{\ell_{def,k}}^B(\mathcal{H}_\pi, x) + \delta,$$

where $\mathcal{C}_{\Phi_{def}}(\pi, x)$ and $\mathcal{C}_{\ell_{def,k}}(\pi, x)$ are the conditional risks at $x$, $\mathcal{C}_{\Phi_{def}}^*(\mathcal{H}_\pi, x) :=$ $\inf_{\pi \in \mathcal{H}_\pi} \mathcal{C}_{\Phi_{def}}(\pi, x)$, and $\mathcal{C}_{\ell_{def,k}}^B(\mathcal{H}_\pi, x) := \inf_{\pi \in \mathcal{H}_\pi} \mathcal{C}_{\ell_{def,k}}(\pi, x)$.

## A.4 ALGORITHM

---

**Algorithm 1** Top-$k$ L2D Training Algorithm

---

**Input:** Dataset $\{(x_i, z_i)\}_{i=1}^I$, entities $\{\hat{a}_j : \mathcal{X} \to \mathcal{Z}\}_{j \in \mathcal{A}}$, policy $\pi \in \Pi$, number of epochs EPOCH, batch size BATCH, learning rate $\nu$.
**Initialization:** Initialize policy parameters $\theta$.
**for** $i = 1$ to EPOCH **do**
    Shuffle dataset $\{(x_i, z_i)\}_{i=1}^I$.
    **for** each mini-batch $\mathcal{B} \subset \{(x_i, z_i)\}_{i=1}^I$ of size BATCH **do**
        Extract input-output pairs $(x, z) \in \mathcal{B}$.
        Query entities $\{\hat{a}_j : \mathcal{X} \to \mathcal{Z}\}_{j \in \mathcal{A}}$.
        Compute the empirical risk minimization:
$$\tilde{\mathcal{E}}_{\Phi_{\mathrm{def},k}^u}(\pi; \theta) = \tfrac{1}{\mathrm{BATCH}} \sum_{(x,z) \in \mathcal{B}} \left[ \Phi_{\mathrm{def},k}^u(\pi, x, z) \right].$$
        Update parameters $\theta$:
$$\theta \leftarrow \theta - \nu \nabla_\theta \tilde{\mathcal{E}}_{\Phi_{\mathrm{def},k}^u}(\pi; \theta). \qquad\qquad \{\text{Gradient update}\}$$
    **end for**
**end for**
**Return:** trained policy $\pi$.

---

---

**Algorithm 2** Cardinality Training Algorithm

---

**Input:** Dataset $\{(x_i, z_i)\}_{i=1}^I$, trained policy $\pi$ from Algorithm 1, entities $\{\hat{a}_j : \mathcal{X} \to \mathcal{Z}\}_{j \in \mathcal{A}}$, cardinality function $k_\theta \in \mathcal{H}_k$, number of epochs EPOCH, batch size BATCH, learning rate $\nu$.
**Initialization:** Initialize cardinality parameters $\theta$.
**for** $i = 1$ to EPOCH **do**
    Shuffle dataset $\{(x_i, z_i)\}_{i=1}^I$.
    **for** each mini-batch $\mathcal{B} \subset \{(x_i, z_i)\}_{i=1}^I$ of size BATCH **do**
        Extract input-output pairs $(x, y) \in \mathcal{B}$.
        Query entities $\{\hat{a}_j : \mathcal{X} \to \mathcal{Z}\}_{j \in \mathcal{A}}$
        Compute the scores $\{\pi(x, j)\}_{j=1}^{|\mathcal{A}|}$ using the trained policy $\pi$.
        Sort these scores and select entries to construct the top-$k$ entity set $\Pi_{|\mathcal{A}|}(x)$.
        Compute the empirical risk minimization:
$$\tilde{\mathcal{E}}_{\Phi_{\mathrm{car}}}(k_\theta; \theta) = \tfrac{1}{\mathrm{BATCH}} \sum_{(x,y) \in \mathcal{B}} \left[ \Phi_{\mathrm{car}}(\Pi_{|\mathcal{A}|}, k_\theta, x, y) \right].$$
        Update parameters $\theta$:
$$\theta \leftarrow \theta - \nu \nabla_\theta \tilde{\mathcal{E}}_{\Phi_{\mathrm{car}}}(k_\theta; \theta). \qquad\qquad \{\text{Gradient update}\}$$
    **end for**
**end for**
**Return:** trained cardinality model $k_\theta$.

---

## A.5 ILLUSTRATION OF TOP-$k(x)$ AND TOP-$k$ L2D

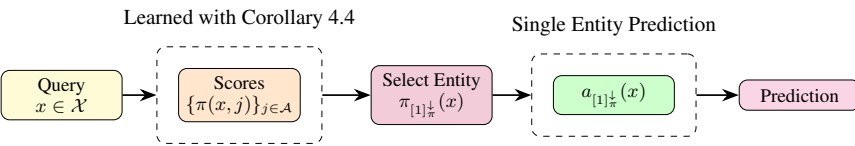

Figure 2: Inference step of Top-1 L2D (Narasimhan et al., 2022; Mao et al., 2023a; 2024c; Mozannar & Sontag, 2020; Mao et al., 2024a): Given a query, we process it through the learned policy $\pi$. We select the entity with the highest score $\hat{\pi}(x) = \arg\max_{j \in \mathcal{A}} \pi(x, j)$. Then, we query this entity and make the final prediction.

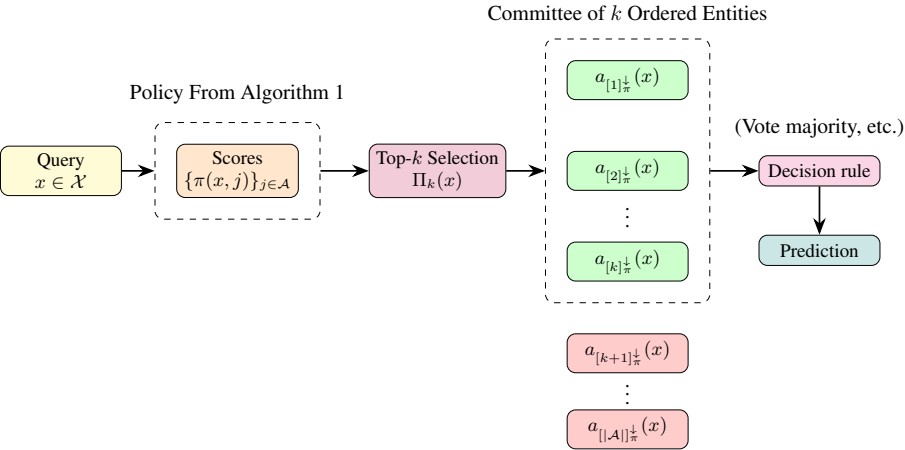

Figure 3: Inference Step of Top-$k$ L2D: Given a query $x$, we first process it through the policy learned using Algorithm 1. Based on this, we select a fixed number $k$ of entities to query, forming the *Top-k Selection Set* $\Pi_k(x)$, as defined in Definition 4.1. By construction, the expected size satisfies $\mathbb{E}_X[|\Pi_k(X)|] = k$. We then aggregate predictions from the selected top-$k$ entities using a decision rule—such as majority vote or weighted voting. The final prediction is produced by this committee according to the chosen rule.

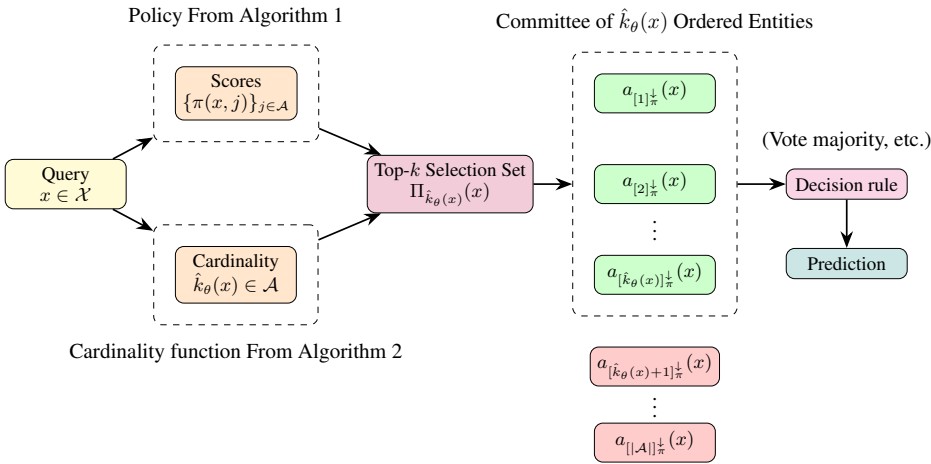

Figure 4: Inference Step of Top-$k(x)$ L2D: Given a query $x$, we process it through both the policy $\pi$, trained using Algorithm 1, and the cardinality function $k_\theta$, trained using Algorithm 2. Based on these two functions, we construct the *Top-k Selection set*. By construction, its expected size satisfies $\mathbb{E}_X[|\Pi_{\hat{k}_\theta(x)}(X)|] = \mathbb{E}_X[\hat{k}_\theta(X)]$. We then aggregate predictions from the top-$\hat{k}_\theta(x)$ entities using a decision rule (e.g., majority vote, weighted voting). The final prediction is produced by this committee of entities according to the chosen decision rule.

## A.6 MODEL CASCADES ARE SPECIAL CASES OF TOP-$k$ AND TOP-$k(x)$ SELECTION

Throughout, let $\mathcal{A}$ be the set of entities. For $j \in \mathcal{A}$ we denote by $\hat{a}_j : \mathcal{X} \to \mathcal{Z}$ the prediction of entity $j$, by $\pi : \mathcal{X} \times \mathcal{A} \to \mathbb{R}$ a *policy score*, and by

$$\Pi_k(x) = \{\pi_{[1]_\pi^\downarrow}(x), \ldots, \pi_{[k]_\pi^\downarrow}(x)\}$$

the *Top-k Selection Set* containing the indices of the $k$ largest scores.

### A.6.1 MODEL CASCADES

**Definition A.4** (Evaluation order and thresholds). Fix a permutation $\rho = (\rho_1, \ldots, \rho_{|\mathcal{A}|})$ of $\mathcal{A}$ (the *evaluation order*) and confidence thresholds $0 < \nu_1 < \nu_2 < \cdots < \nu_{|\mathcal{A}|} < 1$. For each entity let $\mathrm{conf} : \mathcal{X} \times \mathcal{A} \to [0, 1]$ be a confidence measure.

**Definition A.5** (Size–$k$ cascade allocation). For a fixed $k \in \{1, \ldots, |\mathcal{A}|\}$ define the *cascade set*
$$\mathcal{K}_k(x) := \{\rho_1, \ldots, \rho_k\}.$$
The cascade evaluates the entities in the order $\rho$ until it reaches $\rho_k$. If the confidence test $\mathrm{conf}(\rho_k, x) \geq \nu_k$ is satisfied, the cascade *allocates* the set $\mathcal{K}_k(x)$; otherwise it proceeds to the next stage (see Section A.6.3 for the adaptive case).

### A.6.2 EMBEDDING A FIXED-$k$ CASCADE

**Lemma A.6** (Score construction). *For a fixed $k$ define*
$$\pi_k(x, j) := \begin{cases} 2 - \dfrac{\mathrm{rank}_{\mathcal{K}_k(x)}(j)}{k + 1}, & j \in \mathcal{K}_k(x), \\[2mm] -\dfrac{\mathrm{rank}_{\mathcal{A} \setminus \mathcal{K}_k(x)}(j)}{|\mathcal{A}| + 1}, & j \notin \mathcal{K}_k(x), \end{cases}$$
*where $\mathrm{rank}_B(j) \in \{1, \ldots, |B|\}$ is the index of $j$ inside the list $B$ ordered according to $\rho$. Then for every $x \in \mathcal{X}$*
$$\Pi_k(x) = \mathcal{K}_k(x).$$

*Proof.* **Separation.** Scores assigned to $\mathcal{K}_k(x)$ lie in $(1, 2]$, while scores assigned to $\mathcal{A} \setminus \mathcal{K}_k(x)$ lie in $[-1, -\frac{1}{|\mathcal{A}|+1})$; hence all $k$ largest scores belong exactly to $\mathcal{K}_k(x)$.

**Distinctness.** Within each block, consecutive ranks differ by $1/(k+1)$ or $1/(|\mathcal{A}|+1)$, so ties cannot occur. Therefore the permutation returns precisely the indices of $\mathcal{K}_k(x)$ in decreasing order, and the Top-$k$ Selection Set equals $\mathcal{K}_k(x)$. $\square$

**Corollary A.7** (Cascade embedding for any fixed $k$). *For every $k \in \{1, \ldots, |\mathcal{A}|\}$ the policy $\pi_k$ of Lemma A.6 satisfies*
$$\Pi_k(x) = \mathcal{K}_k(x) \quad \forall x \in \mathcal{X}.$$
*Consequently, the Top-$k$ Selection coincides exactly with the size-$k$ cascade allocation.*

*Proof.* Immediate from Lemma A.6. $\square$

### A.6.3 EMBEDDING ADAPTIVE (EARLY-EXIT) CASCADES

Let the cascade stop after a *data-dependent* number of stages $\hat{k}_\theta(x) \in \{1, \ldots, |\mathcal{A}|\}$. Define the cardinality function $\hat{k}_\theta(x)$ and reuse the score construction of Lemma A.6 with $k$ replaced by $\hat{k}_\theta(x)$: $\pi_{\hat{k}_\theta(x)}(x, )$.

**Lemma A.8** (Cascade embedding for adaptive cardinality). *With policy $\pi_{\hat{k}_\theta(x)}$ and cardinality function $\hat{k}_\theta(x)$, the Top-$k(x)$ Selection pipeline allocates $\mathcal{K}_{\hat{k}_\theta(x)}(x)$ for every input $x$. Therefore any adaptive (early-exit) model cascade is a special case of Top-$k(x)$ Selection.*

*Proof.* Applying Lemma A.6 with $k = \hat{k}_\theta(x)$ yields $\Pi_{\hat{k}_\theta(x)}(x) = \mathcal{K}_{\hat{k}_\theta(x)}(x)$. The cardinality function truncates the full Top-$k(x)$ set to its first $\hat{k}_\theta(x)$ elements—precisely $\mathcal{K}_{\hat{k}_\theta(x)}(x)$. $\square$

### A.6.4 EXPRESSIVENESS: MODEL CASCADES VS. TOP-$k$ / TOP-$k(x)$ SELECTION

**Hierarchy.** Every model cascade can be realised by a suitable choice of policy scores and, for the adaptive case, a cardinality function (see App. A.6). Hence
$$\underbrace{\text{Model Cascades}}_{\text{prefix of a fixed order}} \subset \underbrace{\text{Top-}k\text{ Selection}}_{\text{constant } k} \subset \underbrace{\text{Top-}k(x)\text{ Selection}}_{\text{learned } k(x)}.$$
The inclusion is *strict*, for the reasons detailed below.

**Why the inclusion is strict.**

1. **Non-contiguous selection.** A Top-$k$ Selection Set $\Pi_k(x)$ may pick any subset of size $k$ (e.g. $\{1, 2, 5\}$), whereas a cascade always selects a *prefix* $\{\rho_1, \dots, \rho_k\}$ of the evaluation order.

2. **Learned cardinality.** In Top-$k(x)$ Selection the cardinality function $\hat{k}_\theta(x)$ is trained by minimizing a surrogate risk; Theorem 4.7 provides consistency and ensures the optimality of $\Pi_{k(x)}(x)$. Classical cascades, by contrast, rely on fixed confidence thresholds with no statistical guarantee.

3. **Cost-aware ordering.** Lemma 4.5 shows the Bayes-optimal policy orders entities by *expected cost*, which may vary with $x$. Top-$k$ policies can realize such input-dependent orderings by means of the policy scores $\pi(x, j)$. Cascades, in contrast, impose a single, input-independent order $\rho$.

4. **Multi-entity aggregation.** After selecting $k$ entities, Top-$k$ Selection can aggregate their predictions (majority vote, weighted vote, averaging, *etc.*). A cascade, however, *uses only the last entity in the prefix whose confidence test is passed*. Earlier entities are effectively discarded. Thus cascades cannot implement multi-entity aggregation rules.

**Separating example.** Assume $|\mathcal{A}| = 3$ with entities $a_1, a_2, a_3$ and consider a Top-2 Selection policy defined by policy scores $\pi(x, )$ such that

$$\text{on some } x: \quad \pi(x, 1) > \pi(x, 3) > \pi(x, 2) \quad \Rightarrow \quad \Pi_2(x) = \{1, 3\},$$

$$\text{on some } x': \quad \pi(x', 1) > \pi(x', 2) > \pi(x', 3) \quad \Rightarrow \quad \Pi_2(x') = \{1, 2\}.$$

Suppose, for contradiction, that a *cascade with a fixed, input-independent order $\rho$* realizes the same selections as Top-2. Because $\Pi_2(x) = \{1, 3\}$ is not a prefix of any order unless 3 precedes 2, we must have $\rho$ satisfying

$$1 \succ_\rho 3 \succ_\rho 2.$$

But since $\Pi_2(x') = \{1, 2\}$ must also be a prefix of the *same $\rho$*, we must have

$$1 \succ_\rho 2 \succ_\rho 3,$$

a contradiction. Hence no fixed-order cascade can realize this Top-2 Selection Set.

Moreover, even in cases where a Top-$k$ set *is* a prefix (e.g., $\{1, 2\}$), a cascade outputs the prediction of the *last* confident entity in that prefix, whereas Top-$k$ Selection may aggregate the $k$ entities' predictions (e.g., by a weighted vote). Therefore, cascades cannot, in general, implement Top-$k$ aggregation rules.

## A.7 PROOF LEMMA 4.2

**Definition 4.2** (Top-$k$ True Deferral Loss). Let $x \in \mathcal{X}$, $z \in \mathcal{Z}$, and $\Pi_k(x) \subseteq \mathcal{A}$ be the top-$k$ selection set. Let $\mu_j(x, z)$ the cost of selecting entity $j$ for input $(x, z)$. The uniformized top-$k$ true deferral loss is

$$\ell_{\mathrm{def}, k}(\Pi_k(x), z) = \sum_{j=1}^{|\mathcal{A}|} \mu_j(x, z) \mathbf{1}\{j \in \Pi_k(x)\},$$

*Proof.* We will prove this novel true deferral loss for both the one-stage and two-stage regime.

**Two-Stage.** In the standard L2D setting (see 3.2), the deferral loss utilizes the indicator function $\mathbf{1}\{\hat{\pi}(x) = j\}$ to select the most cost-efficient entity in the system. Specifically, we have $\pi = r$, $|\mathcal{A}^{2s}| = J + 1$, and $\mu_j(x, z) = c_j(x, z)$. We can upper-bound the standard two-stage L2D loss by

employing the indicator function over the Top-$k$ Selection Set $\Pi_k(x)$ defined in 4.1:

$$
\begin{aligned}
\ell_{\text{def}}^{2s}(\hat{\pi}(x), z) &= \sum_{j=1}^{J+1} c_j(x, z) \mathbf{1}\{\hat{\pi}(x) = j\} \\
&= \sum_{j=1}^{J+1} \mu_j(x, z) \mathbf{1}\{\hat{\pi}(x) = j\} \\
&\leq \sum_{j=1}^{J+1} \mu_j(x, z) \mathbf{1}\{j \in \Pi_k(x)\} \\
&= \ell_{\text{def},k}^{2s}(\Pi_k(x), z).
\end{aligned}
\tag{5}
$$

Consider a system with two experts $\{m_1, m_2\}$ and one main predictor $g$. Leading to $\mathcal{A} = \{1, 2, 3\}$, and the Top-$|\mathcal{A}|$ Selection Set $\Pi_{|\mathcal{A}|}(x) = \{3, 2, 1\}$. This indicates that expert $m_2$ has a higher confidence score than expert $m_1$ and predictor $g$, i.e., $\pi(x, 3) \geq \pi(x, 2) \geq \pi(x, 1)$. We evaluate $\ell_{\text{def},k}^{2s}$ for different values of $k \leq |\mathcal{A}|$:

**For $k = 1$:** The Selection Set is $\Pi_1(x) = \{3\}$, which corresponds to the standard L2D setting where deferral is made to the most confident entity (Narasimhan et al., 2022; Mao et al., 2023a; Montreuil et al., 2025b). Thus,

$$
\ell_{\text{def},1}^{2s}(\Pi_1(x), z) = \mu_3(x, z) = \alpha_3 \psi(\hat{m}_2(x), z) + \beta_3,
\tag{6}
$$

recovering the same result as $\ell_{\text{def}}^{2s}$ defined in 3.2.

**For $k = 2$:** The Selection Set expands to $\Pi_2(x) = \{3, 2\}$, implying that both expert $m_2$ and expert $m_1$ are queried. Therefore,

$$
\ell_{\text{def},2}^{2s}(\Pi_2(x), z) = \mu_3(x, z) + \mu_2(x, z),
\tag{7}
$$

correctly reflecting the computation of costs from the queried entities.

**For $k = 3$:** The Selection Set further extends to $\Pi_3(x) = \{3, 2, 1\}$, implying that all entities in the system are queried. Consequently,

$$
\ell_{\text{def},3}^{2s}(\Pi_3(x), z) = \mu_3(x, z) + \mu_2(x, z) + \mu_1(x, z),
\tag{8}
$$

incorporating the costs from all entities in the system.

**One-Stage.** The standard One-Stage deferral loss introduced by Mozannar & Sontag (2020) assigns cost based on whether the model predicts or defers:

$$
\ell_{\text{def}}^{1s}(\hat{h}(x), y) = \mathbf{1}\{\hat{h}(x) \neq y\} \mathbf{1}\{\hat{h}(x) \leq n\} + \sum_{j=1}^{J} c_j(x, y) \mathbf{1}\{\hat{h}(x) = n + j\},
$$

This formulation handles two mutually exclusive cases: the model predicts a class label $j \in \{1, \ldots, n\}$ and is penalized if $j \neq y$, or it defers to expert $m_j$ and incurs the expert-specific cost $c_j(x, y)$. However, this formulation relies on a hard-coded distinction between prediction and deferral.

To generalize and simplify the analysis, we introduce a unified cost-sensitive reformulation over the entire entity set $\mathcal{A} = \{1, \ldots, n + J\}$. We define

$$
\mu_j(x, y) = \begin{cases} \alpha_j \mathbf{1}\{j \neq y\} + \beta_j & \text{for } j \leq n, \\ \alpha_j \mathbf{1}\{\hat{m}_{j-n}(x) \neq y\} + \beta_j & \text{for } j > n. \end{cases}
$$

This assigns each entity—whether label or expert—a structured cost combining prediction error and fixed usage cost. The total loss is then

$$
\ell_{\text{def}}^{2s}(\hat{h}(x), y) = \sum_{j=1}^{n+J} \mu_j(x, y) \mathbf{1}\{\hat{h}(x) = j\}.
$$

We now verify that this general formulation is equivalent to the original loss when the cost parameters are selected appropriately.

Consider a binary classification example with $\mathcal{Y} = \{1, 2\}$, two experts $\{m_1, m_2\}$, and parameters $\alpha_j = 1$, $\beta_j = 0$ for all $j$. If the classifier $h$ predicts label $\hat{h}(x) = 1$, then the cost is $\mu_1(x, y) = \mathbf{1}\{1 \neq y\}$, which matches the original unit penalty for incorrect prediction. If $\hat{h}(x) = y$, the cost becomes $\mu_y(x, y) = \mathbf{1}\{y \neq y\} = 0$, correctly yielding no penalty for correct prediction. If instead the classifier $h$ defers to expert $m_1$, i.e., $\hat{h}(x) = n + 1 = 3$, then the loss becomes $\mu_3(x, y) = \mathbf{1}\{m_1(x) \neq y\}$, matching the original expert cost.

Therefore, by using $\pi = h$ and $|\mathcal{A}^{1s}| = J + n$, it follows:

$$
\begin{aligned}
\ell_{\text{def}}^{1s}(\hat{h}(x), y) &= \sum_{j=1}^{J+n} \mu_j(x, y) \mathbf{1}\{\hat{h}(x) = j\} \\
&\leq \sum_{j=1}^{J+n} \mu_j(x, y) \mathbf{1}\{j \in \Pi_k(x)\} \\
&= \ell_{\text{def}, k}^{1s}(\Pi_k(x), y).
\end{aligned}
\tag{9}
$$

$\square$

### A.8 PROOF LEMMA 4.3

**Lemma 4.3** (Upper Bound on the Top-$k$ Deferral Loss). *Let $x \in \mathcal{X}$, $z \in \mathcal{Z}$, and let $1 \leq k \leq |\mathcal{A}|$. Let $\Phi_{01}^u$ a convex surrogate in the cross-entropy family. Then the top-$k$ deferral loss satisfies*

$$
\ell_{def, k}(\Pi_k(x), z) \leq \sum_{j \in \mathcal{A}} \left( \sum_{i \neq j} \mu_i(x, z) \right) \Phi_{01}^u(\pi, x, j) - (|\mathcal{A}| - 1 - k) \sum_{j \in \mathcal{A}} \mu_j(x, z),
$$

*Proof.* Let the entity set $\mathcal{A}$ and the policy $\pi \in \mathcal{H}_\pi$. For a query–label pair $(x, z)$ denote the costs of allocating to an entity $j$ by $\mu_j(x, z) \geq 0$ $(j = 1, \ldots, |\mathcal{A}|)$ and the total cost by

$$
C_{\text{tot}}(x, z) = \sum_{j=1}^{|\mathcal{A}|} \mu_j(x, z).
$$

Define, for each index $j$,

$$
\xi_j(x, z) = \sum_{\substack{q=1 \\ q \neq j}}^{|\mathcal{A}|} \mu_q(x, z) = C_{\text{tot}}(x, z) - \mu_j(x, z).
$$

For any $k \in \{1, \ldots, |\mathcal{A}|\}$ and any size-$k$ decision set $\Pi_k(x) \subseteq \{1, \ldots, |\mathcal{A}|\}$ the top-$k$ deferral loss is

$$
\ell_{\text{def}, k}\big(\Pi_k(x), z\big) = \sum_{j=1}^{|\mathcal{A}|} \mu_j(x, z) \mathbf{1}\{j \in \Pi_k(x)\}
$$

Because $\Pi_k(x)$ and its complement $\overline{\Pi}_k(x)$ form a disjoint partition of $\{1, \ldots, |\mathcal{A}|\}$,

$$
\ell_{\text{def}, k}(\Pi_k(x), z) = \sum_{j \in \Pi_k} \mu_j = C_{\text{tot}} - \sum_{j \in \overline{\Pi}_k} \mu_j.
\tag{10}
$$

For every $j$ we have $\mu_j = C_{\text{tot}} - \xi_j$ with $\xi_j = \sum_{i \neq j} \mu_i$, whence

$$
\sum_{j \in \overline{\Pi}_k} \mu_j = \sum_{j \in \overline{\Pi}_k} (C_{\text{tot}} - \xi_j) = (|\mathcal{A}| - k) C_{\text{tot}} - \sum_{j \in \overline{\Pi}_k} \xi_j,
\tag{11}
$$

with the factor $|\mathcal{A}| - k$ being the cardinality of $\overline{\overline{\Pi}}_k$. Substituting equation 11 into equation 10 yields

$$\ell_{\mathrm{def},k}(\Pi_k(x), z) = C_{\mathrm{tot}} - \left[ (|\mathcal{A}| - k)C_{\mathrm{tot}} - \sum_{j \in \overline{\overline{\Pi}}_k} \xi_j \right] \tag{12}$$

$$= \sum_{j=1}^{|\mathcal{A}|} \xi_j \mathbf{1}\{j \notin \Pi_k\} - (|\mathcal{A}| - k - 1) \sum_{j=1}^{|\mathcal{A}|} \mu_j(x, z). \tag{13}$$

Let us inspect limit cases:

1. $k = 1$. Then $\overline{\overline{\Pi}}_k$ has $|\mathcal{A}| - 1$ indices and the constant term reduces to $-(|\mathcal{A}| - 2)C_{\mathrm{tot}}$; expanding the sum shows $\ell_{\mathrm{def},1} = \mu_{\hat{\pi}(x)}$ as expected for the classical true deferral loss defined in 3.1 and 3.2.

2. $k = |\mathcal{A}|$. The complement is empty, $\sum_{j \notin \Pi_{|\mathcal{A}|}} \xi_j = 0$ and $|\mathcal{A}| - k - 1 = -1$, so the formula gives $\ell_{\mathrm{def},|\mathcal{A}|} = C_{\mathrm{tot}}$, i.e. paying *all* deferral costs — again matching intuition.

Finally, Let $\Phi_{01}^u(\pi, x, j)$ be a multiclass surrogate that satisfies $\mathbf{1}\{j \notin \Pi_k(x)\} \leq \Phi_{01}^u(\pi, x, j)$ for every $j$. As shown by Lapin et al. (2016); Yang & Koyejo (2020); Cortes et al. (2024) the cross-entropy family satisfy this condition. Because each weight $\xi_j(x, z) \geq 0$, we have

$$\ell_{\mathrm{def},k}(\Pi_k, x, z) \leq \sum_{j=1}^{|\mathcal{A}|} \xi_j(x, z) \Phi_{01}^u(\pi, x, j) - (|\mathcal{A}| - k - 1) \sum_{j=1}^{|\mathcal{A}|} \mu_j(x, z)$$

$$= \sum_{j=1}^{|\mathcal{A}|} \left( \sum_{i \neq j} \mu_i(x, z) \right) \Phi_{01}^u(\pi, x, j) - (|\mathcal{A}| - k - 1) \sum_{j=1}^{|\mathcal{A}|} \mu_j(x, z) \tag{14}$$

We have shown the desired relationship. $\qquad \square$

## A.9 PROOF LEMMA 4.5

**Lemma 4.5** (Bayes-Optimal Top-$k$ Selection)**.** *Let $x \in \mathcal{X}$. For each entity $j \in \mathcal{A}$, define the expected cost $\overline{\mu}_j(x) = \mathbb{E}_{Z|X=x}[\mu_j(x, Z)]$, its Bayes-optimal expected cost as $\overline{\mu}_j^B(x) = \inf_{g \in \mathcal{H}_g} \overline{\mu}_j(x)$. Then the Bayes-optimal top-$k$ selection set is*

$$\Pi_k^B(x) = \arg \min_{\substack{\Pi_k \subseteq \mathcal{A} \\ |\Pi_k| = k}} \sum_{j \in \Pi_k} \overline{\mu}_j^B(x) = \{[1]_{\overline{\mu}^B}^\uparrow, [2]_{\overline{\mu}^B}^\uparrow, \ldots, [k]_{\overline{\mu}^B}^\uparrow\},$$

*where $[i]_{\overline{\mu}^B}^\uparrow$ denotes the index of the $i$-th smallest expected cost in $\{\overline{\mu}_j^B(x) : j \in \mathcal{A}\}$. In the one-stage regime, where no base predictor class $\mathcal{H}_g$ is defined, we simply set $\overline{\mu}_j^B(x) = \overline{\mu}_j(x)$.*

*Proof.* Let's consider the Top-$k$ Deferral Loss defined by

$$\ell_{\mathrm{def},k}(\Pi_k(x), z) = \sum_{j=1}^{|\mathcal{A}|} \mu_j(x, z) \mathbf{1}\{j \in \Pi_k(x)\},$$

where $\mu_j(x, z) = \alpha_j \psi(\hat{a}_j(x), z) + \beta_j$ in the two-stage and $\mu_j(x, z) = \alpha_j \mathbf{1}\{\hat{a}_j(x) \neq y\} + \beta_j$ in one-stage setting, is the cost associated with entity $j \in \mathcal{A}$. We define the expected cost as:

$$\overline{\mu}_j(x) = \mathbb{E}_{Z|X=x}[\mu_j(x, Z)]$$

Given the policy $\pi : \mathcal{X} \to \mathcal{A}$, we have the Top-$k$ Selection Set $\Pi_k(x) \subseteq \mathcal{A}$.

**One-stage.** Here $\mu_j(x, y) = \alpha_j \mathbf{1}\{\hat{a}_j(x) \neq y\} + \beta_j$ and $\hat{a}_j$ are fixed (non-optimizable) as they are labels or experts. Thus

$$\overline{\mu}_j(x) = \alpha_j \mathbb{P}(Y \neq \hat{a}_j(x) \mid X = x) + \beta_j$$

$$= \begin{cases} \alpha_j \mathbb{P}(Y \neq j \mid X = x) + \beta_j & \text{if } j \leq n \\ \alpha_j \mathbb{P}(Y \neq \hat{m}_{j-n}(x) \mid X = x) + \beta_j & \text{if } j > n \end{cases}$$

is independent of $\pi$ (or $h$ here). We introduce the conditional risk (Steinwart, 2007; Bartlett et al., 2006) of the Top-$k$ Deferral Loss:

$$\mathcal{C}_{\ell_{\text{def},k}}(\pi, x) = \mathbb{E}_{Y|X=x}\Big[\ell_{\text{def},k}(\Pi_k(x), Y)\Big]$$

$$= \sum_{j=1}^{|\mathcal{A}|} \overline{\mu}_j(x)\mathbf{1}\{j \in \Pi_k(x)\}$$

Hence the Bayes (conditional) risk over policies reduces to choosing a size-$k$ subset minimizing the sum of these expected costs:

$$\mathcal{C}^B_{\ell_{\text{def},k}}(\mathcal{H}_\pi, x) = \inf_{\pi \in \mathcal{H}_\pi} \mathcal{C}_{\ell_{\text{def},k}}(\pi, x)$$

$$= \inf_{\pi \in \mathcal{H}_\pi} \sum_{j=1}^{|\mathcal{A}|} \overline{\mu}_j(x)\mathbf{1}\{j \in \Pi_k(x)\} \tag{15}$$

Let $[i]^{\uparrow}_{\overline{\mu}}$ denote the index of the $i$-th smallest expected cost, so that

$$\overline{\mu}_{[1]^{\uparrow}_{\overline{\mu}}}(x, y) \leq \overline{\mu}_{[2]^{\uparrow}_{\overline{\mu}}}(x, y) \leq \cdots \leq \overline{\mu}_{[n+J]^{\uparrow}_{\overline{\mu}}}(x, y).$$

Then the Bayes-optimal risk is obtained by selecting the $k$ entities with the lowest expected costs:

$$\mathcal{C}^B_{\ell_{\text{def},k}}(\mathcal{H}_\pi, x) = \sum_{i=1}^{k} \overline{\mu}_{[i]^{\uparrow}_{\overline{\mu}}}(x, y).$$

Consequently, the Top-$k$ Selection Set $\Pi^B_k(x)$ that achieves this minimum is

$$\Pi^B_k(x) = \underset{\substack{\Pi_k(x) \subseteq \mathcal{A} \\ |\Pi_k(x)|=k}}{\arg\min} \sum_{j \in \Pi_k(x)} \overline{\mu}^B_j(x) = \{[1]^{\uparrow}_{\overline{\mu}^B}, [2]^{\uparrow}_{\overline{\mu}^B}, \ldots, [k]^{\uparrow}_{\overline{\mu}^B}\}, \tag{16}$$

meaning $\Pi^B_k(x)$ selects the $k$ entities with the lowest optimal expected costs.

**Two-Stage.** Here $\mu_j(x, z) = \alpha_j \psi(\hat{a}_j(x), z) + \beta_j$ and $\hat{a}_j$ are fixed but we have the full control of the predictor $g \in \mathcal{H}_g$. Thus

$$\overline{\mu}_j(x) = \alpha_j \mathbb{E}_{Z|X=x}[\psi(\hat{a}_j(x), Z)] + \beta_j$$

$$= \begin{cases} \alpha_j \mathbb{E}_{Z|X=x}[\psi(\hat{g}(x), Z)] + \beta_j & \text{if } j = 1 \\ \alpha_j \mathbb{E}_{Z|X=x}[\psi(\hat{m}_{j-1}(x), Z)] + \beta_j & \text{if } j > 1 \end{cases}$$

is independent of $\pi$ (or $r$ here) but not $g \in \mathcal{H}_g$ for $\mu_1$. We introduce the conditional risk (Steinwart, 2007; Bartlett et al., 2006) of the Top-$k$ Deferral Loss:

$$\mathcal{C}_{\ell_{\text{def},k}}(\pi, g, x) = \mathbb{E}_{Z|X=x}\Big[\ell_{\text{def},k}(\Pi_k(x), Z)\Big]$$

$$= \sum_{j=1}^{|\mathcal{A}|} \overline{\mu}_j(x)\mathbf{1}\{j \in \Pi_k(x)\}$$

Hence the Bayes (conditional) risk over policies reduces to choosing a size-$k$ subset minimizing the sum of these expected costs:

$$\mathcal{C}^B_{\ell_{\text{def},k}}(\mathcal{H}_\pi, \mathcal{H}_g, x) = \inf_{\pi \in \mathcal{H}_\pi} \inf_{g \in \mathcal{H}_g} \mathcal{C}_{\ell_{\text{def},k}}(\pi, g, x)$$

$$= \inf_{\pi \in \mathcal{H}_\pi} \sum_{j=1}^{|\mathcal{A}|} \overline{\mu}^B_j(x)\mathbf{1}\{j \in \Pi_k(x)\} \tag{17}$$

with $\overline{\mu}^B_1(x) = \inf_{g \in \mathcal{H}_g} \overline{\mu}_1(x)$ and for $j > 1$, $\overline{\mu}^B_j(x) = \overline{\mu}_j(x)$. Let $[i]^{\uparrow}_{\overline{\mu}}$ denote the index of the $i$-th smallest expected cost, so that

$$\overline{\mu^B}_{[1]^{\uparrow}_{\overline{\mu}^B}}(x, z) \leq \overline{\mu^B}_{[2]^{\uparrow}_{\overline{\mu}^B}}(x, z) \leq \cdots \leq \overline{\mu^B}_{[J+1]^{\uparrow}_{\overline{\mu}^B}}(x, z).$$

Then the Bayes-optimal risk is obtained by selecting the $k$ entities with the lowest expected costs:

$$\mathcal{C}^B_{\ell_{\mathrm{def},k}}(\mathcal{H}_\pi, \mathcal{H}_g, x) = \sum_{i=1}^k \overline{\mu}^B_{[i]^\uparrow_{\overline{\mu}^B}}(x, z).$$

Consequently, the Top-$k$ Selection Set $\Pi^B_k(x)$ that achieves this minimum is

$$\Pi^B_k(x) = \operatorname*{arg\,min}_{\substack{\Pi_k(x) \subseteq \mathcal{A} \\ |\Pi_k(x)|=k}} \sum_{j \in \Pi_k(x)} \overline{\mu}_j(x) = \{[1]^\uparrow_{\overline{\mu}}, [2]^\uparrow_{\overline{\mu}}, \ldots, [k]^\uparrow_{\overline{\mu}}\}, \tag{18}$$

meaning $\Pi^B_k(x)$ selects the $k$ entities with the lowest optimal expected costs.

$\square$

## A.10 PROOF COROLLARY 4.6

**Corollary 4.6** (Special cases for $k = 1$). *The Bayes rule in Lemma 4.5 recovers prior Top-1 results:*

1. ***One-stage L2D.*** *For any entity $j$ (labels $j \le n$ and experts $j > n$),*

$$\overline{\mu}^B_j(x) = \alpha_j \mathbb{P}\big(\hat{a}_j(x) \ne Y \big| X = x\big) + \beta_j,$$

*which yields the Top-1 Bayes policy of Mozannar & Sontag (2020).*

2. ***Two-stage L2D.*** *Let $j = 1$ denote the base predictor and $j \ge 2$ the experts. Then*

$$\overline{\mu}^B_1(x) = \alpha_1 \inf_{g \in \mathcal{H}_g} \mathbb{E}_{Z|X=x}\big[\psi\big(\hat{g}(x), Z\big)\big] + \beta_1,$$

$$\text{and for } j \ge 2, \ \overline{\mu}^B_j(x) = \alpha_j \mathbb{E}_{Z|X=x}\big[\psi\big(\hat{m}_{j-1}(x), Z\big)\big] + \beta_j,$$

*recovering the Top-1 allocation in Narasimhan et al. (2022); Mao et al. (2023a); Montreuil et al. (2025b).*

3. ***Selective prediction (reject option).*** *We take the set of label entities and augment it with an abstain entity $\perp$, defined by $\alpha_\perp = 0$ and $\beta_\perp = \lambda > 0$, while label entities use $\alpha_j = 1, \beta_j = 0$. Then*

$$\overline{\mu}^B_j(x) = \mathbb{P}\big(j \ne Y \big| X = x\big) \quad (j \in \{1, \ldots, n\}), \qquad \overline{\mu}^B_\perp(x) = \lambda,$$

*yielding the Chow's rule (Chow, 1970).*

*Proof of Corollary 4.6.* Set $k = 1$ in Lemma 4.5. Then the Bayes rule selects the single index

$$\Pi^B_1(x) = \big\{[1]^\uparrow_{\overline{\mu}^B}\big\} = \Big\{ \operatorname*{arg\,min}_{j \in \mathcal{A}} \overline{\mu}^B_j(x) \Big\},$$

i.e., the entity with the smallest Bayes-optimized conditional expected cost at $x$. We verify the three specializations.

**(1) One-stage L2D.** In one-stage, the entities (labels or fixed experts) do not depend on any $g$, and

$$\mu_j(x, y) = \alpha_j \mathbf{1}\{\hat{a}_j(x) \ne y\} + \beta_j \implies \overline{\mu}^B_j(x) = \overline{\mu}_j(x) = \alpha_j \mathbb{P}\big(\hat{a}_j(x) \ne Y \big| X = x\big) + \beta_j.$$

Thus, $\Pi^B_1(x) = \{\arg\min_j \overline{\mu}_j(x)\}$ selects the entity with the lowest expected cost, which in the one-stage case corresponds to choosing the label or expert with the lowest misclassification probability. This recovers exactly the Bayes-optimal Top-1 policy established in prior one-stage L2D work (Mozannar & Sontag, 2020; Mao et al., 2024a).

**(2) Two-stage L2D.** Let $j = 1$ denote the fixed base predictor entity $a_1(x) = \hat{g}(x)$, and $j \ge 2$ denote (fixed) experts $\hat{m}_{j-1}(x)$. Then

$$\overline{\mu}^B_1(x) = \inf_{g \in \mathcal{H}_g} \mathbb{E}_{Z|X=x}\big[\alpha_1 \psi\big(\hat{g}(x), Z\big) + \beta_1\big] = \alpha_1 \inf_{g \in \mathcal{H}_g} \mathbb{E}_{Z|X=x}\big[\psi\big(\hat{g}(x), Z\big)\big] + \beta_1,$$

while for $j \geq 2$ (no $g$-dependence)

$$\overline{\mu}_j^B(x) = \overline{\mu}_j(x) = \alpha_j \mathbb{E}_{Z|X=x}\left[\psi\big(\hat{m}_{j-1}(x), Z\big)\right] + \beta_j.$$

Hence $\Pi_1^B(x) = \{\arg\min_{j \in \mathcal{A}} \overline{\mu}_j^B(x)\}$ selects, among the base predictor and the experts, the single entity with the smallest Bayes-optimized expected cost, which recovers the standard Top-1 allocation in two-stage L2D (e.g., Narasimhan et al.; Mao et al.; Montreuil et al.).

**(3) Selective prediction (reject option).** Let the action set consist of the $n$ label entities and a reject action $\perp$. Set $\alpha_j = 1, \beta_j = 0$ for labels $j \in \{1, \dots, n\}$, and $\alpha_\perp = 0, \beta_\perp = \lambda > 0$. Write $p_j(x) := \mathbb{P}(Y = j \mid X = x)$. Then

$$\overline{\mu}_j^B(x) = \mathbb{P}(j \neq Y \mid X = x) = 1 - p_j(x), \qquad \overline{\mu}_\perp^B(x) = \lambda.$$

Therefore

$$\min\left\{ \min_{1 \leq j \leq n} \big(1 - p_j(x)\big), \; \lambda \right\} = \min\left\{1 - \max_{1 \leq j \leq n} p_j(x), \; \lambda\right\}.$$

Equivalently, predict the most probable class $j^\star(x) \in \arg\max_j p_j(x)$ if $1 - p_{j^\star}(x) \leq \lambda$ (i.e., $p_{j^\star}(x) \geq 1 - \lambda$), and abstain otherwise. This is precisely Chow's rule (Chow, 1970; Geifman & El-Yaniv, 2017; Cortes et al., 2016). $\qquad\square$

### A.11  PROOF THEOREM 4.7

First, we prove an intermediate Lemma.

**Lemma A.9** (Consistency of a Top-$k$ Loss). *sample A surrogate loss function $\Phi_{01}^u$ is said to be $\mathcal{H}_\pi$-consistent with respect to the top-k loss $\ell_k(\Pi_k(x), j) = \mathbf{1}\{j \in \Pi_k(x)\}$ if, for any $\pi \in \mathcal{H}_\pi$, there exists a non-decreasing, and non-negative, concave function $\Gamma_u^{-1} : \mathbb{R}^+ \to \mathbb{R}^+$ such that:*

$$\sum_{j \in \mathcal{A}} p_j \mathbf{1}\{j \notin \Pi_k(x)\} - \inf_{\pi \in \mathcal{H}_\pi} \sum_{j \in \mathcal{A}} p_j \mathbf{1}\{j \notin \Pi_k(x)\} \leq k\Gamma_u^{-1}\left(\sum_{j \in \mathcal{A}} p_j \Phi_{01}^u(\pi, x, j) - \inf_{\pi \in \mathcal{H}_\pi} \sum_{j \in \mathcal{A}} p_j \Phi_{01}^u(\pi, x, j)\right),$$

*where $p \in \Delta^{|\mathcal{A}|}$ denotes a probability distribution over the set $\mathcal{A}$ and $k \leq |\mathcal{A}|$*

*Proof.* Let the top-$k$ loss be

$$\ell_k(\Pi_k(x), j) = \mathbf{1}\{j \notin \Pi_k(x)\},$$

and define its conditional risk as

$$\mathcal{C}_{\ell_k}(\pi, x) := \mathbb{E}_{Z|X=x}\big[\ell_k(\Pi_k(x), Z)\big] = \sum_{j \in \mathcal{A}} p_j(x) \, \mathbf{1}\{j \notin \Pi_k(x)\},$$

where $p_j(x) = \mathbb{P}(Z = j \mid X = x)$.

The excess conditional risk is

$$\Delta\mathcal{C}_{\ell_k}(\pi, x) := \mathcal{C}_{\ell_k}(\pi, x) - \inf_{\pi \in \mathcal{H}_\pi} \mathcal{C}_{\ell_k}(\pi, x).$$

Assume that the following pointwise calibration inequality holds for an increasing concave function $\Gamma_u^{-1}$ (Awasthi et al., 2022):

$$\sum_{j \in \mathcal{A}} p_j \mathbf{1}\{j \notin \Pi_k(x)\} - \inf_{\pi \in \mathcal{H}_\pi} \sum_{j \in \mathcal{A}} p_j \mathbf{1}\{j \notin \Pi_k(x)\} \leq k\Gamma_u^{-1}\left(\sum_{j \in \mathcal{A}} p_j \Phi_{01}^u(\pi, x, j) - \inf_{\pi \in \mathcal{H}_\pi} \sum_{j \in \mathcal{A}} p_j \Phi_{01}^u(\pi, x, j)\right), \tag{19}$$

we identify the corresponding conditional risks for a distribution $p \in \Delta^{|\mathcal{A}|}$ as done by Awasthi et al. (2022):

$$\mathcal{C}_{\ell_k}(\pi, x) - \inf_{\pi \in \mathcal{H}_\pi} \mathcal{C}_{\ell_k}(\pi, x) \leq k\Gamma_u^{-1}\left(\mathcal{C}_{\Phi_{01}^u}(\pi, x) - \inf_{\pi \in \mathcal{H}_\pi} \mathcal{C}_{\Phi_{01}^u}(\pi, x)\right). \tag{20}$$

Using the definition from Awasthi et al. (2022), we express the expected conditional risk difference as:

$$\mathbb{E}_X[\Delta\mathcal{C}_{\ell_k}(\pi, X)] = \mathbb{E}_X\left[\mathcal{C}_{\ell_k}(\pi, X) - \inf_{\pi \in \mathcal{H}_\pi}\mathcal{C}_{\ell_k}(\pi, X)\right]$$
$$= \mathcal{E}_{\ell_k}(\pi) - \mathcal{E}^B_{\ell_k}(\mathcal{H}_\pi) - \mathcal{U}_{\ell_k}(\mathcal{H}_\pi). \tag{21}$$

Consequently, we obtain:

$$\mathcal{C}_{\ell_k}(\pi, x) - \inf_{\pi \in \mathcal{H}_\pi}\mathcal{C}_{\ell_k}(\pi, x) \le k\Gamma_u^{-1}\left(\mathcal{C}_{\Phi^u_{01}}(\pi, x) - \inf_{\pi \in \mathcal{G}}\mathcal{C}_{\Phi^u_{01}}(\pi, x)\right). \tag{22}$$

Applying the expectation and by the Jensen's inequality yields:

$$\mathbb{E}_X[\Delta\mathcal{C}_{\ell_k}(\pi, X)] \le \mathbb{E}_X\left[k\Gamma_u^{-1}\left(\Delta\mathcal{C}_{\Phi^u_{01}}(\pi, X)\right)\right]$$
$$\mathbb{E}_X[\Delta\mathcal{C}_{\ell_k}(\pi, X)] \le k\Gamma_u^{-1}\left(\mathbb{E}_X\left[\Delta\mathcal{C}_{\Phi^u_{01}}(\pi, X)\right]\right). \tag{23}$$

Then,

$$\mathcal{E}_{\ell_k}(\pi) - \mathcal{E}^B_{\ell_k}(\mathcal{H}_\pi) - \mathcal{U}_{\ell_k}(\mathcal{H}_\pi) \le k\Gamma_u^{-1}\left(\mathcal{E}_{\Phi^u_{01}}(\pi) - \mathcal{E}^*_{\Phi^u_{01}}(\mathcal{H}_\pi) - \mathcal{U}_{\Phi^u_{01}}(\mathcal{H}_\pi)\right). \tag{24}$$

This result implies that the surrogate loss $\Phi^u_{01}$ is $\mathcal{H}_\pi$-consistent with respect to the top-$k$ loss $\ell_k$. From Cortes et al. (2024); Mao et al. (2023b), we have for the cross-entropy surrogates,

$$\Gamma_u(v) = \begin{cases} (1 - \sqrt{1 - v^2}) & u = 0 \\[2mm] \left(\frac{1+v}{2}\log[1+v] + \frac{1-v}{2}\log[1-v]\right) & u = 1 \\[2mm] \frac{1}{v(n+J)^v}\left[\left(\frac{(1+v)^{\frac{1}{1-v}} + (1-v)^{\frac{1}{1-v}}}{2}\right)^{1-v} - 1\right] & u \in (0,1) \\[2mm] \frac{1}{n+J}v & u = 2. \end{cases} \tag{25}$$

$\square$

**Theorem 4.7** (Unified Consistency for Top-$k$ Deferral). *Let $\mathcal{A}$ denote the set of entities. Assume that $\mathcal{H}_\pi$ is symmetric, complete, and regular for top-k deferral, and that in the two-stage case, $\mathcal{H}_g$ is the base predictor class. Let $S := (|\mathcal{A}| - 1)\sum_{j \in \mathcal{A}}\mathbb{E}_X[\overline{\mu}_j(X)]$. Suppose $\Phi^u_{01}$ is $\mathcal{H}_\pi$-consistent for top-k classification with a non-negative, non-decreasing, concave function $\Gamma_u^{-1}$.*

***One-stage.*** *Let $\mathbb{E}_X[\overline{\mu}_j(X)] = \alpha_j\mathbb{P}(\hat{a}_j(X) \ne Y) + \beta_j$. For any $h \in \mathcal{H}_h$,*

$$\mathcal{E}_{\ell_{\mathrm{def},k}}(h) - \mathcal{E}^B_{\ell_{\mathrm{def},k}}(\mathcal{H}_h) + \mathcal{U}_{\ell_{\mathrm{def},k}}(\mathcal{H}_h) \le k\,S\,\Gamma_u^{-1}\left(\frac{\mathcal{E}_{\Phi^u_{\mathrm{def},k}}(h) - \mathcal{E}^*_{\Phi^u_{\mathrm{def},k}}(\mathcal{H}_h) + \mathcal{U}_{\Phi^u_{\mathrm{def},k}}(\mathcal{H}_h)}{S}\right).$$

***Two-stage.*** *Let $\mathbb{E}_X[\overline{\mu}_j(X)] = \alpha_j\mathbb{E}_{X,Z}[\psi(\hat{a}_j(X), Z)] + \beta_j$. For any $(r, g) \in \mathcal{H}_r \times \mathcal{H}_g$,*

$$\mathcal{E}_{\ell_{\mathrm{def},k}}(r, g) - \mathcal{E}^B_{\ell_{\mathrm{def},k}}(\mathcal{H}_r, \mathcal{H}_g) + \mathcal{U}_{\ell_{\mathrm{def},k}}(\mathcal{H}_r, \mathcal{H}_g) \le \mathbb{E}_X[\overline{\mu}_1(X) - \inf_{g \in \mathcal{H}_g}\overline{\mu}_1(X)]$$
$$+ k\,S\,\Gamma_u^{-1}\left(\frac{\mathcal{E}_{\Phi^u_{\mathrm{def},k}}(r) - \mathcal{E}^*_{\Phi^u_{\mathrm{def},k}}(\mathcal{H}_r) + \mathcal{U}_{\Phi^u_{\mathrm{def},k}}(\mathcal{H}_r)}{S}\right)$$

*with $\Gamma_1(v) = \frac{1+v}{2}\log(1+v) + \frac{1-v}{2}\log(1-v)$ (logistic), $\Gamma_0(v) = 1 - \sqrt{1-v^2}$ (exponential), and $\Gamma_2(v) = v/|\mathcal{A}|$ (MAE).*

**One-stage.**

*Proof.* We begin by recalling the definition of the conditional deferral risk and its Bayes-optimal counterpart:

$$\mathcal{C}_{\ell_{\mathrm{def},k}}(\pi, x) = \sum_{j=1}^{n+J}\overline{\mu}_j(x)\mathbf{1}\{j \in \Pi_k(x)\}, \qquad \mathcal{C}^B_{\ell_{\mathrm{def},k}}(\mathcal{H}_\pi, x) = \sum_{i=1}^{k}\overline{\mu}_{[i]^\uparrow_{\overline{\mu}}}(x), \tag{26}$$

where $\overline{\mu}_j(x) = \mathbb{E}_{Y|X=x}[\mu_j(x, Y)]$ denotes the expected cost of selecting entity $j$. The calibration gap at input $x$ is defined as the difference between the incurred and optimal conditional risks:

$$\Delta \mathcal{C}_{\ell_{\text{def},k}}(\pi, x) = \mathcal{C}_{\ell_{\text{def},k}}(\pi, x) - \mathcal{C}^B_{\ell_{\text{def},k}}(\mathcal{H}_\pi, x). \tag{27}$$

To connect this quantity to surrogate risk, we use the reformulation used in the Proof of Lemma 4.3 in Equation 12:

$$\mathcal{C}_{\ell_{\text{def},k}}(\pi, x) = \sum_{j=1}^{n+J} \left( \sum_{i \neq j} \overline{\mu}_i(x) \right) \mathbf{1}\{j \notin \Pi_k(x)\} - (n + J - k - 1) \sum_{j=1}^{n+J} \overline{\mu}_j(x), \tag{28}$$

the second term is independent of the hypothesis $\pi \in \mathcal{H}_\pi$. This yields: To prepare for applying an $\mathcal{H}_\pi$-consistency result, we define normalized weights:

$$p_j = \frac{\sum_{i \neq j} \overline{\mu}_i(x)}{\sum_{j=1}^{n+J} \left( \sum_{i \neq j} \overline{\mu}_i(x) \right)},$$

which form a probability distribution over entities $j \in \mathcal{A}$. Then:

$$\Delta \mathcal{C}_{\ell_{\text{def},k}}(\pi, x) = \sum_{j=1}^{n+J} \left( \sum_{i \neq j} \overline{\mu}_i(x) \right) \left( \sum_{j=1}^{n+J} p_j \mathbf{1}\{j \notin \Pi_k(x)\} - \inf_{\pi \in \mathcal{H}_\pi} \sum_{j=1}^{n+J} p_j \mathbf{1}\{j \notin \Pi_k(x)\} \right).$$

Now, we apply the $\mathcal{H}_\pi$-consistency guarantee of the surrogate loss $\Phi_{01}^u$ for top-$k$ classification (Lemma A.9), which provides:

$$\sum_{j=1}^{n+J} p_j \mathbf{1}\{j \notin \Pi_k(x)\} - \inf_{\pi \in \mathcal{H}_\pi} \sum_{j=1}^{n+J} p_j \mathbf{1}\{j \notin \Pi_k(x)\} \leq$$

$$k\Gamma_u^{-1} \left( \sum_{j=1}^{n+J} p_j \Phi_{01}^u(\pi, x, j) - \inf_{\pi \in \mathcal{H}_\pi} \sum_{j=1}^{n+J} p_j \Phi_{01}^u(\pi, x, j) \right).$$

Multiplying both sides by $\sum_{j=1}^{n+J} \left( \sum_{i \neq j} \overline{\mu}_i(x) \right)$, we obtain:

$$\Delta \mathcal{C}_{\ell_{\text{def},k}}(\pi, x) \leq$$

$$\sum_{j=1}^{n+J} \left( \sum_{i \neq j} \overline{\mu}_i(x) \right) k\Gamma_u^{-1} \left( \frac{\sum_{j=1}^{n+J} (\sum_{i \neq j} \overline{\mu}_i(x)) \Phi_{01}^u(\pi, x, j) - \inf_{\pi \in \mathcal{H}_\pi} \sum_{j=1}^{n+J} (\sum_{i \neq j} \overline{\mu}_i(x)) \Phi_{01}^u(\pi, x, j)}{\sum_{j=1}^{n+J} \left( \sum_{i \neq j} \overline{\mu}_i(x) \right)} \right).$$

Define the calibration gap of the surrogate as:

$$\Delta \mathcal{C}_{\Phi_{\text{def},k}^u}(\pi, x) = \sum_{j=1}^{n+J} \overline{\mu}_j(x) \Phi_{01}^u(\pi, x, j) - \inf_{\pi \in \mathcal{H}_\pi} \sum_{j=1}^{n+J} \overline{\mu}_j(x) \Phi_{01}^u(\pi, x, j),$$

Then,

$$\Delta \mathcal{C}_{\ell_{\text{def},k}}(\pi, x) \leq \sum_{j=1}^{n+J} \left( \sum_{i \neq j} \overline{\mu}_i(x) \right) k\Gamma_u^{-1} \left( \frac{\Delta \mathcal{C}_{\Phi_{\text{def},k}^u}(\pi, x)}{\sum_{j=1}^{n+J} \left( \sum_{i \neq j} \overline{\mu}_i(x) \right)} \right).$$

Taking expectations:

$$\mathcal{E}_{\ell_{\text{def},k}}(\pi) - \mathcal{E}^B_{\ell_{\text{def},k}}(\mathcal{H}_\pi) - \mathcal{U}_{\ell_{\text{def},k}}(\mathcal{H}_\pi) \leq$$

$$k \sum_{j=1}^{n+J} \left( \sum_{i \neq j} \mathbb{E}_X[\overline{\mu}_i(X)] \right) \Gamma_u^{-1} \left( \frac{\mathcal{E}_{\Phi_{\text{def},k}^u}(\pi) - \mathcal{E}^*_{\Phi_{\text{def},k}^u}(\mathcal{H}_\pi) - \mathcal{U}_{\Phi_{\text{def},k}^u}(\mathcal{H}_\pi)}{\sum_{j=1}^{n+J} \left( \sum_{i \neq j} \mathbb{E}_X[\overline{\mu}_i(X)] \right)} \right), \tag{29}$$

Note that we have $\sum_{j=1}^{n+J} \left( \sum_{i \neq j} \mathbb{E}_X[\overline{\mu}_i(X)] \right) = (|\mathcal{A}| - 1) \sum_{j \in \mathcal{A}} \mathbb{E}_X[\overline{\mu}_j(X)]$ with $\mathbb{E}_X[\overline{\mu}_j(X)] = \alpha_j \mathbb{P}(\hat{a}_j(X) \neq Y) + \beta_j$, leading to $S = (|\mathcal{A}| - 1) \sum_{j \in \mathcal{A}} \left( \alpha_j \mathbb{P}(\hat{a}_j(X) \neq Y) + \beta_j \right)$:

$$\mathcal{E}_{\ell_{\text{def},k}}(\pi) - \mathcal{E}^B_{\ell_{\text{def},k}}(\mathcal{H}_\pi) - \mathcal{U}_{\ell_{\text{def},k}}(\mathcal{H}_\pi) \leq$$
$$k \, S \, \Gamma_u^{-1} \left( \frac{\mathcal{E}_{\Phi^u_{\text{def},k}}(\pi) - \mathcal{E}^*_{\Phi^u_{\text{def},k}}(\mathcal{H}_\pi) - \mathcal{U}_{\Phi^u_{\text{def},k}}(\mathcal{H}_\pi)}{S} \right), \tag{30}$$

$\square$

**Two-Stage.**

*Proof.* We begin by recalling the definition of the conditional deferral risk and its Bayes-optimal counterpart:

$$\mathcal{C}_{\ell_{\text{def},k}}(\pi, g, x) = \sum_{j=1}^{J+1} \overline{\mu}_j(x) \mathbf{1}\{j \in \Pi_k(x)\}, \qquad \mathcal{C}^B_{\ell_{\text{def},k}}(\mathcal{H}_\pi, \mathcal{H}_g, x) = \sum_{i=1}^{k} \overline{\mu}^B_{[i]^{\uparrow}_{\overline{\mu}^B}}(x), \tag{31}$$

where $\overline{\mu}_j(x) = \mathbb{E}_{Z|X=x}[\mu_j(x, Z)]$ denotes the expected cost of selecting entity $j$. Note that the conditional risk is different because of the main predictor $g$. The calibration gap at input $x$ is defined as the difference between the incurred and optimal conditional risks:

$$\begin{aligned} \Delta \mathcal{C}_{\ell_{\text{def},k}}(\pi, g, x) &= \mathcal{C}_{\ell_{\text{def},k}}(\pi, g, x) - \mathcal{C}^B_{\ell_{\text{def},k}}(\mathcal{H}_\pi, \mathcal{H}_g, x) \\ &= \sum_{i=1}^{J+1} \overline{\mu}_j(x) \mathbf{1}\{j \in \Pi_k(x)\} - \sum_{i=1}^{k} \overline{\mu}^B_{[i]^{\uparrow}_{\overline{\mu}^B}}(x) \\ &= \sum_{i=1}^{J+1} \overline{\mu}_j(x) \mathbf{1}\{j \in \Pi_k(x)\} - \sum_{i=1}^{k} \overline{\mu}_{[i]^{\uparrow}_{\overline{\mu}}}(x) \\ &\quad + \left( \sum_{i=1}^{k} \overline{\mu}_{[i]^{\uparrow}_{\overline{\mu}}}(x) - \sum_{i=1}^{k} \overline{\mu}^B_{[i]^{\uparrow}_{\overline{\mu}^B}}(x) \right). \end{aligned} \tag{32}$$

Observing that:

$$\sum_{i=1}^{k} \overline{\mu}_{[i]^{\uparrow}_{\overline{\mu}}}(x) - \sum_{i=1}^{k} \overline{\mu}^B_{[i]^{\uparrow}_{\overline{\mu}^B}}(x) \leq \overline{\mu}_1(x) - \inf_{g \in \mathcal{H}_g} \overline{\mu}_1(x) \tag{33}$$

Since the only contribution of $g$ appears through the cost term $\mu_1$, we can rewrite the first term in terms of conditional risks. Importantly, the minimization is carried out only over the decision rule $\pi \in \mathcal{H}_\pi$.

$$\sum_{i=1}^{J+1} \overline{\mu}_j(x) \mathbf{1}\{j \in \Pi_k(x)\} - \sum_{i=1}^{k} \overline{\mu}_{[i]^{\uparrow}_{\overline{\mu}}}(x) = \mathcal{C}_{\ell_{\text{def},k}}(\pi, g, x) - \inf_{\pi \in \mathcal{H}_\pi} \mathcal{C}_{\ell_{\text{def},k}}(\pi, g, x). \tag{34}$$

Using the explicit formulation of the top-$k$ deferral loss in terms of the indicator function $\mathbf{1}\{j \notin \Pi_k(x)\}$ (Equation 12), we obtain:

$$\begin{aligned} \mathcal{C}_{\ell_{\text{def},k}}(\pi, g, x) - \inf_{\pi \in \mathcal{H}_\pi} \mathcal{C}_{\ell_{\text{def},k}}(\pi, g, x) &= \sum_{j=1}^{n+J} \left( \sum_{i \neq j} \overline{\mu}_i(x) \right) \mathbf{1}\{j \notin \Pi_k(x)\} \\ &\quad - \inf_{\pi \in \mathcal{H}_\pi} \sum_{j=1}^{n+J} \left( \sum_{i \neq j} \overline{\mu}_i(x) \right) \mathbf{1}\{j \notin \Pi_k(x)\}. \end{aligned} \tag{35}$$

Using the same change of variable:

$$p_j = \frac{\sum_{i \neq j} \overline{\mu}_i(x)}{\sum_{j=1}^{J+1} \left( \sum_{i \neq j} \overline{\mu}_i(x) \right)},$$

which form a probability distribution over entities $j \in \mathcal{A}$. Then:

$$\mathcal{C}_{\ell_{\mathrm{def},k}}(\pi, g, x) - \inf_{\pi \in \mathcal{H}_\pi} \mathcal{C}_{\ell_{\mathrm{def},k}}(\pi, g, x) = \sum_{j=1}^{J+1} \Big( \sum_{i \neq j} \overline{\mu}_i(x) \Big) \Bigg( \sum_{j=1}^{n+J} p_j \mathbf{1}\{j \notin \Pi_k(x)\}$$
$$- \inf_{\pi \in \mathcal{H}_\pi} \sum_{j=1}^{n+J} p_j \mathbf{1}\{j \notin \Pi_k(x)\} \Bigg)$$

Since the surrogate losses $\Phi_{01}^u$ are consistent with the top-$k$ loss, we apply Lemma A.9:

$$\mathcal{C}_{\ell_{\mathrm{def},k}}(\pi, g, x) - \inf_{\pi \in \mathcal{H}_\pi} \mathcal{C}_{\ell_{\mathrm{def},k}}(\pi, g, x) \leq \sum_{j=1}^{J+1} \Big( \sum_{i \neq j} \overline{\mu}_i(x) \Big) k \Gamma_u^{-1} \Bigg( \sum_{j=1}^{n+J} p_j \mathbf{1}\{j \notin \Pi_k(x)\}$$
$$- \inf_{\pi \in \mathcal{H}_\pi} \sum_{j=1}^{n+J} p_j \mathbf{1}\{j \notin \Pi_k(x)\} \Bigg)$$
$$= \sum_{j=1}^{J+1} \Big( \sum_{i \neq j} \overline{\mu}_i(x) \Big) k \Gamma_u^{-1} \Bigg( \frac{\mathcal{C}_{\Phi_{\mathrm{def},k}}(\pi, g, x) - \inf_{\pi \in \mathcal{H}_\pi} \mathcal{C}_{\Phi_{\mathrm{def},k}}(\pi, g, x)}{\sum_{j=1}^{J+1} \Big( \sum_{i \neq j} \overline{\mu}_i(x) \Big)} \Bigg)$$

Earlier, we have stated:

$$\Delta \mathcal{C}_{\ell_{\mathrm{def},k}}(\pi, g, x) = \mathcal{C}_{\ell_{\mathrm{def},k}}(\pi, g, x) - \mathcal{C}_{\ell_{\mathrm{def},k}}^B(\mathcal{H}_\pi, \mathcal{H}_g, x)$$
$$= \sum_{i=1}^{J+1} \overline{\mu}_j(x) \mathbf{1}\{j \in \Pi_k(x)\} - \sum_{i=1}^{k} \overline{\mu}_{[i]_{\overline{\mu}}^\uparrow}(x) \qquad (36)$$
$$+ \Big( \sum_{i=1}^{k} \overline{\mu}_{[i]_{\overline{\mu}}^\uparrow}(x) - \sum_{i=1}^{k} \overline{\mu}_{[i]_{\overline{\mu}^B}^\uparrow}^B(x) \Big).$$

which is

$$\Delta \mathcal{C}_{\ell_{\mathrm{def},k}}(\pi, g, x) = \mathcal{C}_{\ell_{\mathrm{def},k}}(\pi, g, x) - \inf_{\pi \in \mathcal{H}_\pi} \mathcal{C}_{\ell_{\mathrm{def},k}}(\pi, g, x)$$
$$+ \Big( \sum_{i=1}^{k} \overline{\mu}_{[i]_{\overline{\mu}}^\uparrow}(x) - \sum_{i=1}^{k} \overline{\mu}_{[i]_{\overline{\mu}^B}^\uparrow}^B(x) \Big). \qquad (37)$$
$$\leq \mathcal{C}_{\ell_{\mathrm{def},k}}(\pi, g, x) - \inf_{\pi \in \mathcal{H}_\pi} \mathcal{C}_{\ell_{\mathrm{def},k}}(\pi, g, x) + \Big( \overline{\mu}_1(x) - \overline{\mu}_1^B(x) \Big)$$

Then,

$$\Delta \mathcal{C}_{\ell_{\mathrm{def},k}}(\pi, g, x) \leq \sum_{j=1}^{J+1} \Big( \sum_{i \neq j} \overline{\mu}_i(x) \Big) k \Gamma_u^{-1} \Bigg( \frac{\mathcal{C}_{\Phi_{\mathrm{def},k}^u}(\pi, g, x) - \inf_{\pi \in \mathcal{H}_\pi} \mathcal{C}_{\Phi_{\mathrm{def},k}^u}(\pi, g, x)}{\sum_{j=1}^{J+1} \Big( \sum_{i \neq j} \overline{\mu}_i(x) \Big)} \Bigg)$$
$$+ \Big( \overline{\mu}_1(x) - \overline{\mu}_1^B(x) \Big) \qquad (38)$$

Taking expectations,

$$\mathcal{E}_{\ell_{\mathrm{def},k}}(\pi, g) - \mathcal{E}_{\ell_{\mathrm{def},k}}^B(\mathcal{H}_\pi, \mathcal{H}_g) - \mathcal{U}_{\ell_{\mathrm{def},k}}(\mathcal{H}_\pi, \mathcal{H}_g) \leq \mathbb{E}_X[\overline{\mu}_1(X) - \overline{\mu}_1^B(X)]$$
$$+ \sum_{j=1}^{J+1} \Big( \sum_{i \neq j} \mathbb{E}_X[\overline{\mu}_i(X)] \Big) k \Gamma_u^{-1} \Bigg( \frac{\mathcal{E}_{\Phi_{\mathrm{def},k}^u}(\pi) - \mathcal{E}_{\Phi_{\mathrm{def},k}^u}^*(\mathcal{H}_\pi) - \mathcal{U}_{\Phi_{\mathrm{def},k}^u}(\mathcal{H}_\pi)}{\sum_{j=1}^{J+1} \Big( \sum_{i \neq j} \mathbb{E}_X[\overline{\mu}_i(X)] \Big)} \Bigg) \qquad (39)$$

Similarly, we have $\mathbb{E}_X[\overline{\mu}_j(X)] = \alpha_j \mathbb{E}_{X,Z}[\psi(\hat{a}_j(X), Z)] + \beta_j$. Using $\sum_{j=1}^{n+J} \Big( \sum_{i \neq j} \mathbb{E}_X[\overline{\mu}_i(X)] \Big) = (|\mathcal{A}| - 1) \sum_{j \in \mathcal{A}} \mathbb{E}_X[\overline{\mu}_j(X)]$, leading to $S = (|\mathcal{A}| - 1) \sum_{j \in \mathcal{A}} \Big( \alpha_j \mathbb{E}_{X,Z}[\psi(\hat{a}_j(X), Z)] + \beta_j \Big)$: $\qquad \square$

A.12 BEHAVIOR OF THE CARDINALITY-AWARE DEFERRAL LOSS

For any $x \in \mathcal{X}$ and $k \in \{1, \ldots, |\mathcal{A}|\}$, the conditional risk of selecting the top-$k$ experts is

$$\mathcal{C}_{\ell_{\text{card}}}(k) := \mathbb{E}\big[\ell_{\text{card}}\big(\Pi_k(x), k, x, Z\big) \mid X = x\big]$$

$$= \mathbb{E}\big[d\big(\Pi_k(x), x, Z\big) \mid X = x\big] + \lambda \xi\left(\sum_{i=1}^{k} \beta_{[i]\downarrow_\pi}\right).$$

The Bayes-optimal cardinality function is therefore

$$k_\theta^B(x) \in \arg\min_{k \in \{1, \ldots, |\mathcal{A}|\}} \left\{ \mathbb{E}\big[d\big(\Pi_k(x), x, Z\big) \mid X = x\big] + \lambda \xi\left(\sum_{i=1}^{k} \beta_{[i]\downarrow_\pi}\right)\right\}.$$

For $k \geq 2$, define

$$\delta\mathcal{C}_{\ell_{\text{card}}}(k) := \mathcal{C}_{\ell_{\text{card}}}(k) - \mathcal{C}_{\ell_{\text{card}}}(k-1), \qquad S_k := \sum_{i=1}^{k} \beta_{[i]\downarrow_\pi}.$$

A simple computation gives

$$\delta\mathcal{C}_{\ell_{\text{card}}}(k) = \underbrace{\mathbb{E}\big[d\big(\Pi_k(x), x, Z\big) - d\big(\Pi_{k-1}(x), x, Z\big) \mid X = x\big]}_{:= \delta D_x(k)} + \lambda[\xi(S_k) - \xi(S_{k-1})].$$

Thus, for any $k \in \{1, \ldots, |\mathcal{A}| - 1\}$,

$$\mathcal{C}_{\ell_{\text{card}}}(k+1) \leq \mathcal{C}_{\ell_{\text{card}}}(k) \iff \delta D_x(k+1) + \lambda\big[\xi(S_{k+1}) - \xi(S_k)\big] \leq 0, \tag{40}$$

that is, moving from $k$ to $k+1$ strictly decreases the conditional risk if and only if the marginal reduction in expected error, $-\delta D_x(k+1)$, is at least as large as the marginal regularization cost, $\lambda\big[\xi(S_{k+1}) - \xi(S_k)\big]$.

This shows that consulting and aggregating multiple experts is not ad hoc: for any fixed aggregation rule $d$, it is Bayes-optimal to choose $k_\theta^B(x) > 1$ precisely when using additional experts yields a net decrease of the conditional risk $\mathcal{C}_{\ell_{\text{card}}}(k)$, i.e., when improving the final prediction quality is worthy the price of consulting selected experts.

**Tuning Parameters.** As shown above, the parameter $\lambda$ controls the trade-off between consultation cost and predictive reliability. Increasing $\lambda$ makes the model more cost-sensitive, leading it to select fewer experts. Conversely, decreasing $\lambda$ places greater emphasis on reliability, resulting in the selection of a larger set of experts when beneficial.

A.13 CHOICE OF THE METRIC $d$

The metric $d$ in the cardinality-based deferral loss governs how disagreement between the final prediction and labels is penalized, and its choice depends on application-specific priorities. For instance, it determines how predictions from multiple entities in the Top-$k$ Selection Set $\Pi_k(x) \subseteq \mathcal{A}$ are aggregated into a final decision. In all cases, ties are broken by selecting the entity with the smallest index.

**Classification Metrics for Cardinality Loss.** In classification, common choices include:

- **Top-$k$ True Loss** A binary penalty is incurred when the true label $y$ is not present in the prediction set:

$$d_{\text{top}-k}(\Pi_2(x), 2, y) = \mathbf{1}\{y \notin \{a_{[1]_\pi^\downarrow}(x), \ldots, a_{[k]_\pi^\downarrow}(x)\}\}.$$

  Example: let $\Pi_2(x) = \{3, 1\}$ the metric will compute $d_{\text{top}-k}(\Pi_k(x), k, y) = \mathbf{1}\{y \notin \{a_1(x), a_3(x)\}\}$.

- **Weighted Voting Loss.** Each entity is weighted according to a reliability score, typically derived from a softmax over the scores $\pi(x,)$. The predicted label is obtained via weighted voting:

$$\hat{y} = \arg\max_{y \in \mathcal{Y}} \sum_{j \in \Pi_k(x)} w_j \mathbf{1}\{\hat{a}_j(x) = y\}, \quad \text{with} \quad w_j = \hat{p}(x,j) = \frac{\exp(\pi(x,j))}{\sum_{j'} \exp(\pi(x,j'))}.$$

The loss is defined as $d_{\text{w-vl}}(\Pi_k(x), k, y) = \mathbf{1}\{y \neq \hat{y}\}$.

- **Majority Voting Loss.** All entities contribute equally, and the predicted label is chosen by majority vote:

$$\hat{y} = \arg\max_{y \in \mathcal{Y}} \sum_{j \in \Pi_k(x)} \mathbf{1}\{\hat{a}_j(x) = y\},$$

with the corresponding loss $d_{\text{maj}}(\Pi_k(x), k, y) = \mathbf{1}\{y \neq \hat{y}\}$.

**Regression Metrics for Cardinality Loss.** Let $\ell_{\text{reg}}(z, \hat{z}) \in \mathbb{R}^+$ denote a base regression loss (e.g., squared error or smooth L1). Common choices include:

- **Minimum Cost (Best Expert) Loss.** The error is measured using the prediction from the best-performing entity in the Top-$k$ Selection Set:

$$d_{\min}(\Pi_k(x), k, z) = \min_{j \in \Pi_k(x)} \ell_{\text{reg}}(\hat{a}_j(x), z).$$

- **Weighted Average Prediction Loss.** Each entity is assigned a reliability weight based on a softmax over scores $\pi(x,)$. The predicted output is a weighted average of entity predictions:

$$\hat{z} = \sum_{j \in \Pi_k(x)} w_j \hat{a}_j(x), \quad \text{with} \quad w_j = \frac{\exp(\pi(x,j))}{\sum_{j'} \exp(\pi(x,j'))},$$

and the loss is computed as $d_{\text{w-avg}}(\Pi_k(x), k, z) = \ell_{\text{reg}}(\hat{z}, z)$.

- **Uniform Average Prediction Loss.** Each entity in the Top-$k$ Selection Set contributes equally, and the final prediction is a simple average:

$$\hat{z} = \frac{1}{k} \sum_{j \in \Pi_k(x)} \hat{a}_j(x), \quad d_{\text{avg}}(\Pi_k(x), k, z) = \ell_{\text{reg}}(\hat{z}, z).$$

### A.14 USE OF LARGE LANGUAGE MODELS

Large language models were employed exclusively as writing aids for this manuscript. In particular, we used them to refine the text with respect to vocabulary choice, orthography, and grammar. All conceptual contributions, technical results, proofs, and experiments are original to the authors. The LLMs were not used to generate research ideas, mathematical derivations, or experimental analyses.

## B EXPERIMENTS

### B.1 RESOURCES

All experiments were conducted on an internal cluster using an NVIDIA A100 GPU with 40 GB of VRAM.

### B.1.1 METRICS

For classification tasks, we report accuracy under three evaluation rules. The *Top-k Accuracy* is defined as $\text{Acc}_{\text{top-}k} = \mathbb{E}_X[1 - d_{\text{top-}k}(X)]$, where the prediction is deemed correct if the true label $y$ is included in the outputs of the queried entities. The *Weighted Voting Accuracy* is given by $\text{Acc}_{\text{w-vl}} = \mathbb{E}_X[1 - d_{\text{w-vl}}(X)]$, where entity predictions are aggregated via softmax-weighted voting.

Finally, the *Majority Voting Accuracy* is defined as $\text{Acc}_{\text{maj}} = \mathbb{E}_X[1 - d_{\text{maj}}(X)]$, where all entities in the Top-$k$ Selection Set contribute equally.

For regression tasks, we report RMSE under three aggregation strategies. The *Minimum Cost RMSE* is defined as $\text{RMSE}_{\text{min}} = \mathbb{E}_X[d_{\text{min}}(X)]$, corresponding to the prediction from the best-performing entity. The *Weighted Average Prediction RMSE* is given by $\text{RMSE}_{\text{w-avg}} = \mathbb{E}_X[d_{\text{w-avg}}(X)]$, using a softmax-weighted average of predictions. The *Uniform Average Prediction RMSE* is computed as $\text{RMSE}_{\text{avg}} = \mathbb{E}_X[d_{\text{avg}}(X)]$, using the unweighted mean of entity predictions.

In addition to performance, we also report two resource-sensitive metrics. The *expected budget* is defined as $\overline{\beta}(k) = \mathbb{E}_X\left[\sum_{j=1}^{k} \beta_{[j]_{\pi}^{\downarrow}}\right]$, where $\beta_j$ denotes the consultation cost of entity $j$, and $[j]_{\pi}^{\downarrow}$ is the index of the $j$-th ranked entity by the policy $\pi$. The *expected number of queried entities* is given by $\overline{k} = \mathbb{E}_X[|\Pi_k(X)|]$, where $k$ is fixed for Top-$k$ L2D and varies with $x$ in the adaptive Top-$k(x)$ L2D Settings. Additional details are provided in Appendix A.13.

### B.1.2 TRAINING

We assign fixed consultation costs $\beta_j$ to each entity. In the one-stage regime, class labels ($j \leq n$) incur no consultation cost ($\beta_j = 0$), since predictions from the model itself are free. In the two-stage regime, we similarly set $\beta_1 = 0$ for the base predictor $g$. For experts, we use the cost schedule $\beta_j \in \{0.05, 0.045, 0.040, 0.035, 0.03\}$, with $m_1$ assigned as the most expensive. This decreasing pattern reflects realistic setups where experts differ in reliability and cost. As the surrogate loss, we adopt the multiclass log-softmax surrogate $\Phi_{01}^{u=1}(q, x, j) = -\log\left(\frac{e^{q(x,j)}}{\sum_{j' \in \mathcal{A}} e^{q(x,j')}}\right)$, used both for learning the policy $\pi \in \mathcal{H}_{\pi}$ and for optimizing the adaptive cardinality function $k_{\theta}$. The adaptive function $k_{\theta}$ is trained under three evaluation protocols—Top-$k$ Accuracy, Majority Voting, and Weighted Voting (see A.13 and B.1.1). To balance accuracy and consultation cost, we perform a grid search over the regularization parameter $\lambda \in \{10^{-9}, 0.01, 0.05, 0.25, 0.5, 1, 1.5, \ldots, 6.5\}$, which directly shapes the learned values of $\hat{k}(x)$. Larger $\lambda$ penalizes expensive deferral sets, encouraging smaller $k$. When multiple values of $k$ achieve the same loss, ties are broken by selecting the smallest index according to a fixed ordering of entities in $\mathcal{A}$.

### B.1.3 DATASETS

**CIFAR-10.** A standard image classification benchmark with 60,000 color images of resolution $32 \times 32$, evenly distributed across 10 object categories (Krizhevsky, 2009). Each class has 6,000 examples, with 50,000 for training and 10,000 for testing. We follow the standard split and apply dataset-specific normalization.

**CIFAR-100.** Identical setup to CIFAR-10 but with 100 categories, each containing 600 images.

**SVHN.** The Street View House Numbers (SVHN) dataset (Goodfellow et al., 2013) is a large-scale digit classification benchmark comprising over 600,000 RGB images of size $32 \times 32$, extracted from real-world street scenes. We use the standard split of 73,257 training images and 26,032 test images. The dataset is released under a non-commercial use license.

**California Housing.** The California Housing dataset (Kelley Pace & Barry, 1997) is a regression benchmark based on the 1990 U.S. Census (CC0). It contains 20,640 instances, each representing a geographical block in California and described by eight real-valued features (e.g., median income, average occupancy). The target is the median house value in each block, measured in hundreds of thousands of dollars. We standardize all features and use an 80/20 train-test split.

### B.2 ONE-STAGE

We compare our proposed *Top-$k$* and *Top-$k(x)$* L2D approaches against prior work (Mozannar & Sontag, 2020; Mao et al., 2024a), as well as against random and oracle (optimal) baselines.

### B.2.1 RESULTS ON CIFAR-10

**Settings.** We synthetically construct a pool of 6 experts with overlapping areas of competence. Each expert is assigned to a subset of 5 target classes, where they achieve a high probability of correct prediction ($p = 0.94$). For all other (non-assigned) classes, their predictions are uniformly random (Mozannar & Sontag, 2020; Verma et al., 2022). This design reflects a realistic setting where experts specialize in overlapping but not disjoint regions of the input space. Table 2 reports the classification accuracy of each expert on the CIFAR-10 validation set.

Table 2: Validation accuracy of each expert on CIFAR-10. Each expert specializes in 5 out of 10 classes with high confidence.

| Expert | 1 | 2 | 3 | 4 | 5 | 6 |
|---|---|---|---|---|---|---|
| Accuracy (%) | 52.08 | 52.68 | 52.11 | 52.03 | 52.16 | 52.41 |

**Top-$k$ One-Stage.** We train the classifier $h \in \mathcal{H}_h$ using a ResNet-4 architecture (He et al., 2016), following the procedure described in Algorithm 1 ($\pi = h$). Optimization is performed using the Adam optimizer with a batch size of 2048, an initial learning rate of $1 \times 10^{-3}$, and 200 training epochs. The final policy $h$ is selected based on the lowest Top-$k$ deferral surrogate loss (Corollary 4.4) on a held-out validation set. We report results across various fixed values of $k \in \mathcal{A}^{1s}$, corresponding to the number of queried entities at inference.

**Top-$k(x)$ One-Stage.** Given the trained classifier $h$, we train a cardinality function $k_\theta \in \mathcal{H}_k$ as described in Algorithm A.4. This function is implemented using a CLIP-based image encoder (Radford et al., 2021) followed by a classification head. We train $k$ using the AdamW optimizer (Loshchilov & Hutter, 2017) with a batch size of 128, learning rate of $1 \times 10^{-3}$, weight decay of $1 \times 10^{-5}$, and cosine learning rate scheduling over 10 epochs. To evaluate the learned deferral strategy, we experiment with different decision rules based on various metrics $d$; detailed definitions and evaluation setups are provided in Appendix A.13.

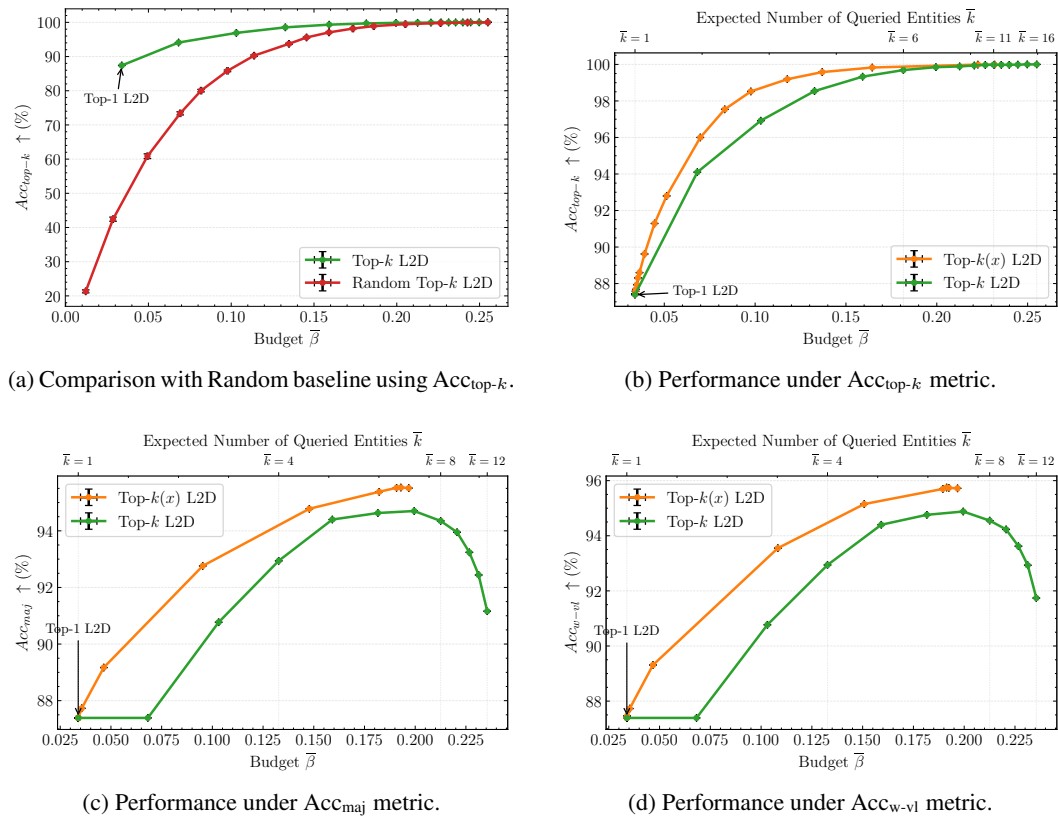

(a) Comparison with Random baseline using $Acc_{top\text{-}k}$.

(b) Performance under $Acc_{top\text{-}k}$ metric.

(c) Performance under $Acc_{maj}$ metric.

(d) Performance under $Acc_{w\text{-}vl}$ metric.

Figure 5: Comparison of Top-$k$ and Top-$k(x)$ One-Stage across four accuracy metrics on CIFAR-10. Top-$k(x)$ achieves better budget-accuracy trade-offs across all settings. For clarity, only the first 12 entities are shown. Results are averaged over 4 independent runs. The Top-1 L2D corresponds to Mozannar & Sontag (2020); Mao et al. (2024a).

**Performance Comparison.** Figure 5 summarizes our results for both Top-$k$ and adaptive Top-$k(x)$ One-Stage surrogates on CIFAR-10. In Figure 5b, we report the Top-$k$ Accuracy as a function of the average consultation budget $\overline{\beta}$. As expected, the Top-1 L2D method (Mozannar & Sontag, 2020) is recovered as a special case of our Top-$k$ framework, and is strictly outperformed as $k$ increases. More importantly, the adaptive Top-$k(x)$ consistently dominates fixed-$k$ strategies for a given budget level across all metrics. Notably, Top-$k(x)$ achieves its highest Majority Voting Accuracy of 95.53% at a budget of $\overline{\beta} = 0.192$, outperforming the best Top-$k$ result of 94.7%, which requires a higher budget of $\overline{\beta} = 0.199$ (Figure 5c). A similar gain is observed under the Weighted Voting metric: Top-$k(x)$ again reaches 95.53% at $\overline{\beta} = 0.191$, benefiting from its ability to leverage classifier scores for soft aggregation (Figure 5d).

This performance gain arises from the ability of the learned cardinality function $k(x)$ to select the most cost-effective subset of entities. For simple inputs, Top-$k(x)$ conservatively queries a small number of entities; for complex or ambiguous instances, it expands the deferral set to improve reliability. Additionally, we observe that increasing $k$ indiscriminately may inflate the consultation cost and introduce potential bias in aggregation-based predictions (e.g., through overdominance of unreliable entities in majority voting). The Top-$k(x)$ mechanism mitigates this by adjusting $k$ dynamically, thereby avoiding the inefficiencies and inaccuracies that arise from over-querying.

### B.2.2 RESULTS ON SVHN

**Settings.** We construct a pool of six experts, each based on a ResNet-18 architecture (He et al., 2016), trained and evaluated on different subsets of the dataset. These subsets are synthetically generated by selecting three classes per expert, with one class overlapping between consecutive experts to ensure partial redundancy. Each expert is trained for 20 epochs using the Adam opti-

mizer (Kingma & Ba, 2014) with a learning rate of $1 \times 10^{-3}$. Model selection is based on the lowest validation loss computed on each expert's respective subset. Table 3 reports the classification accuracy of each trained expert, evaluated on the full SVHN validation set.

Table 3: Accuracy of each expert on the SVHN validation set.

| Expert | 1 | 2 | 3 | 4 | 5 | 6 |
|---|---|---|---|---|---|---|
| Accuracy (%) | 45.16 | 35.56 | 28.64 | 25.68 | 23.64 | 18.08 |

**Top-$k$ and Top-$k(x)$ One-Stage.** We adopt the same training configuration as in the CIFAR-10 experiments, including architecture, optimization settings, and evaluation protocol.

**Performance Comparison.** Figure 6 shows results consistent with those observed on CIFAR-10. Our Top-$k$ One-Stage framework successfully generalizes the standard Top-1 method (Mozannar & Sontag, 2020). Moreover, the adaptive Top-$k(x)$ variant consistently outperforms the fixed-$k$ approach across all three evaluation metrics, further confirming its effectiveness in balancing accuracy and consultation cost.

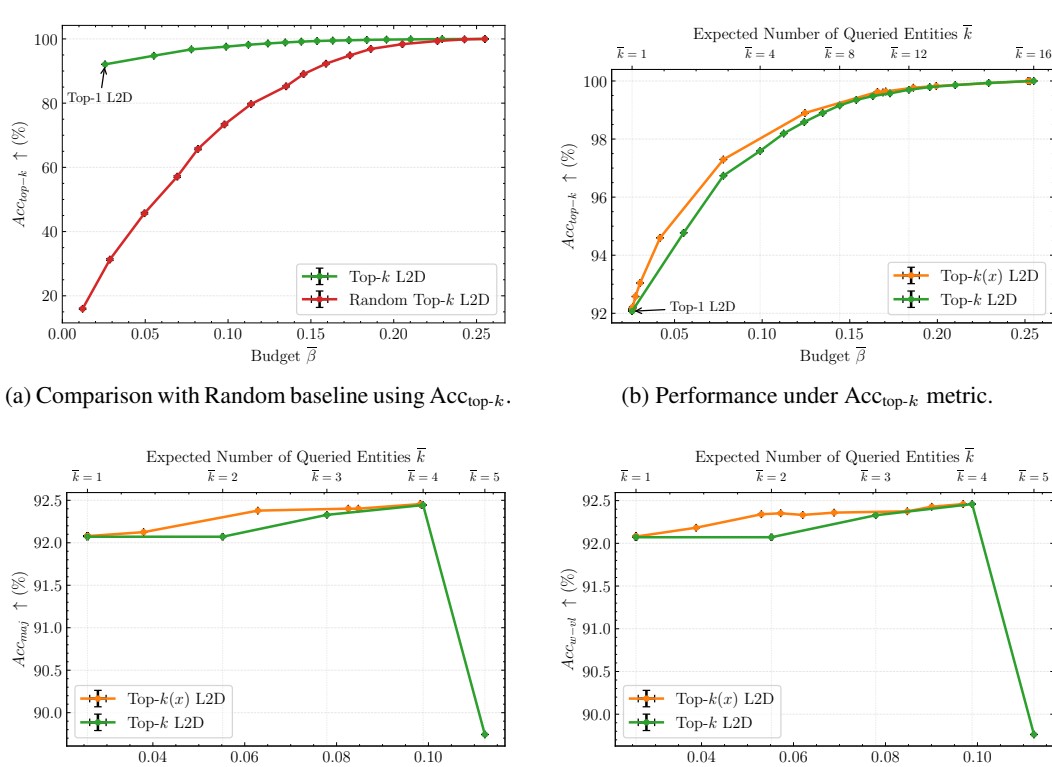

(a) Comparison with Random baseline using $\text{Acc}_{\text{top-}k}$.

(b) Performance under $\text{Acc}_{\text{top-}k}$ metric.

(c) Performance under $\text{Acc}_{\text{maj}}$ metric.

(d) Performance under $\text{Acc}_{\text{w-vl}}$ metric.

Figure 6: Comparison of Top-$k$ and Top-$k(x)$ One-Stage across four accuracy metrics on SVHN. Top-$k(x)$ achieves better budget-accuracy trade-offs across all settings. For clarity, only the first 5 entities are shown. Results are averaged over 4 independent runs. The Top-1 L2D corresponds to Mozannar & Sontag (2020); Mao et al. (2024a).

## B.3 TWO-STAGE

We compare our proposed *Top-k* and *Top-k(x)* L2D approaches against prior work (Narasimhan et al., 2022; Mao et al., 2023a; 2024c; Montreuil et al., 2025b), as well as against random and oracle (optimal) baselines.

### B.3.1 RESULTS ON CALIFORNIA HOUSING.

**Settings.** We construct a pool of 6 regression entities (five experts and one main predictor), each trained on a predefined, spatially localized subset of the California Housing dataset. To simulate domain specialization, each entity is associated with a specific geographical region of California, reflecting scenarios in which real estate professionals possess localized expertise. The training regions are partially overlapping to introduce heterogeneity and ensure that no single entity has access to all regions, thereby creating a realistic setting for deferral and expert allocation.

We train each entity using a multilayer perceptron (MLP) for 30 epochs with a batch size of 256, a learning rate of $1 \times 10^{-3}$, optimized using Adam. Model selection is based on the checkpoint achieving the lowest RMSE on the entity's corresponding validation subset. We report the RMSE on the entire California validation set in Table 4.

Table 4: RMSE $\times 100$ of each entity on the California validation set.

| Entity | 1 | 2 | 3 | 4 | 5 | 6 |
|---|---|---|---|---|---|---|
| RMSE $\times 100$ | 21.97 | 15.72 | 31.81 | 16.20 | 27.06 | 40.26 |

**Top-$k$ L2D.** We train a two-layer MLP following Algorithm 1. The rejector is trained for 100 epochs with a batch size of 256, a learning rate of $5 \times 10^{-4}$, using the Adam optimizer and a cosine learning rate scheduler. We select the checkpoint that achieves the lowest Top-$k$ surrogate loss on the validation set, yielding the final rejector $r$. We report Top-$k$ L2D performance for each fixed value $k \in \mathcal{A}$.

**Top-$k(x)$ L2D.** We train the cardinality function using the same two-layer MLP architecture, following Algorithm 2. The cardinality function is also trained for 100 epochs with a batch size of 256, a learning rate of $5 \times 10^{-4}$, using Adam and cosine scheduling. We conduct additional experiments using various instantiations of the metric $d$, as detailed in Section B.1.1.

**Performance Comparison.** Figures 7, compare Top-$k$ and Top-$k(x)$ L2D across multiple evaluation metrics and budget regimes. Top-$k$ L2D consistently outperforms random baselines and closely approaches the oracle (optimal) strategy under the $\text{RMSE}_{\text{min}}$ metric, validating the benefit of using different entities (Table 4).

In Figure 7b, Top-$k(x)$ achieves near-optimal performance (6.23) with a budget of $\overline{\beta} = 0.156$ and an expected number of entities $\overline{k} = 4.77$, whereas Top-$k$ requires the full budget $\overline{\beta} = 0.2$ and $\overline{k} = 6$ entities to reach a comparable score (6.21). This demonstrates the ability of Top-$k(x)$ to allocate resources more efficiently by querying only the necessary number of entities, in contrast to Top-$k$, which tends to over-allocate costly or redundant experts. Additionally, our approach outperforms the Top-1 L2D baseline (Mao et al., 2024c), confirming the limitations of single-entity deferral.

Figures 7c and 7d evaluate Top-$k$ and Top-$k(x)$ L2D under more restrictive metrics—$\text{RMSE}_{\text{avg}}$ and $\text{RMSE}_{\text{w-avg}}$—where performance is not necessarily monotonic in the number of queried entities. In these settings, consulting too many or overly expensive entities may degrade overall performance. Top-$k(x)$ consistently outperforms Top-$k$ by carefully adjusting the number of consulted entities. In both cases, it achieves optimal performance with a budget of only $\overline{\beta} = 0.095$, a level that Top-$k$ fails to reach. For example, in Figure 7c, Top-$k(x)$ achieves $\text{RMSE}_{\text{avg}} = 8.53$, compared to 10.08 for Top-$k$. Similar trends are observed under the weighted average metric (Figure 7d), where Top-$k(x)$ again outperforms Top-$k$, suggesting that incorporating rejector-derived weights $w_j$ leads to more effective aggregation. This demonstrates that our Top-$k(x)$ L2D selectively chooses the appropriate entities—when necessary—to enhance the overall system performance.

### B.3.2 RESULTS ON SVHN

**Settings.** We construct a pool of 6 convolutional neural networks (CNNs), each trained on a randomly sampled, partially overlapping subset of the SVHN dataset (20%). This setup simulates realistic settings where entities are trained on distinct data partitions due to privacy constraints or

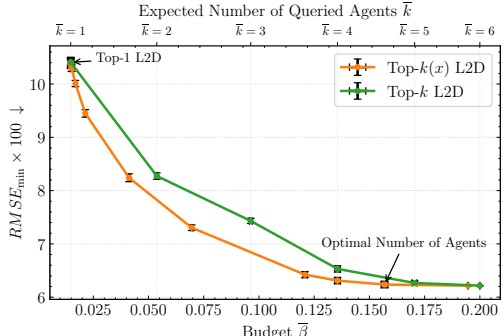

(a) Comparison with Random and Optimal Baselines: We compare Top-$k$ L2D against random and oracle baselines using the RMSE$_\text{min}$ metric defined in Section B.1.1. Our method consistently outperforms both baselines and extends the Top-1 L2D formulation introduced by Mao et al. (2024c).

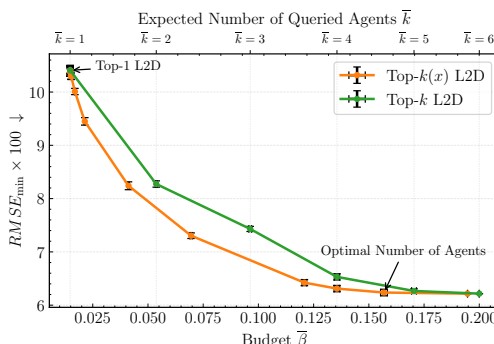

(b) Comparison of Top-$k$ L2D and Top-$k(x)$ L2D under RMSE$_\text{min}$ metric. We report the RMSE$_\text{min}$ metric, along with the budget and the expected number of queried entities. Across all budgets, Top-$k(x)$ consistently outperforms Top-$k$ L2D by achieving lower error with fewer entities and reduced cost.

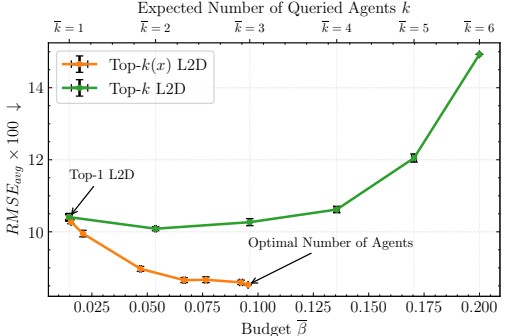

(c) Comparison of Top-$k$ L2D and Top-$k(x)$ L2D under RMSE$_\text{avg}$ metric.

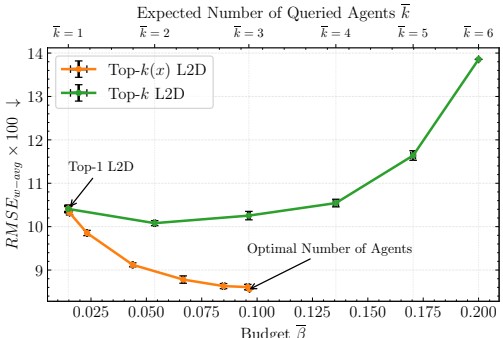

(d) Comparison of Top-$k$ L2D and Top-$k(x)$ L2D under RMSE$_\text{w-avg}$ metric.

Figure 7: Results on the California dataset comparing Top-$k$ and Top-$k(x)$ L2D across four evaluation metrics. Top-$k(x)$ consistently achieves superior performance across all trade-offs. The Top-1 L2D corresponds to Narasimhan et al. (2022); Mao et al. (2024c).

institutional data siloing. As a result, the entities exhibit heterogeneous predictive capabilities and error patterns.

Each entity is trained for 3 epochs using the Adam optimizer (Kingma & Ba, 2014), with a batch size of 64 and a learning rate of $1 \times 10^{-3}$. Model selection is performed based on the lowest loss on each entity's respective validation subset. The table 5 below reports the classification accuracy of each trained entity, evaluated on a common held-out validation set:

Table 5: Accuracy of each entity on the SVHN validation set.

| Entity | 1 | 2 | 3 | 4 | 5 | 6 |
|---|---|---|---|---|---|---|
| Accuracy (%) | 63.51 | 55.53 | 61.56 | 62.60 | 66.66 | 64.26 |

**Top-$k$ L2D.** We train the rejector using a ResNet-4 architecture (He et al., 2016), following Algorithm 1. The model is trained for 50 epochs with a batch size of 256 and an initial learning rate of $1 \times 10^{-2}$, scheduled via cosine annealing. Optimization is performed using the Adam optimizer. We select the checkpoint that minimizes the Top-$k$ surrogate loss on the validation set, yielding the final rejector $r$. We report Top-$k$ L2D performance for each fixed value $k \in \mathcal{A}$.

**Top-$k(x)$ L2D.** We reuse the trained rejector $r$ and follow Algorithm 2 to train a cardinality cardinality function $s \in \mathcal{S}$. The cardinality function is composed of a CLIP-based feature extractor (Radford et al., 2021) and a lightweight classification head. It is trained for 10 epochs with a batch size of 256, a learning rate of $1 \times 10^{-3}$, weight decay of $1 \times 10^{-5}$, and cosine learning rate scheduling. We use the AdamW optimizer (Loshchilov & Hutter, 2017) for optimization.

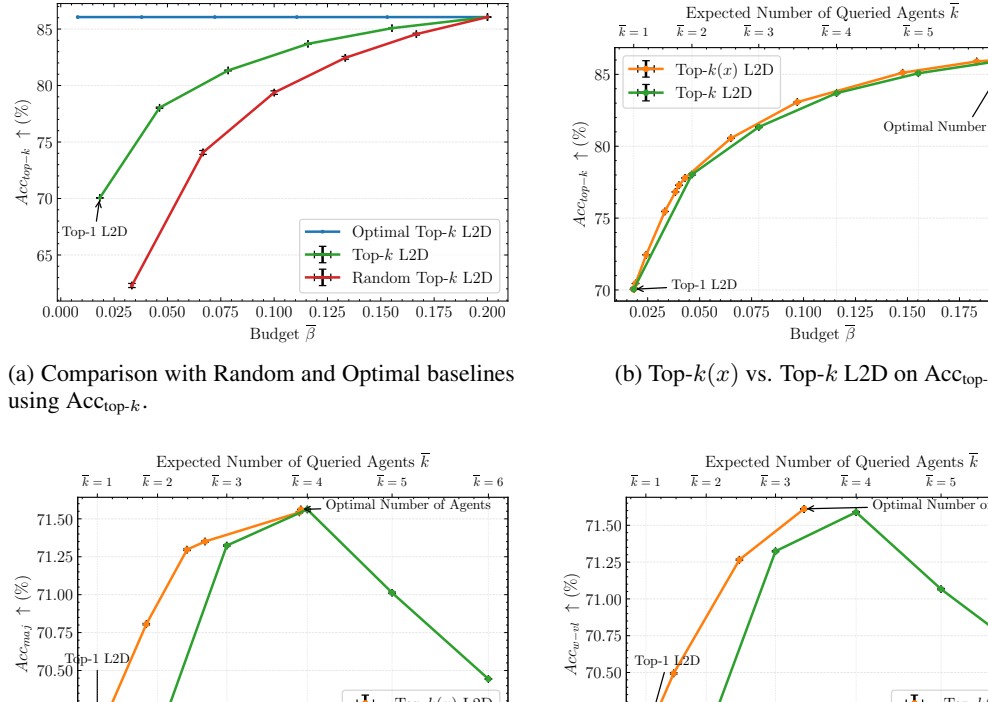

(a) Comparison with Random and Optimal baselines using $\mathrm{Acc}_{\text{top-}k}$.

(b) Top-$k(x)$ vs. Top-$k$ L2D on $\mathrm{Acc}_{\text{top-}k}$.

(c) Performance under $\mathrm{Acc}_{\text{maj}}$ metric.

(d) Performance under $\mathrm{Acc}_{\text{w-vl}}$ metric.

Figure 8: Comparison of Top-$k$ and Top-$k(x)$ L2D across four accuracy metrics on SVHN. Top-$k(x)$ achieves better budget-accuracy trade-offs across all settings. The Top-1 L2D corresponds to Montreuil et al. (2025b).

**Performance Comparison.** Figure 8 compares our *Top-k* and *Top-k(x)* L2D approaches against prior work (Narasimhan et al., 2022; Mao et al., 2023a), as well as oracle and random baselines. As shown in Figure 8a, querying multiple entities significantly improves performance, with both of our methods surpassing the Top-1 L2D baselines (Narasimhan et al., 2022; Mao et al., 2023a). Moreover, our learned deferral strategies consistently outperform the random L2D baseline, underscoring the effectiveness of our allocation policy in routing queries to appropriate entities. In Figure 8b, Top-$k(x)$ L2D consistently outperforms Top-$k$ L2D, achieving better accuracy under the same budget constraints.

For more restrictive metrics, Figures 8c and 8d show that Top-$k(x)$ achieves notably stronger performance, particularly in the low-budget regime. For example, in Figure 8c, at a budget of $\overline{\beta} = 0.41$, Top-$k(x)$ attains $\mathrm{Acc}_{\text{maj}} = 70.81$, compared to $\mathrm{Acc}_{\text{maj}} = 70.05$ for Top-$k$. This performance gap widens further at smaller budgets. Both figures also highlight that querying too many entities may degrade accuracy due to the inclusion of low-quality predictions. In contrast, Top-$k(x)$ identifies a better trade-off, reaching up to $\mathrm{Acc}_{\text{maj}} = 71.56$ under majority voting and $\mathrm{Acc}_{\text{w-vl}} = 71.59$ with weighted voting. As in the California Housing experiment, weighted voting outperforms majority voting, suggesting that leveraging rejector-derived weights improves overall decision quality.

### B.3.3 RESULTS ON CIFAR100.

**entity Settings.** We construct a pool of 6 entities. We train a main predictor (entity 1) using a ResNet-4 (He et al., 2016) for 50 epochs, a batch size of 256, the Adam Optimizer (Kingma & Ba, 2014) and select the checkpoints with the lower validation loss. We synthetically create 5 experts with strong overlapped knowledge. We assign experts to classes for which they have the probability to be correct reaching $p = 0.94$ and uniform in non-assigned classes. Typically, we assign 55 classes to each experts. We report in the Table 6 the accuracy of each entity on the validation set.

Table 6: Accuracy of each entity on the CIFAR100 validation set.

| Entity | 1 | 2 | 3 | 4 | 5 | 6 |
|---|---|---|---|---|---|---|
| Accuracy (%) | 59.74 | 51.96 | 52.58 | 52.21 | 52.32 | 52.25 |

**Top-$k$ L2D.** We train the rejector model using a ResNet-4 architecture (He et al., 2016), following the procedure described in Algorithm 1. The model is optimized using Adam with a batch size of 2048, an initial learning rate of $1 \times 10^{-3}$, and cosine annealing over 200 training epochs. We select the checkpoint that minimizes the Top-$k$ surrogate loss on the validation set, resulting in the final rejector $r$. We report Top-$k$ L2D performance for each fixed value $k \in \mathcal{A}$.

**Top-$k(x)$ L2D.** We reuse the learned rejector $r$ and train a cardinality cardinality function $s \in \mathcal{S}$ as described in Algorithm 2. The cardinality function is composed of a CLIP-based feature extractor (Radford et al., 2021) and a lightweight classification head. It is trained using the AdamW optimizer (Loshchilov & Hutter, 2017) with a batch size of 128, a learning rate of $1 \times 10^{-3}$, weight decay of $1 \times 10^{-5}$, and cosine learning rate scheduling over 15 epochs. To evaluate performance under different decision rules, we conduct experiments using multiple instantiations of the metric $d$; detailed definitions and evaluation protocols are provided in Section B.1.1.

**Performance Comparison.** Figure 9b shows that Top-$k$ L2D outperforms random query allocation, validating the benefit of learned deferral policies. As shown in Figure 9a, Top-$k(x)$ further improves performance over Top-$k$ by adaptively selecting the number of entities per query. In Figures 9c and 9d, Top-$k(x)$ consistently yields higher accuracy across all budget levels, achieving significant gains over fixed-$k$ strategies.

Notably, unlike in other datasets, querying additional entities in this setting does not degrade performance. This is due to the absence of low-quality entities: each entity predicts correctly with high probability (at least 94%) on its assigned class subset. As a result, aggregating predictions from multiple entities improves accuracy by selectively querying them.

Nevertheless, Top-$k(x)$ remains advantageous due to the overlap between entities and their differing consultation costs. When several entities are likely to produce correct predictions, it is preferable to defer to the less costly one. By exploiting this flexibility, Top-$k(x)$ achieves a large performance improvement over Top-$k$ L2D while also reducing the overall budget.

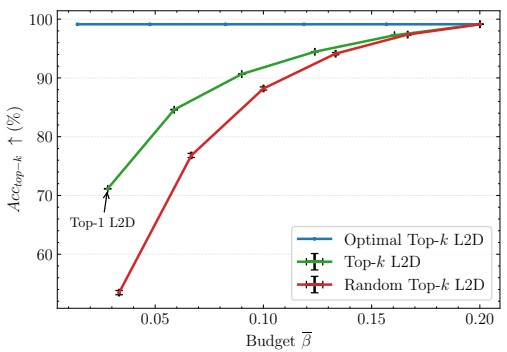

(a) Comparison with Random and Optimal baselines using $Acc_{\text{top-}k}$.

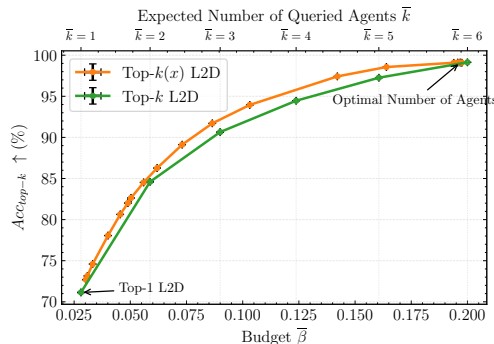
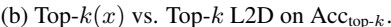

(b) Top-$k(x)$ vs. Top-$k$ L2D on $Acc_{\text{top-}k}$.

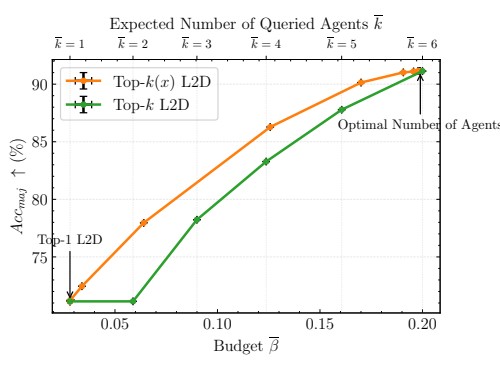

(c) Performance under $Acc_{\text{maj}}$ metric.

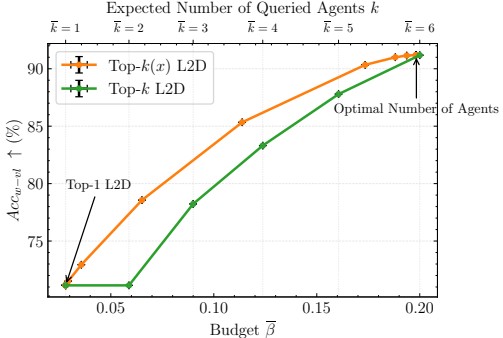

(d) Performance under $Acc_{\text{w-vl}}$ metric.

Figure 9: Comparison of Top-$k$ and Top-$k(x)$ L2D across four accuracy metrics on CIFAR100. Top-$k(x)$ achieves better budget-accuracy trade-offs across all settings. The Top-1 L2D corresponds to Narasimhan et al. (2022); Mao et al. (2023a).

