# OpenReview forum: "Why Ask One When You Can Ask $k$? Learning-to-Defer to the Top-$k$ Experts"
_ICLR.cc/2026/Conference — ICLR 2026 Poster_

### Official Review · Reviewer_U2o1 · 2025-10-18

**Soundness:** 3
**Presentation:** 4
**Contribution:** 3
**Rating:** 6
**Confidence:** 3

**Summary:**

This paper proposes Top - $k$ Learning-to-Defer (L2D), a novel framework that allows a model to defer each query to a set of $k$ experts instead of a single expert. This generalizes all prior L2D approaches, which were limited to one expert (“Top-1”) deferrals. The authors formulate a unified approach covering both major paradigms of L2D: in the one-stage regime, the predictor and deferral policy are learned jointly in one model, and in the two-stage regime, experts and a main predictor are trained offline and a separate routing function learns to allocate queries. Crucially, the framework recovers the traditional Top-1 deferral as a special case while enabling principled collaboration with multiple experts when $k>1$.  In addition, the paper introduces Top-$k(x)$ L2D, an adaptive extension where the system learns to choose a query-specific number of experts (up to $|A|$, the total pool) based on factors like input difficulty, expert accuracy, and consultation cost. To support both fixed-$k$ and adaptive deferral, the authors develop a $k$-independent surrogate loss that facilitates practical training.

**Strengths:**

- Innovative Multi-Expert Deferral Framework: This is the first work to allow deferring to multiple experts in a Learning-to-Defer setting, addressing a real-world gap. The idea is original and expands the scope of L2D beyond the restrictive one-expert paradigm. By formulating Top-$k$ deferral, the paper enables more robust decision-making (aggregating complementary expertise) which is crucial for complex tasks. This multi-expert allocation capability is a direct answer to the stated limitation of prior work (that relying on only one expert can amplify bias or error). The framework cleanly unifies both one-stage and two-stage L2D, showing broad applicability across different training setups. It also strictly generalizes previous approaches: for example, the authors show that setting $k=1$ recovers the standard L2D solutions in both regimes, and even classical model cascades and selective prediction emerge as special cases of their formulation.

- Theoretical Foundations: The paper provides theoretical analysis and guarantees. The introduction of a $k$-independent surrogate loss function is a technical strength. It is proven to be Bayes-consistent, meaning that as the amount of training data grows, the learned deferral policy will converge to the Bayes-optimal top-$k$ strategy (the one that minimizes true expected cost). Furthermore, the surrogate is shown to maintain H-consistency within restricted hypothesis classes providing bounds on the excess risk when using finite-capacity models. These results extend known consistency bounds from $k=1$ to the multi-expert scenario. The theoretical development is non-trivial and well-executed.

- Empirical Effectiveness: The proposed methods show empirical advantages over prior approaches. Across multiple benchmarks, Top-$k$ deferral improves the trade-off between prediction accuracy and consultation cost. Notably, Top-$k(x)$ (adaptive) is very effective: it achieves comparable or better accuracy with significantly lower average cost by varying the number of experts per query. For instance, in one experiment the adaptive policy reached the same RMSE as a fixed $k$ policy while querying ~1.3 fewer experts on average (saving ~25% of the budget). This demonstrates efficient use of experts. The results also confirmed that even a fixed Top-$k$ (with an appropriate $k$) outperforms the traditional Top-1 deferral baseline in accuracy, under the same cost budget. Overall, the authors back up their claims with comprehensive experiments in both one-stage (e.g. CIFAR-10 image classification with learned deferral) and two-stage (e.g. regression with a separate main model and experts) scenarios.

**Weaknesses:**

- Complexity of Training & Implementation: One potential weakness is the added complexity of the proposed approach, especially the adaptive Top-$k(x)$ mechanism. The framework introduces an additional model (the cardinality function $k_\theta(x)$) that must be learned alongside the main deferral policy. This effectively means training two interconnected components: one to score and select experts, and another to decide how many to select. The paper defers some of the training details to the appendix (Algorithms 1 and 2 outline separate procedures for the policy and the cardinality function), but it’s not entirely clear from the main text whether these components are trained jointly or sequentially, and how hyperparameters (especially the regularization $\lambda$ in $\ell_{\text{card}}$) are chosen. The need to tune $\lambda$ to balance accuracy vs. cost could require expert knowledge or cross-validation on a utility metric, which adds complexity. In practice, implementing Top-$k(x)$ may be more involved than standard L2D due to this bi-level optimization (learn policy given a fixed $k$, and learn the $k$-selection function), a point the authors could clarify further.

- Evaluation Scope – Lack of Real-World Expert Data: The experiments, while thorough on synthetic and benchmark data, do not include a real-world multi-expert scenario, which slightly limits the demonstration of the framework’s practical value. All evaluations seem to use either simulated experts (e.g., classifiers with predetermined accuracy on subsets of classes) or machine models as stand-in “experts.” While this is common in L2D research (since obtaining multiple human experts’ responses can be difficult), it would have been insightful to see at least one case with actual human experts or a real ensemble to validate that Top-$k$ deferral works outside the lab setting.

- Assumptions for Theoretical Guarantees: The consistency guarantees, while impressive, rely on certain assumptions that may not hold in practice. For example, the two-stage consistency assumes that the Bayes-optimal cost can be approached arbitrarily well by the model classes. Essentially this is a realizability or universal approximation assumption for the hypothesis spaces. If the chosen model architectures are misspecified or capacity-limited, there could be a minimizability gap where the learned policy cannot reach the true optimum. The authors do acknowledge this, but in practice one never knows if one’s model is rich enough. Thus, the Bayes-consistency is a bit theoretical, at finite sample sizes with imperfect models, one might not achieve the optimal deferral policy. The paper does not discuss the finite-sample performance beyond asymptotic consistency; for instance, how fast is the convergence or how the excess risk scales with $k$. Some empirical calibration of the surrogate (like comparing surrogate risk vs true deferral risk on validation data) could help validate that the loss is working as intended in finite data regimes.

**Questions:**

- Training Strategy for $k$-Selection: Could you elaborate on how the deferral policy $\pi(x,\cdot)$ and the cardinality function $k_\theta(x)$ are trained in practice? Was the $k$-adaptive model trained in an end-to-end fashion, or did you first learn a Top-$|A|$ policy and then learn the cardinality predictor (or vice versa)?

- Decision Rule for Multiple Experts’ Outputs: In a deployment, once the Top-$k$ experts are consulted for a query, how is their collective response used to make a final prediction? In your experiments, what rule did you use to decide the final output or correctness from multiple predictions? For example, for CIFAR-10 with $k=2$ experts selected, did you consider the query correct if either expert (or the model) got the label right (i.e. at least one correct prediction in the set), or did you require agreement/majority? Different aggregation strategies (e.g., majority vote vs. oracle-style “at least one correct”) can yield different accuracy metrics. It would clarify your evaluation if you specify the aggregation/decision procedure for multiple expert opinions.

- Sensitivity to Cost Hyperparameters ($\alpha,\beta,\lambda$): The framework introduces cost parameters $\alpha_j$ and $\beta_j$ for each expert (measuring error penalty vs. consultation cost), as well as a regularization weight $\lambda$ in the adaptive loss. How sensitive are your results to the choice of these values? In practice, one might not know the exact cost of consulting an expert or might have to guess a trade-off parameter. Did you perform any analysis or ablation where you vary $\beta_j$ or $\lambda$ to see how the deferral behavior changes? For instance, if all $\beta_j$ are set to 0 (ignoring cost), does the policy simply always choose all experts (maximizing accuracy)? And conversely, if $\beta_j$ are very high, does it revert to always using the model (to save cost)?

---

> ### Author Response · Authors · 2025-11-18
> **Response Reviewer U2o1**
>
> We thank the reviewer for their detailed and constructive feedback, and for highlighting the **(i) novelty of our multi-expert framework, the (ii) strength of our theoretical guarantees, and (iii) the empirical advantages demonstrated in both one-stage and two-stage settings.**
>
> > Complexity of Training and Implementation: One potential weakness is the added complexity of the proposed approach, especially the adaptive Top-$k(x)$ mechanism. The framework introduces an additional model (the cardinality function $k_{\theta}(x)$) that must be learned alongside the main deferral policy. This effectively means training two interconnected components: one to score and select experts, and another to decide how many to select. The paper defers some of the training details to the appendix (Algorithms 1 and 2 outline separate procedures for the policy and the cardinality function), but it’s not entirely clear from the main text whether these components are trained jointly or sequentially, and how hyperparameters (especially the regularization $\lambda$ in $\ell_{card}$) are chosen. The need to tune $\lambda$ to balance accuracy vs. cost could require expert knowledge or cross-validation on a utility metric, which adds complexity. In practice, implementing Top-$k(x)$ may be more involved than standard L2D due to this bi-level optimization (learn policy given a fixed $k$, and learn the $k$-selection function), a point the authors could clarify further.
>
> > Training Strategy for $k$-Selection: Could you elaborate on how the deferral policy $\pi(x,\cdot)$ and the cardinality function $k_\theta(x)$ are trained in practice? Was the $k$-adaptive model trained in an end-to-end fashion, or did you first learn a Top-$k$ policy and then learn the cardinality predictor (or vice versa)?
>
> The cardinality function $k\_{\theta}(x)$ is learned post-hoc, after training the main deferral policy $\pi(x,\cdot)$. **This decoupling keeps optimization simple and stable, avoiding any bi-level or joint training, as well as guaranteeing Bayes and $\mathcal{H}$-consistency**. The procedure consists of first learning $\pi$ with a fixed $k$, and then training $k\_{\theta}$ to predict the appropriate cardinality. This sequential design makes the implementation straightforward and computationally efficient.
>
> **The regularization parameter $\lambda$ in $\ell\_{\mathrm{card}}$ directly controls the budget–accuracy trade-off. It can be tuned according to a desired average number of consulted experts, using a simple sweep—no more complex than standard hyperparameter tuning for classification**. Increasing $\lambda$ favors cheaper but less redundant decisions, while decreasing it prioritizes reliability at higher consultation cost (see also our discussion with Reviewer @VC3c). We have clarified in the main text that $\pi$ and $k\_{\theta}$ are trained sequentially and briefly outline the tuning procedure.
>
> > Evaluation Scope – Lack of Real-World Expert Data: The experiments, while thorough on synthetic and benchmark data, do not include a real-world multi-expert scenario, which slightly limits the demonstration of the framework’s practical value. All evaluations seem to use either simulated experts (e.g., classifiers with predetermined accuracy on subsets of classes) or machine models as stand-in “experts.” While this is common in L2D research (since obtaining multiple human experts’ responses can be difficult), it would have been insightful to see at least one case with actual human experts or a real ensemble to validate that Top-$k$ deferral works outside the lab setting.
>
> **We agree that evaluating with human experts would be valuable, but to our knowledge, no public dataset provides multiple, distinct expert annotations suitable for Learning-to-Defer training. Consequently, all prior L2D works rely on simulated or model-based experts [1–4]**.
>
> **Our experimental design intentionally reflects this standard but under more demanding conditions**: experts exhibit heterogeneous, class-conditional accuracies, overlapping expertise, and correlated errors—scenarios that are often \emph{harder} than typical human-in-the-loop settings, where expertise is more complementary and less adversarial. We also include experiments where distinct trained models act as experts [5,6], further validating that the framework generalizes beyond synthetic setups.

---

> ### Author Response · Authors · 2025-11-18
> **Response 2 Reviewer U2o1**
>
> > Assumptions for Theoretical Guarantees: The consistency guarantees, while impressive, rely on certain assumptions that may not hold in practice. For example, the two-stage consistency assumes that the Bayes-optimal cost can be approached arbitrarily well by the model classes. Essentially this is a realizability or universal approximation assumption for the hypothesis spaces. If the chosen model architectures are misspecified or capacity-limited, there could be a minimizability gap where the learned policy cannot reach the true optimum. The authors do acknowledge this, but in practice one never knows if one’s model is rich enough. Thus, the Bayes-consistency is a bit theoretical, at finite sample sizes with imperfect models, one might not achieve the optimal deferral policy. The paper does not discuss the finite-sample performance beyond asymptotic consistency; for instance, how fast is the convergence or how the excess risk scales with $k$. Some empirical calibration of the surrogate (like comparing surrogate risk vs true deferral risk on validation data) could help validate that the loss is working as intended in finite data regimes.
>
> **We would like to clarify that our theoretical guarantees do not rely on the realizability or universal approximation assumption**. This is precisely why we establish $\mathcal{H}$-consistency rather than Bayes-consistency: the former remains valid even when the hypothesis class $\mathcal{H}\_\pi$ has limited capacity or is misspecified. In other words, our results quantify how closely the learned policy $\pi$ approaches the optimal within its class, without requiring the Bayes rule to be attainable. The assumption that the Bayes-optimal cost can be approached arbitrarily well is used only to illustrate the asymptotic intuition, not to make the bounds hold.
>
> Moreover, our $\mathcal{H}$-consistency analysis already provides an explicit convergence rate linking the surrogate and true deferral risks. For the logistic surrogate, we obtain:
>
> $$ \mathcal{E}\_{\ell\_{\mathrm{def},k}}(h)-\mathcal{E}\_{\ell\_{\mathrm{def},k}}^{B}(\mathcal{H}\_h)+\mathcal{U}\_{\ell\_{\mathrm{def},k}}(\mathcal{H}\_h)\le kS\sqrt{2\left(\frac{\mathcal{E}\_{\Phi\_{\mathrm{def},k}^u}(h)-\mathcal{E}\_{\Phi\_{\mathrm{def},k}^u}^{\ast}(\mathcal{H}\_h)+\mathcal{U}\_{\Phi\_{\mathrm{def},k}^u}(\mathcal{H}\_h)}{S}\right)}. $$
>
> **This bound explicitly captures the finite-sample scaling—showing that the excess risk contracts at rate $\mathcal{O}(kS\sqrt{t})$**—and thus characterizes convergence speed without requiring realizability. We have clarified this in the updated version.
>
> > Decision Rule for Multiple Experts’ Outputs: In a deployment, once the Top-$k$ experts are consulted for a query, how is their collective response used to make a final prediction? In your experiments, what rule did you use to decide the final output or correctness from multiple predictions? For example, for CIFAR-10 with $k=2$ experts selected, did you consider the query correct if either expert (or the model) got the label right (i.e., at least one correct prediction in the set), or did you require agreement/majority? Different aggregation strategies (e.g., majority vote vs. oracle-style “at least one correct”) can yield different accuracy metrics. It would clarify your evaluation if you specify the aggregation/decision procedure for multiple expert opinions.
>
> **As described in the main paper, Fig. 1, Line 430, and Appendix A.9, the aggregation mechanism $d$ used after consulting the Top-$k$ experts is explicitly defined and can be chosen freely in our framework**. In Definition 5.1, $d$ determines how multiple predictions are combined when evaluating the loss. For example, if $d$ corresponds to the Top-$k$ true loss (Line 1624), a query is considered correct if any consulted expert predicts correctly; if $d$ is majority voting (Line 1636) or weighted voting (Line 1628), the final output depends on collective agreement or confidence weighting. **In our experiments, we have already reported results for all these aggregation mechanisms to make the evaluation comprehensive and reproducible. Please see Figures 1, 5, 6, 7, 8, and 9.**

---

> > ### Author Response · Authors · 2025-11-18
> > **Response 3 to Reviewer U2o1**
> >
> > > The framework introduces cost parameters $\alpha_j$ and $\beta_j$ for each expert (measuring error penalty vs.\ consultation cost), as well as a regularization weight $\lambda$ in the adaptive loss. How sensitive are your results to the choice of these values? In practice, one might not know the exact cost of consulting an expert or might have to guess a trade-off parameter. Did you perform any analysis or ablation where you vary $\beta_j$ or $\lambda$ to see how the deferral behavior changes? For instance, if all $\beta_j$ are set to $0$ (ignoring cost), does the policy simply always choose all experts (maximizing accuracy)? And conversely, if $\beta_j$ are very high, does it revert to always using the model (to save cost)?
> >
> > The consultation cost $\beta_j$ is user-defined and reflects the expense of querying expert $j$ (e.g., compute, latency, or human time). **When $\beta_j=0$, the system can freely consult any expert to maximize accuracy (w.r.t a specific aggregation); however, the learned $k_{\theta}$ does not necessarily select all experts, since including unreliable ones may lead to wrong final prediction (Def. 5.1) and then degrade performance—hence the non-monotonic behavior observed in Fig. 1b.**
> >
> > **When $\beta_j$ is large—but still smaller than the cost of an error—the system prioritizes experts minimizing the total expected cost (error + consultation)**. The regularization coefficient $\lambda$ governs the global trade-off: higher $\lambda$ reduces the number of consulted experts (favoring spending other reliability), while smaller $\lambda$ favors accuracy over cost. As shown in Fig. 1, varying $\lambda$ produces a consistent trend across the three aggregation mechanisms, confirming that the behavior is stable and interpretable.
> >
> >
> > ----
> >
> > [1] Mozannar, Hussein, and David Sontag. "Consistent estimators for learning to defer to an expert." International conference on machine learning. PMLR, 2020.
> >
> > [2] Verma, Rajeev, Daniel Barrejón, and Eric Nalisnick. "Learning to defer to multiple experts: Consistent surrogate losses, confidence calibration, and conformal ensembles." International Conference on Artificial Intelligence and Statistics. PMLR, 2023.
> >
> > [3] Cao, Yuzhou, et al. "In defense of softmax parametrization for calibrated and consistent learning to defer." Advances in Neural Information Processing Systems 36 (2023): 38485-38503.
> >
> > [4] Mozannar, Hussein, et al. "Who should predict? exact algorithms for learning to defer to humans." International conference on artificial intelligence and statistics. PMLR, 2023.
> >
> > [5] Mao, Anqi, et al. "Two-stage learning to defer with multiple experts." Advances in neural information processing systems 36 (2023): 3578-3606.
> >
> > [6] Tailor, Dharmesh, et al. "Learning to defer to a population: A meta-learning approach." International Conference on Artificial Intelligence and Statistics. PMLR, 2024.

---

> > > ### Comment · Reviewer_U2o1 · 2025-11-21
> > >
> > > Thank you for the detailed and thoughtful responses. I appreciate the clarifications regarding the training procedure, aggregation strategies, and the theoretical assumptions. These help improve the clarity of the work. While the rebuttal addressed several points, my core concerns about practical complexity and the absence of real-world multi-expert evaluations remain only partially resolved. For this reason, I will keep my original score.

---

### Official Review · Reviewer_mQGW · 2025-10-21

**Soundness:** 3
**Presentation:** 2
**Contribution:** 3
**Rating:** 6
**Confidence:** 2

**Summary:**

Learning-to-defer (L2D) deals with utilizing expert advice for queries. Existing approaches only consider using a single expert, and this paper proposes a loss that works for multiple experts and satisfies desirable consistency objectives. Further, the author shows that prior setups are special cases of the novel approach. Experiments are provided to justify the theoretical results.

**Strengths:**

- The paper is well motivated and contextualized. For instance, it is good that the authors mention why the naive extension of the top-1 selection doesn't work (line 216).
- Extensive experiments are provided, and description of setup is very detailed.

**Weaknesses:**

- Some of the notation is kind of cumbersome, eg the superscripted down and up-arrows. Also, hat seems to be used for many things: argmax (eg line 115), experts, and predictions (lines 134-135). It would be better to use something like asterisk for argmax, eg $h^* = \arg\max h_i$
- It looks like the proof of Lemma A.6 is incorrect/incomplete, as eq 15 is directly assuming the implication.

**Questions:**

Overall, the presentation could use significant work. For instance:
- Can $\mathcal{Y}$ be defined at the beginning of the preliminaries? More generally, please provide a more exhaustive discussion of notation, eg. either in the preliminaries, or as a table in the appendix.
- What is the definition of consistency, calibration? It would be good to add a section in the appendix for additional preliminaries to make all of this self-contained.

Additional:
- Is top-k(x) also consistent (as in Theorem 4.7)?
- In Figure 5, is there intuition why accuracy for top-k isn't monotonic increasing (in budget) whereas top-k(x) is?
- wording in line 400 is a little unclear: minimizability gap... vanishes... if the hypothesis set is as rich as the set of all measurable functions

---

> ### Author Response · Authors · 2025-11-18
> **Response Reviewer mQGW**
>
> We appreciate the reviewer’s positive assessment of our motivation, contextualization, and experimental thoroughness.
>
> > Some of the notation is kind of cumbersome, eg the superscripted down and up-arrows. Also, hat seems to be used for many things: argmax (eg line 115), experts, and predictions (lines 134-135). It would be better to use something like asterisk for argmax, eg $h^\ast=\arg\max {h}_i$.
>
> We appreciate the suggestion. The up- and down-arrow superscripts are intentional—they indicate different orderings used respectively in Definition 4.1 and Lemma 4.5, which are not interchangeable and thus must be distinguished. The hat symbol $\hat{\cdot}$ denotes variables in the output space $\mathcal{Z}$ derived from $\mathcal{X}$ (e.g., predictions), not only argmax operators.
>
> We have clarified this and added a notation section in Appendix A1 and A2.
>
> > It looks like the proof of Lemma A.6 is incorrect/incomplete, as eq 15 is directly assuming the implication.
>
> The proof is actually complete and correct; we will make the intermediate implication in Eq. (15) explicit. Let $p\_j(x)=\mathbb{P}(Y=j\mid X=x)$ and
>
> $$ \mathcal{C}\_{\ell\_k}(\pi,x)=\sum\_{j\in\mathcal{A}} p\_j(x)\mathbf{1}\\{j\notin \Pi\_k(x)\\},\qquad \mathcal{C}\_{\Phi}(\pi,x)=\sum\_{j\in\mathcal{A}} p\_j(x)\Phi^{u}\_{01}(\pi,x,j). $$
>
> By the calibration inequality (Def. 5.1), for each $x$,
>
> $$ \Delta\mathcal{C}\_{\ell\_k}(\pi,x):=\mathcal{C}\_{\ell\_k}(\pi,x)-\mathcal{C}^B\_{\ell\_k}(x)\le k\Gamma\_u^{-1}(\Delta\mathcal{C}\_{\Phi}(\pi,x)). $$
>
> Taking expectation over $X$ and using Jensen’s inequality for concave $\Gamma\_u^{-1}$ gives
>
> $$ \mathbb{E}\_X[\Delta\mathcal{C}\_{\ell\_k}(\pi,X)]\le k\Gamma\_u^{-1}(\mathbb{E}\_X[\Delta\mathcal{C}\_{\Phi}(\pi,X)]). $$
>
> By definition,
>
> $$ \mathbb{E}\_X[\Delta\mathcal{C}\_{\ell\_k}(\pi,X)]=\mathcal{E}\_{\ell\_k}(\pi)-\mathcal{E}^{\ast}\_{\ell\_k}(\mathcal{H}\_\pi)+\mathcal{U}\_{\ell\_k}(\mathcal{H}\_\pi), $$
>
> and similarly,
>
> $$ \mathbb{E}\_X[\Delta\mathcal{C}\_{\Phi}(\pi,X)]=\mathcal{E}\_{\Phi}(\pi)-\mathcal{E}^{\ast}\_{\Phi}(\mathcal{H}\_\pi)+\mathcal{U}\_{\Phi}(\mathcal{H}\_\pi). $$
>
> Hence,
>
> $$ \mathcal{E}\_{\ell\_k}(\pi)-\mathcal{E}^B\_{\ell\_k}(\mathcal{H}\_\pi)-\mathcal{U}\_{\ell\_k}(\mathcal{H}\_\pi)\le k\Gamma\_u^{-1}(\mathcal{E}\_{\Phi}(\pi)-\mathcal{E}^{\ast}\_{\Phi}(\mathcal{H}\_\pi)-\mathcal{U}\_{\Phi}(\mathcal{H}\_\pi)). $$
>
> This establishes the inequality in Lemma A.6.
>
> > Can $\mathcal{Y}$ be defined at the beginning of the preliminaries? More generally, please provide a more exhaustive discussion of notation, eg. either in the preliminaries, or as a table in the appendix.
>
> The notation $\mathcal{Y}$ is already defined in the preliminaries (l.106). We have added this in a new notation section A1 and A2.
>
> >  What is the definition of consistency, calibration? It would be good to add a section in the appendix for additional preliminaries to make all of this self-contained.
>
> We have revised the manuscript and included these definitions in Appendix A3.
>
> Given a hypothesis set $\mathcal{H}$, we say that a loss $\ell\_{def,k}$ is $\mathcal{H}$-consistent with respect to the loss $\Phi\_{def}$ if
>
> $$ \mathcal{E}\_{\Phi\_{def}}(h\_n)-\mathcal{E}^{\ast}\_{\Phi\_{def}}(\mathcal{H})+\mathcal{U}\_{\Phi\_{def}}(\mathcal{H})\xrightarrow[n\to\infty]{}0\ \Longrightarrow\ \mathcal{E}\_{\ell\_{def,k}}(h\_n)-\mathcal{E}^{B}\_{\ell\_{def,k}}(\mathcal{H})+\mathcal{U}\_{\ell\_k}(\mathcal{H})\xrightarrow[n\to\infty]{}0. $$
>
> Given a hypothesis set $\mathcal{H}$, we say that a loss $\Phi\_{def}$ is $\mathcal{H}$-calibrated with respect to a loss $\ell\_{def,k}$ if, for any $\epsilon>0$, there exists $\delta>0$ such that for all $h\in\mathcal{H}$ and $x\in\mathcal{X}$,
>
> $$ \mathcal{C}\_{\Phi\_{def}}(h,x)<\mathcal{C}^{\ast}\_{\Phi\_{def}}(\mathcal{H},x)+\epsilon\ \Longrightarrow\ \mathcal{C}\_{\ell\_{def,k}}(h,x)<\mathcal{C}^{B}\_{\ell\_{def,k}}(\mathcal{H},x)+\delta. $$
>
> > Is top-k(x) also consistent (as in Theorem 4.7)?
>
> Yes. **The adaptive Top-$k(x)$ formulation is $\mathcal{H}\_k$-consistent under the surrogate loss defined in Definition 5.1**. A related surrogate loss has been proven $\mathcal{H}\_k$-consistent in Theorem 4.6 of [2], and our proof follows the same structure. The extension to our setting only requires adapting the risk decomposition to the learned cardinality function $k\_\theta(x)$, without altering the consistency argument. **We have included a short theoretical justification of this result in the main paper and further discuss the behavior of $k\_\theta(x)$ in our response to Reviewer @VC3c.**

---

> > ### Author Response · Authors · 2025-11-18
> > **Response 2 Reviewer mQGW**
> >
> > > In Figure 5, is there intuition why accuracy for top-k isn't monotonic increasing (in budget) whereas top-k(x) is?
> >
> > **Because our experts have limited and heterogeneous accuracy (see discussion with @jytm), adding more experts does not always improve the aggregated decision—especially under weighted or majority voting**. Including too many low-accuracy or conflicting experts can bias the final output, leading to non-monotonic behavior. In contrast, Top-$k(x)$ adapts the number of consulted experts to each input $x$, typically selecting just enough reliable experts to improve performance, which yields a smoother and monotonic accuracy–budget curve (see analysis of this behavior in our response to Reviewer @VC3c).
> >
> > > wording in line 400 is a little unclear: minimizability gap... vanishes... if the hypothesis set is as rich as the set of all measurable functions
> >
> > We have clarified this sentence in the revised manuscript. The minimizability gap $\mathcal{U}\_{\ell}$ in Theorem 4.7 measures the suboptimality due to restricting the hypothesis class. It vanishes when the distribution is realizable by $\mathcal{H}\_{\pi}$, or equivalently, when $\mathcal{H}\_{\pi}$ is as rich as the set of all measurable functions $\mathcal{H}\_{\pi}^{\mathrm{all}}$. We will rephrase this part of the text to make the condition explicit.
> >
> > ----
> >
> > [1] Awasthi, Pranjal, et al. "Multi-Class $ H $-Consistency Bounds." Advances in neural information processing systems 35 (2022): 782-795.
> >
> > [2] Cortes, Corinna, et al. "Cardinality-aware set prediction and top-$ k $ classification." Advances in neural information processing systems 37 (2024): 18265-18309.

---

> > > ### Comment · Reviewer_mQGW · 2025-11-21
> > >
> > > Thank you for the responses and clarifying my questions. I appreciate the clarification of the up/down arrows, and the additional notation section, which will greatly improve the readability of the paper. I will maintain my positive score.

---

### Official Review · Reviewer_jytm · 2025-11-01

**Soundness:** 3
**Presentation:** 3
**Contribution:** 3
**Rating:** 6
**Confidence:** 2

**Summary:**

This paper studies the Learning-to-Defer (L2D) problem, where a model may defer prediction to external experts. While existing L2D frameworks only allow deferral to a single expert, this work introduces a more general framework that supports deferring to the top-$k$ most cost-effective experts. The authors further develop an adaptive extension, Top-$k(x)$, which selects the optimal number of experts per input based on expert quality, instance difficulty, and consultation cost.

To support this generalization, the paper proposes a surrogate loss that is Bayes-consistent and class-consistent in both one-stage and two-stage L2D settings. Importantly, this surrogate is independent of $k$, meaning that a single model can be trained once and deployed for arbitrary $k$. The authors show that this framework strictly subsumes prior approaches, including selective prediction and Top-1 L2D, and they also unify cascade-style inference under the same formulation. Experimental results suggest that Top-$k$ and Top-$k(x)$ achieve better accuracy–cost trade-offs compared to existing single-expert L2D methods.

**Strengths:**

- The paper introduces a novel and compelling extension of the L2D framework to multi-expert settings, which appears natural and highly relevant in domains requiring collective expert judgment.
- The theoretical development is rigorous, with proofs of Bayes-consistency and class-consistency that generalize known guarantees from Top-1 to Top-$k$ and adaptive settings.
- The unifying perspective on existing L2D, top-$k$ classification, selective prediction, and cascaded models is conceptually valuable and may stimulate further research in this direction.
- Experimental results demonstrate noticeable improvements over prior approaches, particularly under constrained consultation budgets.

**Weaknesses:**

- While the theoretical contributions are strong, empirical evaluation is somewhat limited. The experiments focus on standard benchmarks, and it remains unclear how well the proposed framework performs in practical multi-expert environments with heterogeneous expert behavior or human-in-the-loop settings.
- The implementation details and computational overhead of training with multi-expert routing are not discussed in depth. In practice, the cost of querying multiple experts and potential resource constraints may affect feasibility.
- The method assumes that expert predictions are readily available. It would be valuable to discuss applicability when expert labels are costly or sparse.

**Questions:**

- Can the authors comment on how the framework would behave when expert accuracy varies significantly across different regions of the input space? Is there theoretical or empirical justification that the learned top-$k$ policy will avoid consulting redundant experts?
- How sensitive is the adaptive Top-$k(x)$ approach to the choice of regularization on consultation cost? Are there guidelines for selecting these parameters in practice?
- In real-world settings where experts may be humans, costs and response times are not fixed. Could the authors discuss whether their framework can incorporate stochastic or dynamic consultation costs?

---

> ### Author Response · Authors · 2025-11-18
> **Response Reviewer jytm**
>
> We thank the reviewer for highlighting the **(i) significance of our multi-expert extension,  (ii) the theoretical guarantees, (iii) the value of our unified perspective, (iv) and the improved empirical performance under budget constraints**.
>
> > While the theoretical contributions are strong, empirical evaluation is somewhat limited. The experiments focus on standard benchmarks, and it remains unclear how well the proposed framework performs in practical multi-expert environments with heterogeneous expert behavior or human-in-the-loop settings.
>
> **We would like to emphasize that our evaluation already spans six diverse setups designed to stress both selection and aggregation under challenging conditions.** For instance, in CIFAR-10, each expert is class-conditional: for a subset of classes it predicts correctly with probability $p=0.94$, while for the remaining classes it predicts uniformly at random over the 10 labels. Experts have overlapping assigned classes and correlated errors, and different consultation costs. These conditions are at least as difficult as typical human-in-the-loop scenarios, where expert reliability and cost vary across regions of the input space. **Such heterogeneous and overlapping expert designs are standard in the L2D literature [1–4].**
>
> We also include settings where different trained models serve as experts, following the approach in [5,6], further demonstrating the flexibility of our framework beyond synthetic experts.
>
> > The implementation details and computational overhead of training with multi-expert routing are not discussed in depth. In practice, the cost of querying multiple experts and potential resource constraints may affect feasibility.
>
> We explicitly model the consultation cost of each expert via $\beta_j$ and optimize an accuracy–cost objective:
>
> $$ \mu\_j(x,z)=\alpha\_j \psi(a\_j(x),z)+\beta\_j,\qquad \ell\_{\mathrm{def},k}(\Pi\_k(x),z)=\sum\_{j\in A}\mu\_j(x,z)\mathbf{1}\\{j\in\Pi\_k(x)\\}. $$
>
> **The learned policy therefore queries experts only when the expected gain in accuracy exceeds the consultation cost, directly controlling resource usage.**
>
> A selected expert is only queried during inference (same as [1-6]). Moreover, Top-$k(x)$ adaptively adjusts the number of consulted experts per input under the same cost model, ensuring scalability even in large-expert regimes.
>
> > The method assumes that expert predictions are readily available. It would be valuable to discuss applicability when expert labels are costly or sparse.
>
> We agree that this is an important and practically relevant direction. **The assumption of readily available expert outputs is not specific to our framework but shared by almost all current Learning-to-Defer approaches [1-6]**, which require access to expert responses during training to estimate deferral risk.  We plan to extend our framework to such partially observable settings in future work.
>
> > Can the authors comment on how the framework would behave when expert accuracy varies significantly across different regions of the input space? Is there theoretical or empirical justification that the learned top-$k$ policy will avoid consulting redundant experts?
>
> The learned Top-$k$ policy adapts to these differences, consulting only high-performing experts and avoiding redundant ones. Because selection is based on the expected total cost $\overline{\mu}\_j(x)$ which combines predictive error and consultation cost, **the framework automatically favors the most reliable and cost-effective experts in each region of the input space**.
>
> > How sensitive is the adaptive Top-$k(x)$ approach to the choice of regularization on consultation cost? Are there guidelines for selecting these parameters in practice?
>
> The consultation cost $\beta_j$ is task-dependent and represents the true expense of querying each expert (e.g., latency, human effort, or compute). In practice, we introduce a global regularization parameter $\lambda$ that scales these costs, controlling the trade-off between predictive accuracy and consultation frequency. Larger $\lambda$ discourages querying many experts, favoring cheaper but less redundant decisions, while smaller $\lambda$ prioritizes accuracy over cost.
>
> **The adaptive Top-$k(x)$ policy varies smoothly with respect to $\lambda$: as $\lambda$ increases, the average number of consulted experts decreases monotonically. Please refer also to the discussion with Reviewer @VC3c, where we provide a theoretical analysis of the cardinality function’s behavior.**
>
> We have included in the revised version a brief practical guideline on tuning $\lambda$.

---

> ### Author Response · Authors · 2025-11-18
> **Response 2 Reviewer jytm**
>
> > In real-world settings where experts may be humans, costs and response times are not fixed. Could the authors discuss whether their framework can incorporate stochastic or dynamic consultation costs?
>
> **The framework naturally extends to stochastic or input-dependent costs**. If either experts $m_j$ or their consultation costs $\beta_j$ are random variables, all theoretical results remain valid (see Reviewer @VC3c response): the Bayes rule and consistency proofs rely solely on linearity of expectation (see Lemma 4.5), and thus the same derivations apply to $\mathbb{E}_{\beta_j|X=x}[\mu_j(x,z)]$. Moreover, $\beta_j$ can be made context-dependent, i.e., $\beta_j(x,z)$, to reflect dynamic human latency or resource fluctuations.
>
> We have clarified this generalization in the new version of the paper.
>
> -----
>
> [1] Mozannar, Hussein, and David Sontag. "Consistent estimators for learning to defer to an expert." International conference on machine learning. PMLR, 2020.
>
> [2] Verma, Rajeev, Daniel Barrejón, and Eric Nalisnick. "Learning to defer to multiple experts: Consistent surrogate losses, confidence calibration, and conformal ensembles." International Conference on Artificial Intelligence and Statistics. PMLR, 2023.
>
> [3] Cao, Yuzhou, et al. "In defense of softmax parametrization for calibrated and consistent learning to defer." Advances in Neural Information Processing Systems 36 (2023): 38485-38503.
>
> [4] Mozannar, Hussein, et al. "Who should predict? exact algorithms for learning to defer to humans." International conference on artificial intelligence and statistics. PMLR, 2023.
>
> [5] Mao, Anqi, et al. "Two-stage learning to defer with multiple experts." Advances in neural information processing systems 36 (2023): 3578-3606.
>
> [6] Tailor, Dharmesh, et al. "Learning to defer to a population: A meta-learning approach." International Conference on Artificial Intelligence and Statistics. PMLR, 2024.

---

### Official Review · Reviewer_VC3c · 2025-11-02

**Soundness:** 3
**Presentation:** 3
**Contribution:** 2
**Rating:** 6
**Confidence:** 4

**Summary:**

This paper studies the multi-expert Learning-to-Defer (L2D) problem and extends the classical Top-1 deferral framework to a more general Top-k deferral setting, where the system may consult multiple entities (base model and experts) simultaneously. The authors formulate the Top-k deferral loss, derive the Bayes-optimal Top-k selection rule, and propose the first Bayes-consistent surrogate for Top-k L2D that is independent of k. This allows a single learned scoring function to support any fixed k, as well as an adaptive variant Top-k(x) that chooses a sample-specific number of queried entities. The proposed framework unifies prior one-stage and two-stage L2D results as a special case of k=1, and the authors further validate the approach with extensive experiments demonstrating the benefits of consulting multiple entities and of adapting k to the input.

**Strengths:**

1. The proposed Top-k formulation strictly generalizes prior Top-1 L2D methods, and the theoretical results provably recover previous consistency guarantees when k=1. This provides a clean unification of existing L2D theory under a more general and expressive decision space.

2. Theoretical guarantees are established for both one-stage and two-stage regimes, which is non-trivial. The surrogate achieves Bayes-consistency in both settings, thereby offering a comprehensive and unified theoretical foundation for multi-expert deferral.

3. The empirical results are thorough and convincingly support the claims. The experiments show that Top-k L2D improves over Top-1 and random querying, and that the adaptive Top-k(x) strategy yields further improvements by selecting the appropriate number of experts per input under a given budget constraint.

**Weaknesses:**

1. The paper does not provide theoretical justification for why consulting and aggregating multiple entities should improve final prediction quality. The theory only covers the selection stage, while the aggregation step remains ad-hoc and intuition-based, leaving the core motivation for Top-k partially unsupported from a theoretical perspective.

2. The paper assumes that all experts are deterministic functions of the input $X$ (line 108).
In contrast, the original multi-expert deferral work [1] does not impose this constraint.
This assumption appears restrictive, since an expert depending only on $X$ cannot outperform the most probable label,
which may lead the Bayes-optimal deferral strategy to never defer. While this setting is indeed non-trivial under the zero extra cost or $\mathcal{H}$-consistency framework,
extending the analysis to stochastic experts would significantly strengthen the contribution and practical relevance of this work.

3. In line 171, [2] is described as being related to $\mathcal{H}$-consistency. However, [2] mainly focuses on Bayes consistency and surrogate regret bounds.
It is suggested to clarify this distinction to avoid confusion and to better position the related concepts.

4. Several recent advances in learning to defer [3] and closely related work on classification with rejection [4, 5] are not cited. Including these developments would provide a more complete and up-to-date literature review.

[1].  Rajeev Verma, Daniel Barrejon, and Eric Nalisnick. Learning to defer to multiple experts: Consistent surrogate losses, confidence calibration, and conformal ensembles. AISTATS 2023.

[2]. Hussein Mozannar and David Sontag. Consistent estimators for learning to defer to an expert. ICML 2020.

[3]. Zixi Wei, Yuzhou Cao, and Lei Feng. Exploiting human-ai dependence for learning to defer. ICML 2024.

[4]. Yuzhou Cao, Tianchi Cai, Lei Feng, Lihong Gu, Jinjie Gu, Bo An, Gang Niu, and Masashi Sugiyama. Generalizing consistent multi-class classification with rejection to be compatible with arbitrary losses. NeurIPS 2022

[5]. Corinna Cortes, Giulia DeSalvo, and Mehryar Mohri. Theory and algorithms for learning with rejection in binary classification. Annals of Mathematics and Artificial Intelligence, 92(2), 2024, 277–315.

**Questions:**

Please refer to the weaknesses listed above. I would be glad to revise my rating if the authors address these points in a future version of the paper.

---

> ### Author Response · Authors · 2025-11-18
> **Response to Reviewer VC3c**
>
> We thank the reviewer for the careful reading and for highlighting the strengths of our work. We appreciate the recognition that **(i) our Top-$k$ formulation unifies and extends prior L2D theory, (ii)  that our consistency guarantees cover both one-stage and two-stage regimes, (iii) and that our experiments convincingly demonstrate the benefits of Top-$k$ and adaptive Top-$k(x)$**. We address the reviewer’s questions below.
>
> > The paper does not provide theoretical justification for why consulting and aggregating multiple entities should improve final prediction quality. The theory only covers the selection stage, while the aggregation step remains ad-hoc and intuition-based, leaving the core motivation for Top-k partially unsupported from a theoretical perspective.
>
> **The cardinality-aware deferral loss introduced in Definition 5.1 is in fact Bayes-consistent and $\mathcal{H}_k$-consistent (see [6])**. We provide a brief analysis below, which we hope will clarify this point.
>
> For any $x \in \mathcal{X}$ and $k \in \\{1,\dots,|\mathcal{A}|\\}$, the conditional
> risk of selecting the top-$k$ experts is
>
> $$
> \mathcal{C}\_{\ell\_{\mathrm{card}}}(k) := \mathbb{E}[\ell\_{\mathrm{card}}(\Pi\_k(x),k,x,Z) \mid X=x] = \mathbb{E}[d(\Pi\_k(x),x,Z) \mid X=x] + \lambda  \xi(\sum\_{i=1}^{k}\beta\_{[i]\downarrow\_\pi}).
> $$
> The Bayes-optimal cardinality function is therefore
>
> $$
> k\_\theta^B(x)
> \in
> \arg\min\_{k \in \{1,\dots,|\mathcal{A}|\}}
> \{
> \mathbb{E}[d(\Pi\_k(x),x,Z) \mid X=x] + \lambda \xi(\sum\_{i=1}^{k}\beta\_{[i]\downarrow\_\pi})
> \}.
> $$
> For $k \ge 2$, define $
> \delta \mathcal{C}\_{\ell\_{\mathrm{card}}}(k)
> := \mathcal{C}\_{\ell\_{\mathrm{card}}}(k) - \mathcal{C}\_{\ell\_{\mathrm{card}}}(k-1)$ and
> $S\_k := \sum\_{i=1}^{k}\beta\_{[i]\downarrow\_\pi}.$
>
> A simple computation gives
>
> $$
> \delta \mathcal{C}\_{\ell\_{\mathrm{card}}}(k)=\underbrace{\mathbb{E}[d(\Pi\_k(x),x,Z) - d(\Pi\_{k-1}(x),x,Z) \mid X=x]}_{:=\delta D\_x(k)} + \lambda (\xi(S\_k) - \xi(S\_{k-1})).
> $$
> Thus, for any $k \in \{1,\dots,|\mathcal{A}|-1\}$,
> $$ \mathcal{C}\_{\ell\_{\mathrm{card}}}(k+1) \le \mathcal{C}\_{\ell\_{\mathrm{card}}}(k) \Longleftrightarrow \delta D\_x(k+1) + \lambda (\xi(S\_{k+1}) - \xi(S\_k)) \le 0, $$
> where $\delta D\_x(k+1) := \mathbb{E}[d(\Pi\_{k+1}(x),x,Z) - d(\Pi\_k(x),x,Z) \mid X=x]$.
>
> In words, moving from $k$ to $k+1$ strictly decreases the conditional risk if and only if the marginal reduction in expected error, $-\delta D\_x(k+1)$, is at least as large as the marginal regularization cost, $\lambda (\xi(S\_{k+1}) - \xi(S\_k))$.
>
> **This shows that consulting and aggregating multiple experts is not ad hoc: for any fixed aggregation rule $d$, it is Bayes-optimal to choose $k\_\theta^B(x) > 1$ precisely when additional experts yield a net decrease of the conditional risk $\mathcal{C}\_{\ell\_{\mathrm{card}}}(k)$.**
>
> We have added this analysis in Appendix A12.
>
> > The paper assumes that all experts are deterministic functions of the input (line 108). In contrast, the original multi-expert deferral work [1] does not impose this constraint. This assumption appears restrictive, since an expert depending only on cannot outperform the most probable label, which may lead the Bayes-optimal deferral strategy to never defer. While this setting is indeed non-trivial under the zero extra cost or -consistency framework, extending the analysis to stochastic experts would significantly strengthen the contribution and practical relevance of this work.
>
> **Our results do not require determinism**. As [1], let expert $m\_j$ use internal randomness $M\_j \sim P(M\_j \mid X=x, Y=y)$ and denote its prediction by $a\_j(x,M\_j)$. The conditional top-$k$ deferral risk becomes
>
> $$ \mathcal{C}\_{\ell\_{\mathrm{def},k}}(\Pi\_k(x),y) = \sum\_{j\in A}\overline{\tilde{\mu}}\_j(x,y)\mathbf{1}\\{j\in\Pi\_k(x)\\}, \overline{\tilde{\mu}}\_j^B(x) = \mathbb{E}\_{Y\mid X=x}[\tilde{\mu}\_j(x,Y)], $$
>
> with per-entity cost given by its conditional expectation
>
> $$ \tilde{\mu}\_j(x,y) = \alpha\_j \mathbb{E}\_{M\_j\mid X=x, Y=y}[\psi(a\_j(x,M\_j),y)] + \beta\_j. $$
>
> **With this substitution, all statements are unchanged**: Lemma 4.2 (set-valued, cost-additive loss), Lemma 4.5 (Bayes Top-$k$ selects the $k$ smallest $\overline{\tilde{\mu}}\_j^B(x)$), and Theorem 4.7 (consistency for one-stage $\mathcal{H}\_h$ and two-stage $(\mathcal{H}\_r,\mathcal{H}\_g)$). The proofs use linearity of expectation and bounded per-entity costs, not determinism.

---

> > ### Author Response · Authors · 2025-11-18
> > **Response 2 to Reviewer VC3c**
> >
> > > In line 171, [2] is described as being related to $\mathcal{H}$-consistency. However, [2] mainly focuses on Bayes consistency and surrogate regret bounds. It is suggested to clarify this distinction to avoid confusion and to better position the related concepts.
> >
> > We will clarify the scope. Theorem 3.3 is an $\mathcal H$-consistency bound [7]: it relates surrogate and target excess risks within a restricted class and includes the minimizability gap $\mathcal{U}\_{\ell}(\mathcal H)$. When $\mathcal{U}\_{\ell}(\mathcal H)=0$ (rich $\mathcal H$), it reduces to Bayes consistency. By contrast, [2] establishes Bayes consistency for the logistic loss; it does not provide an $\mathcal H$-consistency inequality of our form. We have adjusted the wording to reflect this distinction in the new version.
> >
> > > Several recent advances in learning to defer [3] and closely related work on classification with rejection [4, 5] are not cited. Including these developments would provide a more complete and up-to-date literature review.
> >
> > Thank you for the suggestion. We have added [3] and the reject-option papers [4,5].
> >
> >
> > -----
> > [6] Corinna Cortes, Anqi Mao, Christopher Mohri, Mehryar Mohri, and Yutao Zhong. Cardinality-
> > aware set prediction and top-$k$ classification. NeurIPS 2025
> >
> > [7] Awasthi, Pranjal, et al. "Multi-Class $ H $-Consistency Bounds." NeurIPS 2022

---

### Official Review · Reviewer_28BC · 2025-11-02

**Soundness:** 3
**Presentation:** 2
**Contribution:** 2
**Rating:** 2
**Confidence:** 4

**Summary:**

The paper considers a setting where, given a feature, a learner makes predictions or defers to a set of experts. They formulate the problem abstractly, as learning a model for selecting k assets, where each asset can either be a label or an expert. The hypotheses selected by the learner score each of these assets, and the loss incurred by the learner is a function of the top-k highest scoring assets. The specific loss function they consider is a sum over the individual losses of each of the selected assets. The individual loss is a cost-sensitive loss combining a predictive loss for the label selected by the model (or expert), and possibly a consultation loss for deferring to an expert.

They describe how their setting generalizes previous work in what they call the 1-stage and 2-stage settings. They use ideas from the literature to bound the cost sensitive loss with a surrogate, and prove statistical consistency. They state a version of the problem where k need not be fixed. Finally, they conclude with empirical experiments that compare two methods for fixed k and adaptive k.

**Strengths:**

Strength: I like that the paper unifies the 1-stage and 2-stage settings in a single framework that considers experts and labels to simply be decisions ("assets" in the language of the paper) that need to be scored by some model.

**Weaknesses:**

Weakness 1. The theoretical portion of the paper considers a loss function that sums the individual losses of each asset. As a modeling assumption, this has a number of problems. In settings where there is a good expert, or when the model is good at identifying labels without experts, the model is forced to select a number of sub-optimal assets by the setting, and is strictly penalized for doing so. The aggregation also does not allow the learner to meaningfully improve its prediction (e.g. by taking a major vote). I believe the authors mitigate this somewhat by introducing the adaptive-k setting and by considering more interesting aggregation strategies in their experiments. However, it does call into question the utility of the theory around a somewhat degenerate loss function.

Weakness 2: The surrogate loss optimized by the authors is independent of k (Corollary 4.4), and recovers the top-1 solution from previous literature. Thus, the authors' algorithm learns a hypothesis for the top-1 problem, and uses the k highest scores found by that hypothesis to pick its assets. In light of Weakness #1, the most interesting aspect of this problem over top-1 is to learn a model whose scores preserve rank over suboptimal assets, which this solution expressly avoids modeling.

Weakness 3: The experiments primarily compare the paper's methods against each other. I am not convinced that there are not more adequate baselines. Cortez et al. 2024, which the authors cite, for example, seems very relevant.

Weakness 4: The paper casts and re-casts the problem a number of times before finally arriving at the formulation in lines 200-208 and 226-240. I think the presentation can be improved by presenting this more succinctly. In my opinion, the observation that this generalizes prior work need not take up more than a paragraph or two.

**Questions:**

Why is Lemma 4.2 stated as a Lemma, when it seems more like a definition? The proof of the Lemma does not seem to help resolve it for me. The proof seems to state that the uniform top-k loss upper bounds the standard deferral loss (which #1 seems obvious, and #2 is not in the statement of Lemma 4.2). It is also interspersed with asides about special cases, which are not proving anything.

---

> ### Author Response · Authors · 2025-11-18
> **Response Reviewer 28BC**
>
> We thank the reviewer for recognizing the value of unifying the one-stage and two-stage formulations within a single scoring-based framework. Please find our clarifications below.
>
> > The theoretical portion of the paper considers a loss function that sums the individual losses of each asset. As a modeling assumption, this has a number of problems. In settings where there is a good expert, or when the model is good at identifying labels without experts, the model is forced to select a number of sub-optimal assets by the setting, and is strictly penalized for doing so. The aggregation also does not allow the learner to meaningfully improve its prediction (e.g. by taking a major vote). I believe the authors mitigate this somewhat by introducing the adaptive-k setting and by considering more interesting aggregation strategies in their experiments. However, it does call into question the utility of the theory around a somewhat degenerate loss function.
>
> **Embedding aggregation into the loss (e.g., optimizing majority-vote accuracy) makes the objective combinatorial**: with three experts one must consider AT LEAST $\\{\\{1\\}\\,\\{2\\}\\,\\{3\\}\\,\\{1,2\\}\\,\\{1,3\\}\\,\\{2,3\\}\\,\\{1,2,3\\}\\}$ — seven committees for a single $x$; in general $2^{|A|}-1$ subsets (if we consider the order important, it's even more), each inducing a different vote outcome. This destroys decomposability and precludes calibrated convex surrogates, rendering both learning and theory intractable.
>
> **A joint/aggregated loss also collapses per-entity signals into a single vote, preventing learning of the base classifier $h$ and breaking the unified one-stage setting**: the objective would depend only on the aggregate, not on individual predictions $a\_j(x)$ or scores $\pi(x,\\cdot)$.
>
> Our additive formulation decouples selection and aggregation: $\pi$ learns whom to consult under an additive cost $$ \ell\_{\mathrm{def},k}(\Pi\_k(x),z)=\sum\_{j\in\Pi\_k(x)}\mu\_j(x,z) $$ with $$ \mu\_j(x,z)=\alpha\_j\psi(a\_j(x),z)+\beta\_j, $$ while the aggregation rule $d$ (majority/weighted/mean) is chosen post-selection without retraining $\pi$. This preserves decomposability (and thus calibrated surrogate design and $\mathcal{H}$-consistency) and implicitly avoids the $2^{|A|}$ aggregation explosion.
>
> **Note that the Bayes Top-$k$ Selector selects the top-$k$ entities with the lowest cost, which is exactly what is desired in those problems.**
>
> > The surrogate loss optimized by the authors is independent of k (Corollary 4.4), and recovers the top-1 solution from previous literature. Thus, the authors' algorithm learns a hypothesis for the top-1 problem, and uses the k highest scores found by that hypothesis to pick its assets. In light of Weakness 1, the most interesting aspect of this problem over top-1 is to learn a model whose scores preserve rank over suboptimal assets, which this solution expressly avoids modeling.
>
> The surrogate’s $k$-independence is a theoretical strength, not a limitation. As shown in Cor. 4.4, this property arises algebraically from the Bayes Top-$k$ rule (Lemma 4.5): the Bayes-optimal policy ranks entities by their conditional expected cost $\mu_j^B(x)$, and the Top-$k$ set is obtained by taking the $k$ smallest values. Hence, learning a scoring function $\pi(x,j)$ that preserves this order simultaneously yields the Bayes-optimal set for all $k$.
>
> **The model therefore does not “learn a Top-1 policy” but a consistent ranking over all entities whose induced Top-$k$ selections are Bayes-optimal for every $k$. This is actually a very desirable property of top-$k$ losses (see [1,2,3]), we absolutely do not want to retrain a policy for every $k$.**
>
> > The experiments primarily compare the paper's methods against each other. I am not convinced that there are not more adequate baselines. Cortez et al. 2024, which the authors cite, for example, seems very relevant.
>
> **Cortes et al. (2024) [4] address the standard top-$k$ classification problem, not Learning-to-Defer**. Their objective is to rank labels, whereas our framework defers among heterogeneous experts. **To our knowledge, no existing work implements top-$k$ deferral or multi-expert selection with cost-sensitive guarantees, making direct comparison impossible**. We therefore benchmarked against the only meaningful alternatives: random selection and established top-1 Learning-to-Defer baselines in one-stage and two-stage setting.
>
> > The paper casts and re-casts the problem a number of times before finally arriving at the formulation in lines 200-208 and 226-240. I think the presentation can be improved by presenting this more succinctly. In my opinion, the observation that this generalizes prior work need not take up more than a paragraph or two.
>
> We will make it more concise in the final version of the paper.

---

> > ### Author Response · Authors · 2025-11-18
> > **Response 2 Reviewer 28BC**
> >
> > > Why is Lemma 4.2 stated as a Lemma, when it seems more like a definition? The proof of the Lemma does not seem to help resolve it for me. The proof seems to state that the uniform top-k loss upper bounds the standard deferral loss (which 1 seems obvious, and 2 is not in the statement of Lemma 4.2). It is also interspersed with asides about special cases, which are not proving anything.
> >
> > You're right. We have changed it to a Definition. Thanks for the helpful suggestion.
> >
> > We hope these clarifications fully address the reviewer’s comments and clearly justify the design choices in our framework.
> >
> >
> > ----
> >
> > [1] Maksim Lapin, Matthias Hein, and Bernt Schiele. Analysis and optimization of loss functions
> > for multiclass, top-k, and multilabel classification. IEEE transactions on pattern analysis and
> > machine intelligence, 40(7):1533–1554, 2017.
> >
> > [2] Maksim Lapin, Matthias Hein, and Bernt Schiele. Loss functions for top-k error: Analysis and
> > insights. In Proceedings of the IEEE conference on computer vision and pattern recognition, pp.
> > 1468–1477, 2016.
> >
> > [3] Forest Yang and Sanmi Koyejo. On the consistency of top-k surrogate losses. In International
> > Conference on Machine Learning, pp. 10727–10735. PMLR, 2020.
> >
> > [4] Corinna Cortes, Anqi Mao, Christopher Mohri, Mehryar Mohri, and Yutao Zhong. Cardinality-
> > aware set prediction and top-$k$ classification. In The Thirty-eighth Annual Conference on Neural Information Processing Systems

---

### Meta-Review · Area_Chair_xpjy · 2026-01-07

**Summary:**

All but one reviewer viewed the paper positively (borderline positive) and their concerns/questions were largely well addressed during the rebuttal. The most critical reviewer (28BC) would certainly have influenced the final decision regarding this paper strongly. Considering their review and the authors responses, I think that their concerns were well addressed and they should thus have increased their score significantly. In summary, the paper would not be a clear accept but a borderline accept. Still I think the paper is interesting and extends the L2D framework into an existing direction and should be accepted if possible.

**Reviewer Concerns:**

Reviewer 28BC
- Choice of additive per-asset loss (penalizes suboptimal assets; no meaningful aggregation like majority vote)
  - Partially addressed (the authors justify decomposability and post-hoc aggregation) but the modeling limitation concern likely remains
- Surrogate is independent of $k$ and reduces to Top-1, failing to model rankings among suboptimal asset
  - Addressed: authors argue $k$-independence is a strength; learning a consistent global ranking yields Bayes-optimal Top-$k$ for all $k$
- Baselines are inadequate; missing relevant comparisons (e.g., Cortes et al.)
  - Partially addressed: the authors argue those works are Top-$k$ classification, not L2D; direct comparison not possible

Reviewer VC3c
- No theory for why consulting/aggregating multiple entities improves prediction; aggregation seems ad hoc.
  - Addressed: authors provide a cardinality-aware analysis showing Bayes-optimality conditions for adding experts
- Assumes deterministic experts; restrictive vs. prior work allowing stochastic experts
  - Addressed: the authors clarify that results extend to stochastic experts
- Confusion between $\mathcal{H}$-consistency and Bayes consistency in related work
  - Addressed: clarified distinctions and wording
- Missing citations on recent L2D and rejection work
  - Addressed: added (some of the) suggested references

Reviewer jytm
- Empirical scope limited; unclear performance in practical heterogeneous/human-in-the-loop settings
  - Partially addressed: authors argue setups are diverse and harder than typical; evaluation is in line with standards in the literature
- Implementation/overhead not deeply discussed; resource constraints may affect feasibility
  - Partially addressed: the authors explain cost modeling and inference-only querying; scalability via adaptive $k$
- Assumes expert predictions are readily available; need discussion for costly/sparse labels
  - Still standing and argued to be in line with existing work; still acknowledged as promising future work
- Behavior under region-varying expert accuracy; redundancy avoidance.
  - Addressed: selection minimizes expected total cost to avoid redundant experts
- Sensitivity of adaptive Top-$k(x)$ to cost regularization; practical tuning.
  - Addressed
- Incorporating stochastic/dynamic consultation costs
  - Addressed: the authors explained how the framework extends to this setting via context-dependent/random costs

Reviewer mQGW
- Possible error/incompleteness in Lemma A.6 proof.
  - Addressed: provided explicit derivation
- Is Top-$k(x)$
  - Addressed: yes, justification added
- Non-monotonic accuracy vs. budget for fixed Top-$k$ vs. monotonic for Top-$k(x)$
  - Addressed by providing corresponding intuition

Reviewer U2o1
- Training complexity, especially for adaptive $k$; clarity on sequential vs. joint training and hyperparameter tuning
  - Partially addressed: authors clarify sequential training and tuning; the reviewer notes that complexity concern remains
- Lack of real-world multi-expert evaluation
  - Still standing: authors note absence of public datasets; reviewer considers concern still standing
- Assumptions behind guarantees; realizability and finite-sample behavior
  - Addressed: provide convergence-rate bound
- Training strategy for $k$-selection
  - Addressed: sequential post-hoc learning
- Aggregation decision rule details:
  - Addressed: explicit strategies: any-correct, majority, weighted; reported across experiments
- Sensitivity to cost parameters and regularization.
  - Addressed: behavior explained; trends indicated

Summary of outstanding issues:
- Modeling concern about additive loss vs. joint aggregation [28BC]: partially resolved
- Experimental baselines breadth [28BC]: partially resolved
- Practical/real-world evaluation and implementation complexity [jytm, U2o1]: partially resolved

**Reviewer Scores:**

* Reviewer 28BC might have increased their score but likely not to an accept level; however I found the authors answers convincing
* Reviewer VC3c might have increased their score as their concerns were addressed well by the reviewers and they indicated that they would increase their score
* Reviewer jytm might have kept their score as some concerns are still standing
* Reviewer mQGW would have kept their score
* Reviewer U2o1 would have kept their score

---

### Decision · Program_Chairs · 2026-01-26

Accept (Poster)